# WAYS v1: A hydrological model for root zone water storage simulation on a global scale

Ganquan Mao[1] and Junguo Liu[1]

[1]School of Environmental Science and Engineering, Southern University of Science and Technology, Shenzhen 518055, China

**Correspondence:** Junguo Liu (junguo.liu@gmail.com, liujg@sustech.edu.cn)

**Abstract.** The soil water stored in the root zone is a critical variable for many applications, as it plays a key role in several hydrological and atmospheric processes. Many studies have been conducted to obtain reliable information on soil water in the root zone layer. However, most of them are mainly focused on the soil moisture within a certain depth rather than the water stored in the entire rooting system. In this work, a hydrological model named WAYS is developed to simulate the root zone water storage (RZWS) on a global scale. The model is based on a well validated lumped model and has now been extended to a distribution model. To reflect the natural spatial heterogeneity of the plant rooting system across the world, a key variable that influences RZWS, i.e., root zone storage capacity (RZSC), is integrated into the model. The newly developed model is first evaluated based on runoff and RZWS simulations across ten major basins. The results show the ability of the model to mimic RZWS dynamics in most of the regions through comparison with proxy data, the Normalized Difference Infrared Index (NDII). The model is further evaluated against station observations, including flux tower and gauge data. Despite regional differences, generally good performances are found for both the evaporation and discharge simulation. Compared to existing hydrological models, WAYS's ability to resolve the field-scale spatial heterogeneity of RZSC and simulate RZWS may offer benefits for many applications, e.g., agriculture and land-vegetation-climate interaction investigations. However, the results from this study suggest an additional evaluation of RZWS is required for the regions where the NDII might not be the correct proxy.

## 1 Introduction

Soil moisture is one of the critical variables in earth system dynamics (Sheffield and Wood, 2008) and is claimed an Essential Climate Variable by the World Meteorological Organization due to its key role in several hydrological and atmospheric processes (Legates et al., 2011). The soil water stored in the plant root zone is of great importance in some fields of application, e.g., agriculture, as it represents the reservoir of the plant available water and mediates numerous subsurface processes (Sabater et al., 2007; Wang et al., 2015; Cleverly et al., 2016). A fundamental limiting factor that constrains crop yields is the water resources in the root zone (Tobin et al., 2017). The water stored in the root zone is also directly linked with one of the important water resources for ecosystems, i.e., green water resources, as green water is defined as the water that originates from precipitation that is stored in the unsaturated soil and eventually consumed by plants through evapotranspiration (Falkenmark and Rockström, 2006; Liu and Yang, 2010).

There are several methods for soil moisture estimation, including in situ measurements, satellite-based approaches and model simulation (Paulik et al., 2014; Dumedah et al., 2015; Colliander et al., 2017; Zhang et al., 2017; Berg et al., 2017). Especially in recent years, a variety of specific sensors and systems have been built for global soil moisture measurement, e.g., the Advanced Microwave Sounding Radiometer for Earth Observation System (AMSR-E) as well as the AMSR-2 (Njoku et al.,

2003) and the Soil Moisture Ocean Salinity (SMOS) (Kerr et al., 2010) and Soil Moisture Active Passive (SMAP) missions (Entekhabi et al., 2010). These sensors are able to provide continuous estimations of soil moisture worldwide.

Obtaining reliable root zone water storage is still challenging, as it cannot be directly observed (González-Zamora et al., 2016). Satellite remote sensing itself can only detect the soil water at the surface layer (in most cases with a depth of 5 cm) and has the shortcoming that it cannot look at the deep soil profile (Petropoulos et al., 2015). Considerable effort has been

made recently by researchers to retrieve root zone soil moisture (RZSM), a variable that is very close to RZWS. Tobin et al. (2017) developed an exponential filter to leverage the remotely sensed surface soil moisture to produce RZSM. Faridani et al. (2017) and Baldwin et al. (2017) applied a soil moisture analytical relationship (SMAR) model to generate RZSM, where the surface soil moisture is the input. Apart from remote sensing-based approaches, hydrological models and land surface models are important tools for moisture simulation, as they work both in the past and in future scenarios (Xia et al., 2014; Sheikh

et al., 2009; Albergel et al., 2018; Samaniego et al., 2018). Additionally, many studies estimate RZSM by combining remotely sensed soil moisture with different models using data assimilation techniques (Rebel et al., 2012; Renzullo et al., 2014a, b). However, all these studies estimated the root zone soil moisture until a certain depth, e.g., 100 cm, thus still retaining the drawback of being unable to accurately calculate the water stored in the entire root zone layer. Moreover, the root depth is location dependent and can reach a depth of more than 30 meters (Fan et al., 2017).

Alternatively, RZSM can also be obtained by investigating and applying the relationship between RZSM and different vegetation indices derived from MODIS or Landsat satellites, e.g., the Normalized Difference Vegetation Index (NDVI) and the Enhanced Vegetation Index (EVI) (Santos et al., 2014; Wang et al., 2007; Schnur et al., 2010; Liu et al., 2012). Nevertheless, their work either stays at a certain soil depth assuming a consistent rooting depth or estimates only the water content ratio assuming a homogeneous soil profile, rather than the water amount covering the entire spatially heterogeneous rooting system

(Fan et al., 2017). To date, studies that directly focus on root zone water storage are still rare.

Recently, Sriwongsitanon et al. (2016) investigated the relation between root zone water storage and the Normalized Difference Infrared Index and found a promising correspondence between them in a river basin in Thailand, especially in the dry seasons, where water stress exists. However, the NDII is an index value that reflects only the dynamics of RZWS rather than the absolute value. Moreover, remote sensing-based approaches only allow historical analyses. While the ability to predict RZWS,

usually by employing models, is still missing, which is crucial for impact studies, e.g., agricultural drought analysis (Keyantash and Dracup, 2002), the work of Sriwongsitanon et al. (2016) provided enlightenment for future RZWS-related studies, as their findings support NDII as a potential proxy for RZWS. This is critical for mitigating the major challenge, i.e., the lack of direct observation of root zone water storage for evaluation, in the field of hydrological modeling.

In this study, a global hydrological model is developed to simulate root zone water storage, a key variable for ecohydrological

studies. Though many global hydrological models (GHMs) have already been developed, most of them are similar regarding the

general hydrological component simulations (Sood and Smakhtin, 2015), and the developed model has its unique scheme for root zone process depiction; thus, it enables RZWS simulations with the ability to consider the global spatially heterogeneous rooting systems. The model has input requirements similar to most of the existing GHMs and can also generate general hydrological variables in addition to RZWS. Since it simulates RZWS, which is of great importance for both hydrology and

ecology, it will be further developed in the future for water and ecosystem-related applications. The newly developed model is named the Water And ecoSystem Simulator (WAYS). The ultimate goal of this study is to test the feasibility of WAYS for RZWS simulation on a global scale, an added-value feature useful for many applications.

## 2 Model Description

### 2.1 General Overview

WAYS is a hydrological model implementation in Python. It is a process-based model that assumes water balance at the grid cell level. The development of WAYS is based on a lumped conceptual model with an HBV-like model structure, called the FLEX model (Fenicia et al., 2011; Gao et al., 2014a). The FLEX model has been widely used and validated at the basin scale to simulate the soil moisture content and root zone water storage (Gao et al., 2014b; Nijzink et al., 2016; de Boer-Euser et al., 2016; Sriwongsitanon et al., 2016). Benefiting from its flexible modeling framework, we have now extended it to a spatially

distributed global hydrological model. In addition, some improvements have been made to increase the model capacity at the global scale, e.g., a more sophisticated soil water storage capacity strategy and more land cover support.

WAYS is a raster-based model that calculates the water balance and simulates the hydrological processes in a fully distributed way. It works on a daily time step, and the model structure consists of five conceptual reservoirs: the snow reservoir $S_w$ (mm) representing the surface snow storage, the interception reservoir $S_i$ (mm) expressing the water intercepted in the canopy, the

root zone reservoir $S_r$ (mm) describing the root zone water storage in the unsaturated soil, the fast response reservoir $S_f$ (mm), and the slow response reservoir $S_s$ (mm). Two lag functions are applied to describe the lag time from the storm to peak flow ($T_{lagF}$) and the lag time of recharge from the root zone to the groundwater ($T_{lagS}$). In addition to the water balance equation, each reservoir also has process functions to connect the fluxes entering or leaving the storage compartment (so-called constitutive functions). Figure 1 provides a schematic representation of how the vertical water balance is modeled in WAYS,

and the basic equations are shown in Table 1. In Figure 1, the flowchart represents the conceptualized hydrological cycle in the model, and the schematic drawing shows the corresponding water fluxes and stocks in the real world. Since some of the fluxes are intermediate variables, they are shown in the flowchart but not visualized in the schematic drawing. For instance, $R_f$ is the generated preferential runoff in the root zone layer before the split of the runoff into surface runoff and subsurface runoff. The effective precipitation $P_e$ is the sum of snowmelt and precipitation throughfall. The conceptualized hydrological cycle of the

model can be briefly described as follows. The precipitation that can drop as rainfall or snowfall depends on the temperature. The snowfall will be stored in the snow reservoir, and the rainfall will be intercepted by the canopy before it reaches the surface. After the interception, the rainfall penetrates the canopy and reaches the surface as precipitation throughfall. The effective precipitation that consists of the throughfall and the snowmelt will partially infiltrate into the soil, and the rest runs

away as runoff. The runoff is then split into surface runoff and subsurface runoff depending on the texture. A part of the infiltration will be stored in the soil for plants, and the rest will percolate into the deep soil and reach the groundwater table as groundwater recharge. The parameters that regulate the different simulation steps are described below, and the changes we made to the original FLEX model are highlighted. The original lumped model FLEX has 28 parameters in total that consider four land use types in the basin (Gao et al., 2014a). To reduce the computation cost of calibration and avoid overfitting issues at the global scale, some calibrated parameters are replaced by the empirical values from the literature, e.g., the snowmelt ratio $F_{DD}$, the capacity of the interception reservoir $S_{i,max}$, the groundwater recharge factor $f_s$ and the maximum value of groundwater recharge $R_{s,max}$.

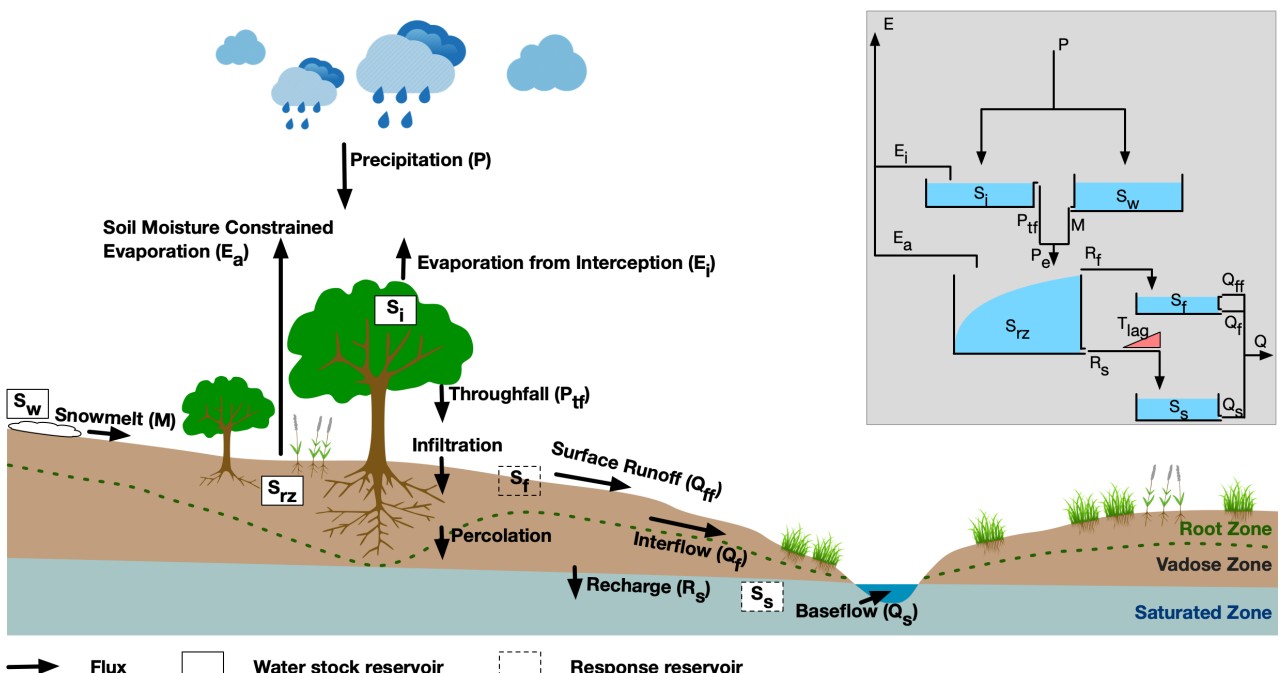

**Figure 1.** Model structure of WAYS

## 2.2 Interception and Snow Routine

In the WAYS model, the precipitation is allowed to be intercepted by the canopy or stored as snow before entering into the root zone reservoir.

Interception occurs during the days with rain when the temperature is above the threshold temperature $T_t$. The interception reservoir stores the precipitation intercepted by the canopy before it reaches the soil that will directly evaporate back into the atmosphere. The canopy water balance equation is shown in Eq. (1), where the precipitation $P$ (mm/day) is the inflow, and the precipitation throughfall $P_{tf}$ (mm/day) and the interception evaporation $E_i$ (mm/day) are the outflows. The calculation of the precipitation throughfall $P_{tf}$ is simply based on comparing the rainfall $P_r$ (mm/day) to the water already stored in the

interception reservoir $S_i$ (mm) and the capacity of the interception reservoir $S_{i,max}$ (mm) (Eq. (2)). In the FLEX model, the interception evaporation $E_i$ is assumed to be the potential evaporation, and the interception capacity is a calibrated parameter. In WAYS, the interception evaporation $E_i$ is calculated based on the potential evaporation $E_0$ (mm/d), the storage of the interception reservoir $S_i$ (mm) and the interception reservoir storage capacity $S_{i,max}$ (Eq. (3)) following Deardorff (1978).

The interception capacity $E_{i,max}$ is calculated by using Eq. (4), where $m_c$ is 0.3 mm and $L$ is the leaf area index, which is calculated based on a modified phenology model in Jolly et al. (2005) obtained by replacing the original vapor pressure stress function with the soil moisture in the model (Wang-Erlandsson et al., 2014).

The snow simulation is based on a simple degree-day algorithm (Rango and Martinec, 1995) that has been successfully applied in hydrological models in many studies (Comola et al., 2015; Bair et al., 2016; Krysanova and Hattermann, 2017).

The water balance in the snow reservoir is described in Eq. (5), and the constitutive equations are shown in Eq. (6) in Table 1. Below the threshold temperature $T_t$ (°C), the precipitation $P$ (mm/day) falls as snow $P_s$ (mm/day) and is added to the snow storage $S_w$ (mm). Above the threshold temperature $T_t$, snow melts if it is available at a certain ratio per degree ($F_{DD}$). Both the threshold temperature $T_t$ and the snowmelt ratio $F_{DD}$ are parameters calibrated in the FLEX model. Following Müller Schmied et al. (2014), $T_t$ is set to 0 °C, and $F_{DD}$ is set for different land cover classifications from 1.5 mm/d per degree to

6 mm/d per degree in WAYS. It is also important to be aware that the snowmelt water is conceptualized in the model as directly infiltrating into the soil in the model, thus effectively bypassing the interception reservoir.

## 2.3   Root Zone Routine

The effective root zone routine is the core of the WAYS model. It controls both the evapotranspiration and the runoff generation by precipitation partitioning. Similar to the interception and snow routine, the change of root zone water storage $S_{rz}$ (mm)

over time $t$ (day) is described in Eq. (7), with effective precipitation $P_e$ (mm/day) as the inflow and soil moisture constrained evaporation $E_a$ (mm/day) and runoff $R$ (mm/day) as outflows. In the FLEX model, the runoff generation is calculated based on the widely used beta function of the Xinanjiang model (Zhao, 1992), which is a function of the relative soil moisture in the unsaturated soil layer. The beta function for calculation of runoff in WAYS is replaced by a modified version from the work of Sriwongsitanon et al. (2016) to link the function to the water storage in the root zone layer. Depending on the root zone water

storage $S_{rz}$, a part of the effective precipitation turns into runoff, and the rest infiltrates into the soil and recharges the root zone layer. The runoff coefficient is determined by both the relative soil water content $S_{rz}/S_{rz,max}$ in the root zone and the shape parameter $\beta$ describing the spatial process heterogeneity over pixels at the global scale. The root zone storage capacity used in WAYS is derived by applying the method in Wang-Erlandsson et al. (2016), which calculates the soil moisture deficit based on satellite-based evaporation and precipitation, while it is a calibrated parameter in FLEX.

The soil moisture constrained evaporation, sometimes also known as actual evapotranspiration, is calculated as a function of the potential evaporation leftover $E_0 - E_i$ (mm/day), the relative soil water content $S_{rz}/S_{rz,max}$, the shape parameter $\beta$ and the scale parameter $C_e$, which indicates the fraction of $S_{rz,max}$ above which the transpiration is no longer limited by soil moisture stress. Since the root zone routine connects both the runoff and evapotranspiration and the runoff generation function has been modified, the actual evapotranspiration function in WAYS is also accordingly revised from the original one in the

FLEX model (Sriwongsitanon et al., 2016). The scale parameter $C_e$ is set to 0.5 in the FLEX model when applied at the basin scale, and it becomes a calibrated parameter in WAYS at the global scale.

## 2.4 Slow Response Routine

The water balance in the slow response reservoir $S_s$ (mm) is simple, with the groundwater recharge $R_s$ (mm/day) as the inflow
and baseflow $Q_s$ (mm/day) as the outflow (Eq. (11)). The groundwater recharge $R_s$ is depicted in WAYS by applying the splitter function described in Eq. (12). It separates the runoff into preferential flow and groundwater recharge based on the groundwater recharge factor $f_s$, which ranges between 0 and 1. In WAYS, the amount of groundwater recharge is also limited by the maximum groundwater recharge $R_{s,max}$ (mm/day) for each grid cell, which is specified by the soil texture, while there is no constraint on the maximum value for groundwater recharge in the FLEX model. The values of $R_{s,max}$ used in this study
are 7, 4.5 and 2.5 for sandy soil, loamy soil and clayey soil, respectively, following Döll and Fiedler (2008).

The groundwater recharge factor $f_s$ is a calibrated parameter in the FLEX model, while in WAYS, it is now determined by applying the approach developed by Döll and Fiedler (2008), in which it is a function of global digital maps of the slope, soil texture, geology, and permafrost. The method is simple and computationally inexpensive, and it was validated at the global scale in many subsequent publications, e.g., Döll et al. (2012) and Döll et al. (2014). All the related parameters are provided in
look-up tables in the work of Döll and Fiedler (2008), and the only changes we made are that the input data of the groundwater recharge method, e.g., the global relief data and the global soil texture map, have been accordingly updated based on the newly available data (Hanasaki et al., 2018). The outflow of the slow response reservoir, i.e., the baseflow, is modeled with the function described in Eq. 13, where the baseflow coefficient $K_s$ is set to 100 globally following the work of Döll et al. (2003).

## 2.5 Fast Response Routine

The preferential flow $R_f$ (mm/day) is routed directly into the fast response reservoir $S_f$ (mm), and it is divided into surface runoff $Q_{ff}$ (mm/day) and interflow $Q_f$ (mm/day). The water balance in the fast response reservoir is shown in Eq. (14). In the FLEX model, it is assumed that the preferential flow is routed into the fast response reservoir based on a lag-function that represents the time lag between a storm and preferential runoff generation. In WAYS, we have assumed that the preferential flow will route into the fast response reservoir directly without any delay globally, as it is run at the daily time scale.

Similar to the slow response reservoir, the fast response reservoir is also set as a linear-response reservoir, representing a linear relationship between water storage and water release. The surface runoff generation is only active when the storage of the fast response reservoir exceeds the specified threshold $S_{ftr}$, with a generation ratio $K_{ff}$ (Eq. (16)), while the interflow $Q_f$ is simply calculated in proportion to the already stored water in the fast response reservoir using the fraction of $1/K_f$ (Eq. (17)).

## 2.6 Additional Model Adaptation

In addition to the abovementioned model description, some modifications and assumptions are necessary to adapt the model to the global scale. In WAYS, the actual evaporation from open water bodies is assumed to be the potential evapotranspiration, and the freezing of open water bodies is not considered in the model. Potential evapotranspiration is derived by the Hamon equation (Hamon, 1961) in the FLEX model, and it is now replaced by the using the Penman-Monteith FAO 56 PM method (Allen et al., 1998) for the following reason. The Hamon method is found to have less robustness in different climatic conditions as well as drawbacks in the daily variability of the PET simulation due mainly to the relatively simple equation in the Hamon method, as it only employs the average air temperature as an input (Bai et al., 2016; Droogers and Allen, 2002). In contrast, the Penman-Monteith FAO 56 PM method is based on fundamental physical principles and is found to be the most reliable method for potential evapotranspiration estimation when sufficient meteorological data exist (Chen et al., 2005; Kingston et al., 2009). The Penman-Monteith FAO 56 PM method is recommended by FAO and other studies based on thorough analyses of PET method intercomparisons (Allen et al., 1998; Jian biao et al., 2005; Vörösmarty et al., 1998). In the FLEX model, capillary rise from groundwater is also considered. However in WAYS, the feature for capillary rise simulation is currently disabled, as it cannot be taken into account when no information is available at the global scale. The WAYS model is written in Python version 3.6. To benefit from a supercomputer, the model is designed with full support for parallel computation.

**Table 1.** Water balance and constitutive equations used in WAYS

| Reservoirs | Water balance equations | | Constitutive equations | | Reference |
|---|---|---|---|---|---|
| Interception reservoir | $\dfrac{dS_i}{dt} = P_r - E_i - P_{tf}$ | (1) | $P_{tf} = max(0, P_r - (S_{i,max} - S_i))$ | (2) | - |
| | | | $E_i = E_p \left(\dfrac{S_i}{S_{i,max}}\right)^{2/3}$ | (3) | Deardorff (1978) |
| | | | $S_{i,max} = m_c L$ | (4) | Wang-Erlandsson et al. (2014) |
| Snow reservoir | $\dfrac{dS_w}{dt} = \begin{cases} -M & \text{if } T > T_t \\ P_s & \text{if } T \le T_t \end{cases}$ | (5) | $M = \begin{cases} min(S_w, F_{DD}(T - T_t)) & \text{if } T > T_t \\ 0 & \text{if } T \le T_t \end{cases}$ | (6) | Rango and Martinec (1995) |
| Root zone reservoir | $\dfrac{dS_{rz}}{dt} = P_e - R - E_a$ | (7) | $P_e = P_{tf} + M$ | (8) | - |
| | | | $\dfrac{R}{P_e} = 1 - \left(1 - \dfrac{S_{rz}}{(1+\beta)S_{rz,max}}\right)^{\beta}$ | (9) | Sriwongsitanon et al. (2016) |
| | | | $E_a = (E_0 - E_i) \cdot min\left(1, \dfrac{S_{rz}}{C_e S_{rz,max}(1+\beta)}\right)$ | (10) | Sriwongsitanon et al. (2016) |
| Slow response reservoir | $\dfrac{dS_s}{dt} = R_s - Q_s$ | (11) | $R_s = min(f_s R, R_{s,max})$ | (12) | Döll and Fiedler (2008) |
| | | | $Q_s = S_s/K_s$ | (13) | Döll et al. (2003) |
| Fast response reservoir | $\dfrac{dS_f}{dt} = R_f - Q_{ff} - Q_f$ | (14) | $R_f = R - R_s$ | (15) | - |
| | | | $Q_{ff} = max(0, S_f - S_{ftr})/K_{ff}$ | (16) | - |
| | | | $Q_f = S_f/K_f$ | (17) | - |

Note: all the time scale-dependent parameters need to be divided by $\Delta t$ to make the equations dimensionally correct and suitable for any other time scales.
- in the reference column indicates that the formula is taken from the FLEX model.

## 3 Model Setup

For the assessment of model performance, the WAYS model is applied at the global scale with a spatial resolution of 0.5 degrees for the historical period from 1971 to 2010. Two simulations are conducted based on two products of the global root zone storage capacity from Wang-Erlandsson et al. (2016). The model is calibrated in the period 1986-1995 and validated in the period of 2001-2010 depending on the availability of the reference data.

### 3.1 Driving Data

#### 3.1.1 Meteorological Data

The model is driven by the climate data set from the Global Soil Wetness Project 3 (Kim, 2017), GSWP3 (http://hydro.iis.u-tokyo.ac.jp/GSWP3/), for the historical period from 1971 to 2010. The GSWP3 data set is generated based on the Twentieth Century Reanalysis Project (Compo et al., 2011). It has been proven to be able to represent realistic submonthly variability over the entire 20*th* century (1901-2010) and has been used as a forcing data set in several other hydrological modeling studies (Veldkamp et al., 2017; Masaki et al., 2017; Liu et al., 2017; Tangdamrongsub et al., 2018). The climate variables used in this study include precipitation, minimum temperature, maximum temperature, relative humidity, surface downwelling longwave radiation, surface downwelling shortwave radiation and wind speed at 10 meters. All the variables are on a daily scale and have a 0.5 degree spatial resolution; in addition, the wind speed at 10 meters is converted to the wind speed at 2 meters based on the conversion function in Allen et al. (1998), as it is required by the Penman-Monteith FAO 56 PM method for potential evapotranspiration calculations.

#### 3.1.2 Land Use Data

The land cover data that we used are the Global Mosaics of the standard MODIS land cover type data product (MCD12Q1) with a spatial resolution of 0.5 degrees in the year of 2001, which are derived from the IGBP Land Cover Type Classification (17 classes) and are reprojected into geographic coordinates of latitude and longitude on the WGS 1984 coordinate reference system (Friedl et al., 2010).

#### 3.1.3 Root Zone Storage Capacity

The root zone storage capacity (RZSC) data are a crucial parameter in WAYS. The global root zone storage capacity data used in this study are from Wang-Erlandsson et al. (2016), derived by using the "Earth observation-based" method. This method determines the soil moisture deficit at the global scale by using the state-of-the-art observation-based precipitation data and satellite-based evaporation data, under the assumption that vegetation optimizes its root zone storage capacity to bridge critical dry periods and does not invest more in its roots than necessary. This method has been well justified (de Boer-Euser et al., 2019) and overcomes the shortcomings of the traditional methods (look-up table approach; field observation-based approach) at the global scale, such as data scarcity, location bias, and risks of unlikely vegetation and soil combinations due to data uncertainty

(Feddes et al., 2001). The method has been shown to increase the model performance at both the basin and global scales (Gao et al., 2014b; Nijzink et al., 2016; Wang-Erlandsson et al., 2016). Moreover, it has been proven to be able to produce plausible root zone storage capacity in boreal regions by investigating the relationship between RZSC and numerous environmental factors, including climate variables, vegetation characteristics, and catchment characteristics (de Boer-Euser et al., 2019).

Since there are two global root zone storage capacity products ($S_{R,CHIRPS-CSM}$ and $S_{R,CRU-SM}$) presented by Wang-Erlandsson et al. (2016) based on different precipitation and evaporation data sets and there is no preference for either product, in this study, both RZSC products are used. $S_{R,CHIRPS-CSM}$ covers the latitudes from 50°N to 50°S and is derived based on the United States Geological Survey (USGS) Climate Hazards Group InfraRed Precipitation with Stations (CHIRPS) precipitation data (Funk et al., 2014) and the ensemble mean of three satellite-based global-scale evaporation data sets: the
Commonwealth Scientific and Industrial Research Organization (CSIRO) Moderate Resolution Imaging Spectroradiometer (MODIS) Reflectance Scaling EvapoTranspiration (CMRSET) data (Guerschman et al., 2009), the Operational Simplified Surface Energy Balance (SSEBop) data (Senay et al., 2013), and the MODIS evapotranspiration (MOD16) data (Mu et al., 2011). $S_{R,CRU-SM}$ covers the latitudes from 80°N to 56°S and is derived by using the Climatic Research Unit Time Series version 3.22 precipitation data (Harris et al., 2014) together with the ensemble mean of only SSEBop and MOD16 because CMRSET
overestimates evaporation at high latitudes (Wang-Erlandsson et al., 2016). Since Wang-Erlandsson et al. (2016) suggested that a Gumbel normalization of RZSC by land cover types with different return periods could further improve the model performance, we have accordingly adjusted the RZSC in this study. The two selected global root zone storage capacity products are shown in Figure S13, and their mean latitudinal values are shown in Figure S14. Similar patterns and magnitudes of RZSC can be found, and there is good agreement between the two products at different latitudes, especially at low latitudes around
the equator, where the products reflect the fluctuation with high consistency. A large difference is seen mainly in the northern midlatitude area, where the absolute difference in percentage is still less than 20%.

### 3.2   Calibration Data

The WAYS model has a few parameters, and while some of them are obtained independently from the literature, some have to be determined by model calibration (see Table 2). The WAYS model is calibrated against the ISLSCP II UNH/GRDC
Composite Monthly Runoff data (Fekete et al., 2011) from 1986 to 1995 at a 0.5 degree resolution, which are composite runoff data that combine simulated water balance model runoff estimates and monitored river discharge. The ISLSCP II UNH/GRDC Composite Monthly Runoff data also comprise a standard data set in the second phase of ISIMIP (Inter-Sectoral Impact Model Inter-comparison Project) (ISIMIP2a) (Warszawski et al., 2014) for calibration and validation, as it assimilates discharge measurements at gauge stations and preserves the spatial specificity of the water balance while being constrained by the station
observations. The data can be downloaded from The Oak Ridge National Laboratory Distributed Active Archive Center (ORNL DAAC) (https://daac.ornl.gov/cgi-bin/dsviewer.pl?ds_id=994).

**Table 2.** Parameter ranges of the WAYS model

| Parameter | Range | Literature | Parameter | Range |
|-----------|-------|------------|-----------|-------|
| $S_{i,max}$ | distributed | Wang-Erlandsson et al. (2014) | $\beta$ | (0, 2) |
| $S_{rz,max}$ | distributed | Wang-Erlandsson et al. (2016) | $C_e$ | (0.1, 0.9) |
| $R_{s,max}$ | 7/4.5/2/5 (Sand/Loam/Clay) | Döll and Fiedler (2008) | $K_f$ | (1, 40) |
| $K_s$ | 100 | Döll et al. (2003) | $K_{ff}$ | (1, 9) |
| $f_s$ | distributed | Döll and Fiedler (2008) | $S_{ftr}$ | (10, 200) |
| $F_{DD}$ | distributed | Müller Schmied et al. (2014) | $T_{lag}$ | (0, 5) |
| $T_t$ | 0 | Müller Schmied et al. (2014) | | |

## 3.3 Validation Data

In this study, the ERA-Interim/Land runoff data are used for validation of the runoff simulation, and the Normalized Difference Infrared Index (NDII) is used for the validation of the WAYS model for root zone water storage simulation. Considering the time period of coverage of both data sets (ERA-Interim/Land: 1979-2010, NDII: 2000-present) and the study period (1971-
2010) of this work, the period 2001-2010 is selected as the validation period. For runoff evaluation, ISIMIP2a simulations are also included, as they use the same climate forcing as our study in the same period. The purpose of inclusion of the ISIMIP2a simulations for comparison can be found in the model evaluation section (see Section 4).

### 3.3.1 ERA-Interim/Land Runoff Data

ERA-Interim/Land is a global land surface reanalysis data set produced by the European Centre for Medium-Range Weather
Forecasts (ECMWF) (Balsamo et al., 2015). The gridded data set ERA-Interim/Land is selected for model evaluation mainly because the current version of the WAYS model does not include a runoff routing model on the global scale. Therefore, the results are not comparable with observed gauge data. Since the ERA-Interim/Land data set is well assessed with a quality check through comparison with ground-based and remote sensing observations, it has been used as reference data for many studies (Xia et al., 2014; Dorigo et al., 2017). ERA-Interim/Land runoff data are one of the variables in the ERA-Interim/Land
reanalysis data set and are widely used as benchmark data (Alfieri et al., 2013; Orth and Seneviratne, 2015; Reichle et al., 2017) due to their good agreement with the Global Runoff Data Centre (GRDC) data set and large improvement compared to the ERA-Interim runoff reanalysis data, which were used as one of the reference data sets (Wang-Erlandsson et al., 2014; Balsamo et al., 2015). The ERA-Interim/Land runoff data used in this study were downloaded from the ECMWF website (http://apps.ecmwf.int/datasets/) at a 0.5 degree resolution and a daily scale from 2001 to 2010.
It should be noted that there are other reanalysis runoff data available, such as ERA-Interim, GLDAS and NECP. However, they show low robustness based on the available research results. For instance, GLDAS v1.0-CLM was found to overestimate runoff globally, and GLDAS v1.0-Noah generated more surface runoff over the northern middle-high latitudes (Lv et al., 2018). GLDAS v2.0-Noah showed a significant underestimation trend in exorheic basins (Wang et al., 2016). The snowmelt-runoff

peak magnitude simulated by GLDAS v2.1-Noah was found to be excessively high in June and July (Lv et al., 2018). NECP runoff was found to be too high during the winter and too low during the summer in the Mississippi River Basin (Roads and Betts, 2000). ERA-Interim was found to be less close to the observed stream flows compared with ERA-Interim/Land data (Balsamo et al., 2015).

### 3.3.2   NDII Data

NDII was developed by Hardisky et al. (1983) for satellite imagery analysis based on calculations of the ratios of different values between infrared reflectance (NIR) and shortwave infrared reflectance (SWIR). NDII has been found to have a strong correlation with the vegetation water content and canopy water thickness (Serrano et al., 2000; Jackson et al., 2004; Hunt and Yilmaz, 2007; Wilson and Norman, 2018). It can also be used to effectively determine the water stress of plants by taking advantage of the property of shortwave infrared reflectance, which has a negative relationship with leaf water content because of the large absorption by leaves (Steele-Dunne et al., 2012; Friesen et al., 2012; van Emmerik et al., 2015). Recently, Sriwongsitanon et al. (2016) found a promising linkage between NDII and the root zone water storage. Even though NDII reflects the dynamics of RZWS better in moisture stress periods than in moisture stress-free periods, the general good correspondence between NDII and RZWS indicates that NDII has potential as a proxy for RZWS. Therefore, in this study, NDII is used as the benchmark to assess the performance of the model in RZWS depiction.

NDII is calculated by applying the following equation from Hardisky et al. (1983):

$$NDII = \frac{\rho_{0.85} - \rho_{1.65}}{\rho_{0.85} + \rho_{1.65}} \tag{18}$$

where $\rho_{0.85}$ is the reflectance at the $0.85\,\mu m$ wavelength and $\rho_{1.65}$ is the reflectance at the $1.65\,\mu m$ wavelength. NDII is a normalized index that ranges between $-1$ and $1$. A low value of NDII indicates high canopy water stress, which also reflects that there is less water content in the root zone (Sriwongsitanon et al., 2016).

In our work, NDII is computed based on the satellite data MODIS level 3 surface reflectance product (MOD09A1) (Vermote, 2015), which provides an estimate of the surface spectral reflectance of Terra MODIS Bands 1 through 7 corrected for atmospheric conditions such as gases, aerosols, and Rayleigh scattering in the sinusoidal projection. The MOD09A1 product is available on an 8-day temporal scale with a $500\,m$ spatial resolution globally from 2000-02-24 until the present. Each MOD09A1 pixel contains the value selected from all the acquisitions within the 8-day composite on the basis of high observation coverage, low viewing angle, the absence of clouds or cloud shadow, and aerosol loading. The satellite image processing and NDII calculation are performed by using the Google Earth Engine platform (http://earthengine.google.com). Some of the MOD09A1 images are missing. In total, 452 NDII rasters are generated for the validation period (2001-2010).

### 3.4   Calibration Strategy

A global parameter optimization algorithm (Tolson and Shoemaker, 2007), dynamically dimensioned search (DDS), has been applied in this study for model parameter calibration. DDS is designed for computationally expensive optimization problems

and has been used in many studies related to distributed hydrological model calibration at global and regional scales (Moore et al., 2010; Kumar et al., 2013; Rakovec et al., 2016; Nijzink et al., 2018; Smith et al., 2018).

Since the reference data, i.e., ISLSCP II UNH/GRDC data, are at a monthly temporal scale, the runoff simulated by WAYS in the calibration period (1986-1995) is also averaged to the monthly scale for consistency. The criterion of fit for calibration is the Nash-Sutcliffe efficiency coefficient (NSE), and the DDS optimization algorithm is run with 2000 iterations for each grid cell for parameter estimation, as suggested by the author of DDS (Tolson and Shoemaker, 2007).

## 4    Model Evaluation

To evaluate model performance, simulated runoff and root zone water storage values are compared to the reference data (see Section 3.3) for the validation period (2001-2010) in ten major river basins of the world considering the coverage of the root zone storage capacity products ($S_{R,CHIRPS-CSM}$ covers only the latitudes from $50°$N to $50°$S).

### 4.1    Runoff Evaluation

WAYS simulated runoff values are compared to the ERA-Interim/Land runoff as well as to the multimodel global runoff simulations from ISIMIP2a. ISIMIP is a community-driven global platform that supports model intercomparison studies at both global and regional scales, while ISIMIP2a focuses on the historical period, and all the models are driven by four state-of-the-art climate forcing factors (Warszawski et al., 2014). Since WAYS uses the same driving data as the ISIMIP2a models and the ISIMIP2a simulations have been widely discussed in many studies (Schewe et al., 2014; Müller Schmied et al., 2016; Gernaat et al., 2017; Zaherpour et al., 2018), we also perform a comparison between WAYS and the ISIMIP2a models to further evaluate our model. To make the climate forcing consistent with the WAYS model, only the GSWP3-driven simulations are used for comparison. The evaluation is performed at the monthly scale, even though the WAYS model simulates the runoff at the daily scale, because only monthly runoff data are available for some of the ISIMIP2a models (Warszawski et al., 2014).

Figure 2 shows the time series of runoff from reference data and different models. WAYS_CRU in the legend indicates the runoff simulated by the WAYS model with root zone storage capacity product $S_{R,CRU-SM}$, and WAYS_CHIRPS denotes the simulation with RZSC product $S_{R,CHIRPS-CSM}$. First, it can be seen that the two WAYS simulations with different RZSC products show extremely good correspondence in all selected basins. This result is consistent with the investigation of RZSC data sets in Section 3.1.3, where there is a high consistency in the two used products, which even show that RZSC itself naturally exhibits high variability along the latitudes (see Figure S13 and S14). This result confirms the robustness of the RZSC products that we used in our WAYS model for runoff simulation. The results show good agreements between WAYS simulations and the reference data, i.e., ERA-Interim/Land in the selected basins, while the ISIMIP2a models present stark differences in the simulated runoff. For example, the ISIMIP2a models show a clear trend of overestimation in some of the basins (Mississippi, Ganges, Yangtze, Parana and Murray Darling), where the spread of the runoff ensembles is also large. This result occurs partly because some of the ISIMIP2a models are not calibrated at all (Zaherpour et al., 2018), whereas WAYS is calibrated to a Composite Monthly Runoff data set that assimilates the monitored river discharge (Fekete et al., 2011).

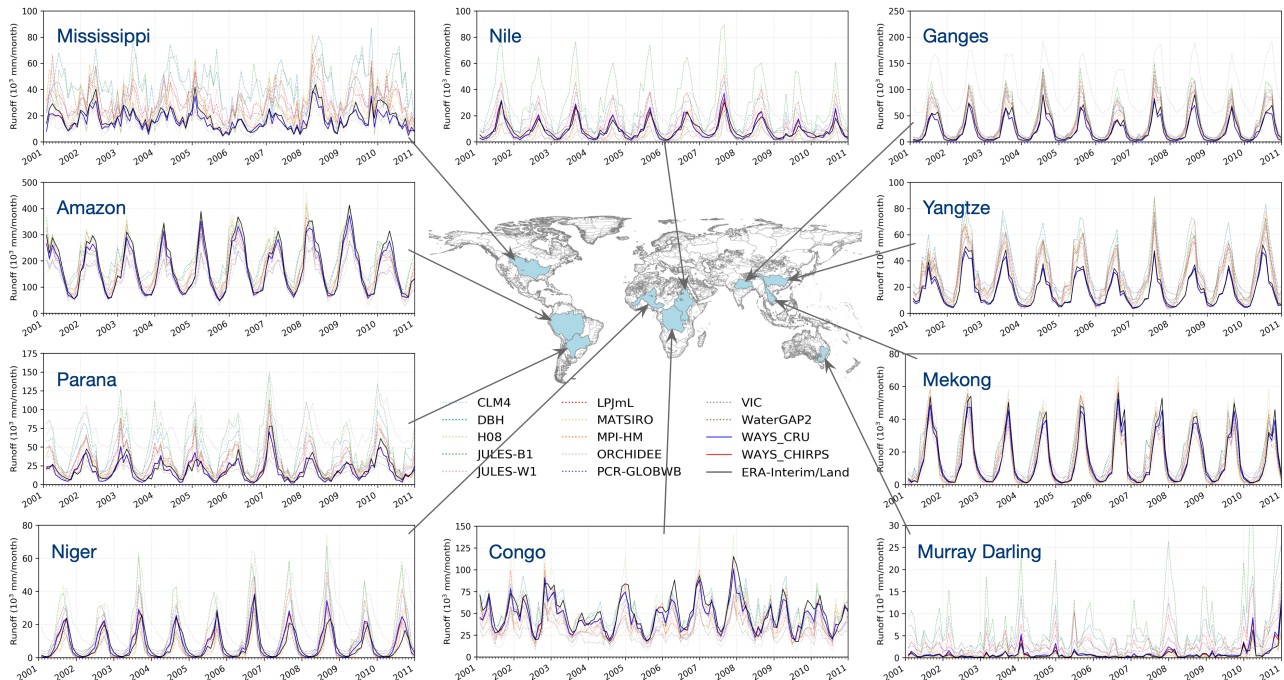

**Figure 2.** Time series of monthly runoff simulated by WAYS and the ISIMIP2a models, as well as the reference data. The basins highlighted in the world map indicate the selected catchments for model evaluation. The solid lines in blue and red indicate the WAYS simulations with two different RZSC products. The solid line in black indicates the ERA-Interim/Land data, and dashed lines represent the ISIMIP2a model simulations. In some plots, the red line is not visible and is covered by the blue line due to the small differences between the two WAYS runs. WAYS is calibrated using Composite Monthly Runoff data, while the ISIMIP2a models are not calibrated for the simulation.

In the Mekong River basin, all the models show a high consistency in monthly runoff generation, with a narrow spread of the ensemble. This result may be due to the natural characteristics of the Mekong River, i.e., highly predictable timing and size of the wet-season peak. In addition, precipitation in this region is concentrated in an extremely regular wet-season peak under the impact of tropical monsoons (Adamson et al., 2009). The manner in which WAYS outperforms the other models is also observed in the northernmost (Mississippi) and southernmost (Murray Darling) catchments of our selected basins, while the ISIMIP2a models show extremely large differences in the runoff simulations with large uncertainties. The good performance is particularly highlighted in the Murray Darling basin, where the monthly runoffs are extremely low, due mainly to the anthropogenic climate impacts (Cai and Cowan, 2008; Potter and Chiew, 2011), which is extremely difficult for other models to capture without overestimation (see Figure 2). A slight overestimation is found in the WAYS model in two African basins, i.e., the Nile and Niger. This result can be explained by the general overestimation of the precipitation value in climate forcing data GWSP3 in these regions (Muller Schmied et al., 2016). In these two regions, the ISIMIP2a simulations also show dramatic overestimations. In contrast, the models show a trend of underestimation in another African basin, the Congo. This result might be caused by both the quality of precipitation and the complexity of natural processes here (Tshimanga and Hughes, 2014).

Wang-Erlandsson et al. (2014) reported in their work that Congo precipitation and runoff estimates are particularly uncertain in general. It is worth highlighting that the WAYS model can still capture the monthly variability of runoff in this basin well.

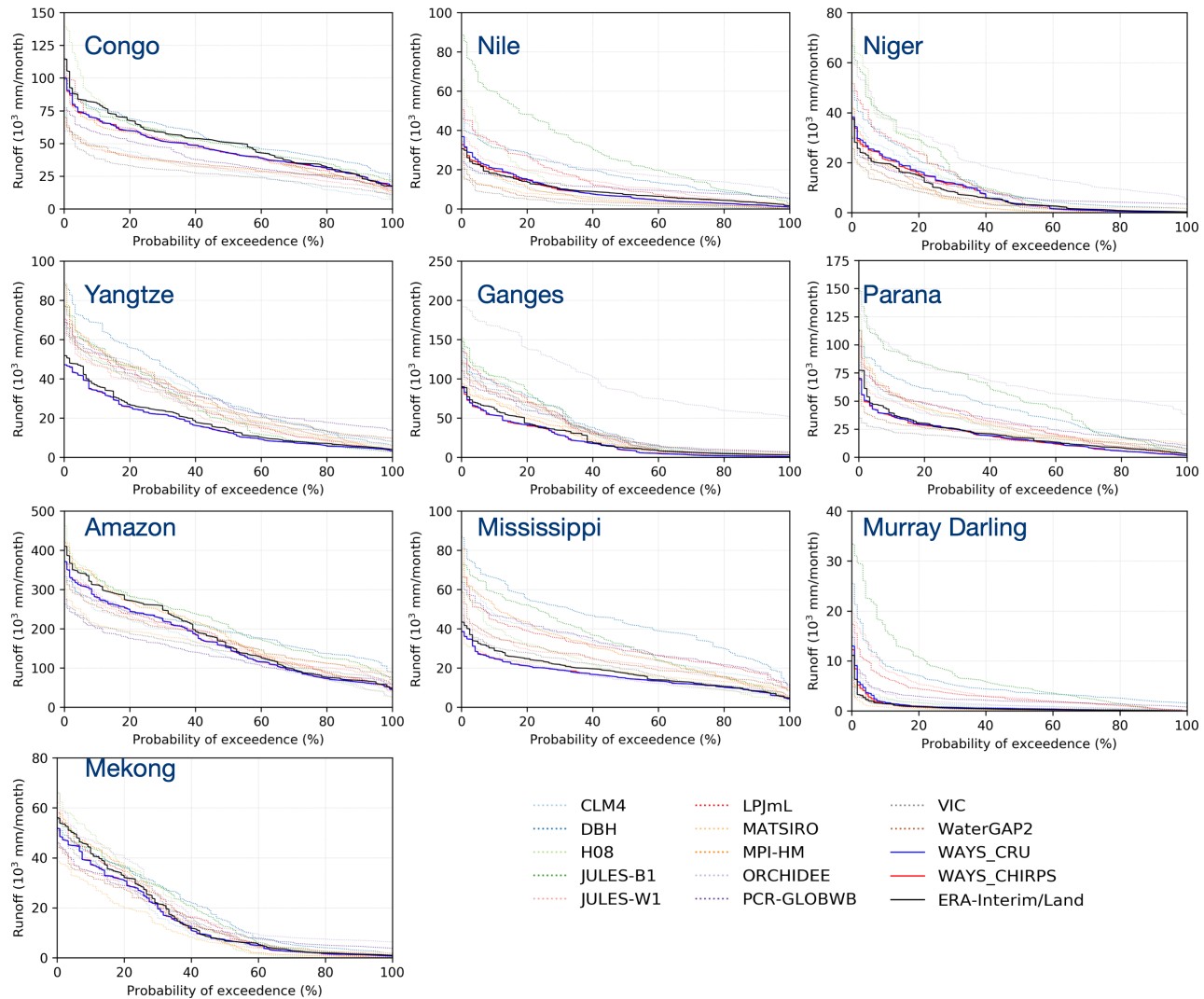

**Figure 3.** The probability of exceedance for monthly runoff simulated by different models as well as the reference data in ten selected basins. The solid lines in blue and red indicate WAYS simulations with two different RZSC products. The solid line in black indicates the ERA-Interim/Land data, and dashed lines represent the ISIMIP2a model simulations. In some plots, the red line is not visible and is covered by the blue line due to the small differences between the two WAYS runs. WAYS is calibrated using Composite Monthly Runoff data, while the ISIMIP2a models are not calibrated for the simulation.

To evaluate the ability of the WAYS model to replicate the distribution, a comparison study on the probability of exceedance is conducted, and the result is shown in Figure 3. The probability of exceedance reveals the model performance at different magnitudes. With a visual inspection, we can see that WAYS is able to reproduce the runoff distribution well with a good match

to the ERA-Interim/Land data, especially in the Congo, Parana and Mississippi basins, while the ISIMIP2a model simulated runoff is skewed differently than that of the ERA-Interim/Land runoff distribution. In a few basins (Nile, Ganges, Parana and Mississippi), some of the ISIMIP2a models even show a bear-sized shift of distribution relative to the reference data, highlighting that these models struggle to simulate the monthly runoff at all different magnitudes. In the Nile and Niger basins,
WAYS also shows a slight offset for both simulations, but it still lies within the uncertainty range. The results also show a large uncertainty in the runoff simulations in the upper tails, which reflects the larger deviation in the high values produced than in the middle- and low-value simulations for the models. Such biases in reproducing the runoff distribution in the ISIMIP2a models, in turn, deliver large ensemble spreads in the time series.

To further assess the performance of the WAYS model, three general metrics for runoff comparison are selected for the eval-
uation, i.e., the Nash-Sutcliffe Efficiency (NSE), root mean squared error (RMSE) and percent bias (PBIAS). The estimated scores from the monthly runoff time series for WAYS and the ISIMIP2a models are presented in Figure 4. For better comparison, the NSE values are converted to the 1-NSE values; thus, numbers closer to 0 indicate better performance. The model performance of WAYS is generally better than that of the ISIMIP2a models, and the estimated scores based on different criteria are also close to the benchmarks. The 1-NSE comparison (Figure 4 (a)) indicates that the model performance of WAYS in
the selected basins, except for the Niger and Nile, is particularly favorable compared to the ISIMIP2a models. In these basins, both of the WAYS simulations (WAYS_CRU and WAYS_CHIRPS) are ranked in the top five (14 model simulations in total in the comparison). In six basins, both of the WAYS simulations have 1-NSE metric scores less than 0.3, resulting in a value of NSE of greater than 0.7. In the Yangtze, Amazon and Mekong, the WAYS model is even ranked as the best one, with both of the simulations outperforming the others. The relatively low performance of WAYS in the Niger and Nile is the result of
the model slightly overestimating the middle and high runoff values (Figure 3). The RMSE comparison (Figure 4 (b)) delivers information similar to the 1-NSE comparison, in which WAYS shows generally better performance. In the Amazon, all the model simulations show large RMSE due to the large value of monthly runoff in this catchment. By examining the percent bias (Figure 4 (c)), it is evident that the WAYS model performs well in most of the basins, as the scores of the two WAYS simulations are close to the benchmark. A relatively poor performance of the WAYS model in the percent bias assessment is
found in the Murray Darling basin, with PBIAS values of approximately 100%, but they are still within the uncertainty range based on a check with other models. This large value may be caused by the extremely low runoff-induced low value of the benchmark, in which a slight difference in the absolute value will cause a large difference in the percentage.

Combining the time series analysis, most commonly used metric examination in hydrology and probability of exceedance assessment, our results show a comprehensive assessment of the model performance in runoff simulation. The strong performance
of WAYS with the subtle difference between the runoff simulation and reference data in all the tests indicates the particularly favorable applicability of WAYS in runoff simulation across major basins. Even though relatively poor performances are found in two African basins, the biases are still within the uncertainty range based on investigations of other models. Such a trend of overestimation could also be explained by the overestimation of the precipitation value in the forcing data in these regions (Muller Schmied et al., 2016). In addition, it is worth acknowledging that global hydrological models show large differences
in runoff simulations across basins. Previous studies emphasized that large ensemble spreads from GHMs could be caused by

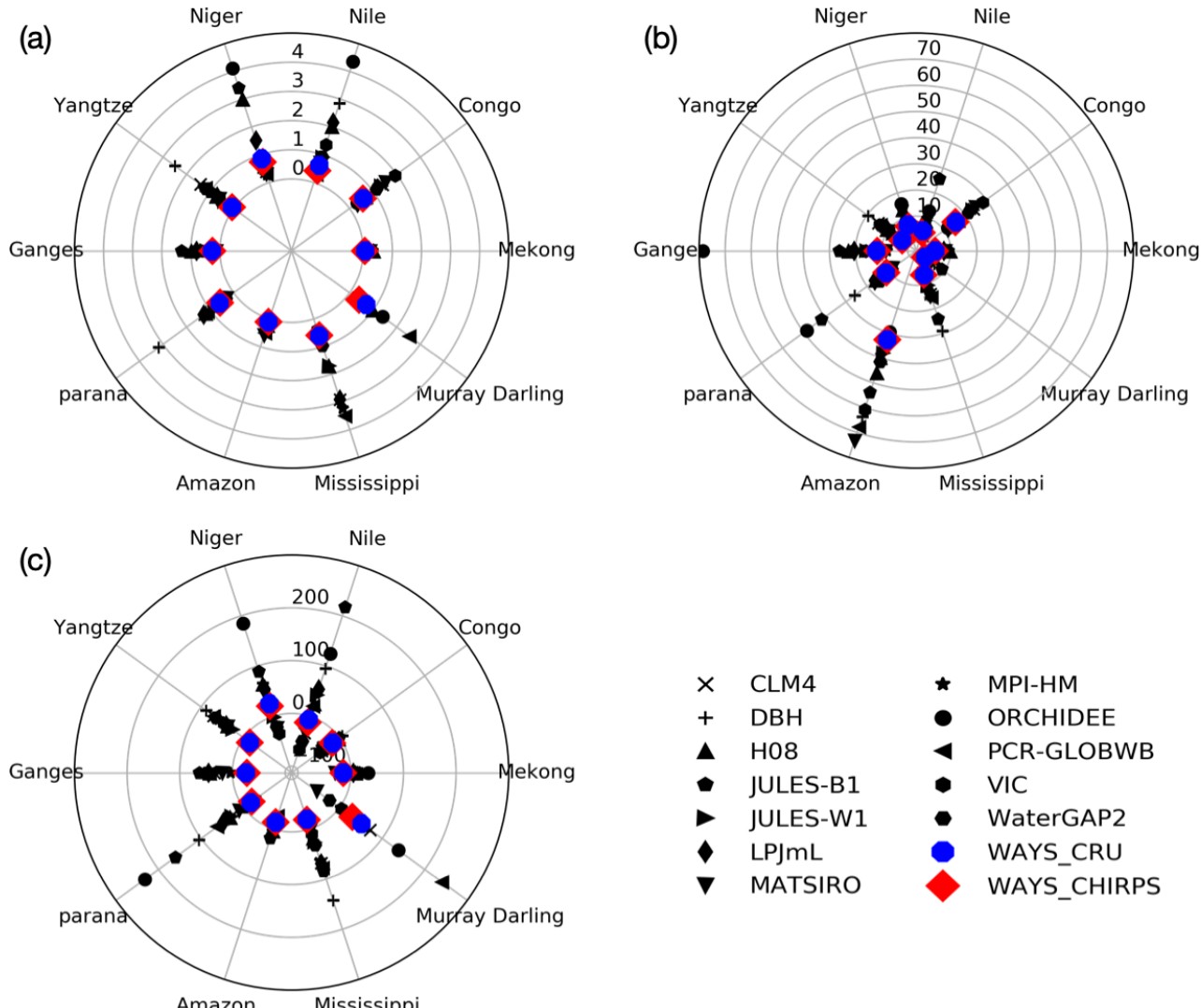

**Figure 4.** The catchment clockwise pole plot according to different metrics, (a) 1-NSE, (b) RMSE, and (c) PBIAS. Colored markers indicate the score for the WAYS model with two different simulations, and black markers represent the score for ISIMIP2a models. For all the metrics, the value of 0 is the benchmark.

model structural uncertainties (Haddeland et al., 2011; Gudmundsson et al., 2012). The lack of physical process representations, e.g., transmission loss, in the hydrological models can also explain some of the biases between the simulated runoff and the reference data (Gosling and Arnell, 2011).

The performance of WAYS is further evaluated against the gauge observations. Since WAYS does not have a native runoff routing module at the moment, a third-part runoff routing tool, CaMa-flood, is applied to route the WAYS simulated runoff (Yamazaki et al., 2011). The evaluation results can be found in the Supplementary Information (SI).

## 4.2 Validation of Root Zone Water Storage

Similar to the runoff evaluation, the performance of the simulation of root zone water storage by the WAYS model is also evaluated at ten major river basins in the period from 2001 to 2010. The spatial pattern as well as the RZWS dynamics at different latitudes and in different months can be found in the SI of the paper. Since the NDII is a normalized index and on an 8-day temporal scale, the WAYS simulated root zone water storage is first averaged over an 8-day temporal scale and then normalized to the range between 0 and 1 before the comparison. A few time steps are missing in the NDII data set. To keep the compared data sets consistent, only pair-wised RZWS data are selected for the model evaluation.

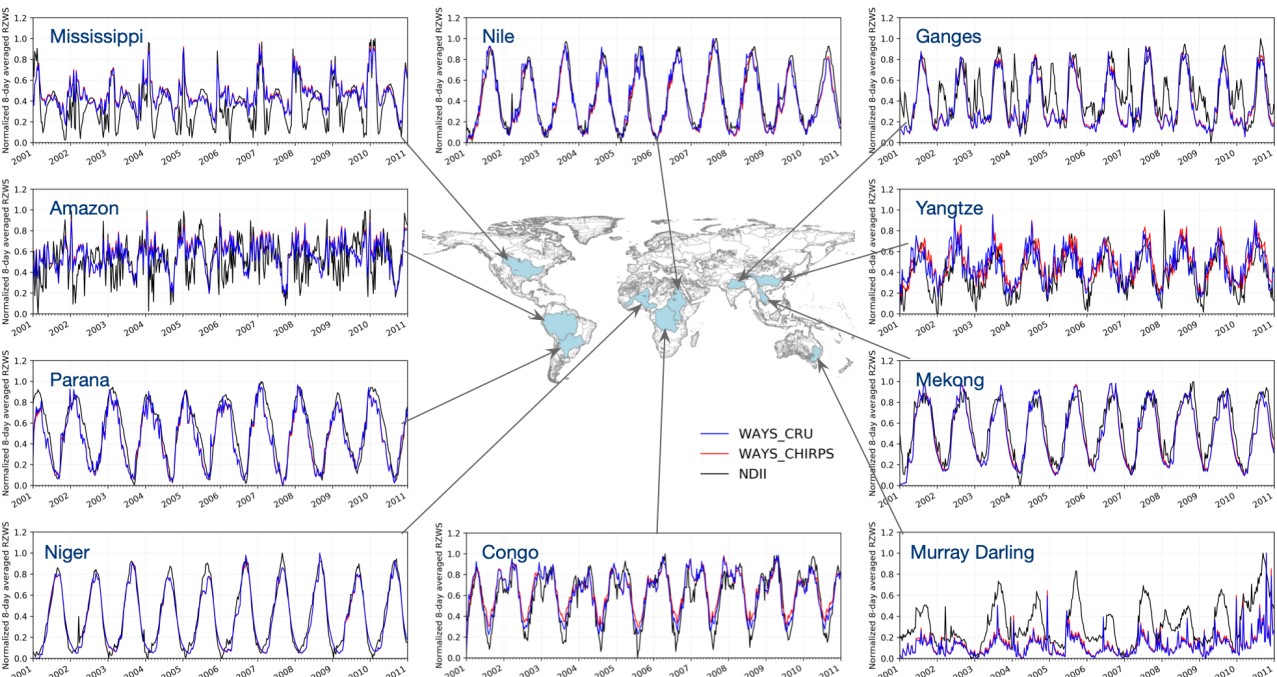

**Figure 5.** Time series of 8-day normalized RZWS simulated by the WAYS model and NDII value.

The 8-day NDII values are compared to the 8-day averaged root zone water storage values of the WAYS model, and the results are shown in Figure 5 and Table 3. Figure 5 shows a comparison of the time series of NDII and simulated RZWS in the selected basins, and Table 3 presents the corresponding rank correlation (Spearman's rho) between NDII and RZWS. The

**Table 3.** The rank correlation of NDII and WAYS-simulated RZWS in ten selected basins

| Selected River Basins | Models | |
| --- | --- | --- |
| | WAYS_CRU | WAYS_CHIRPS |
| Congo | 0.872 | 0.871 |
| Nile | 0.951 | 0.967 |
| Niger | 0.975 | 0.975 |
| Yangtze | 0.713 | 0.764 |
| Ganges | 0.803 | 0.817 |
| Parana | 0.931 | 0.934 |
| Amazon | 0.593 | 0.552 |
| Mississippi | 0.689 | 0.677 |
| Murray Darling | 0.614 | 0.636 |
| Mekong | 0.936 | 0.938 |

RZWS simulated by GEPCI-hydro is not compared to the other model, as the RZWS variable is not available in other GHMs. For ISIMIP2a, some models produced the root zone soil moisture within a fixed depth of the soil profile in the model structure. However, this is still a different variable compared with the root zone water storage.

First, it is clear that NDII shows totally different patterns in different basins. Clear seasonal cycles are shown in the Nile, Mekong, and Niger river basins and so on. Camel-like structures are observed in the Ganges and Congo basins, and relatively complex patterns are represented in the Mississippi, Murray Darling and Amazon basins. The simulated RZWS shows good agreement in the time series with NDII in most of the selected basins. High values of rank correlation are also detected in these regions. Seven catchments of ten have a rank correlation value higher than 0.7, especially in the Nile, Niger, Parana and Mekong, where the correlation coefficients are even higher than 0.9, indicating the strong model performance of WAYS in these basins for root zone water storage simulation, as the NDII reflects the soil water content in the root zone (Sriwongsitanon et al., 2016). The two simulated RZWS time series with different root zone storage capacity products also show identical behavior with subtle differences, except in the Yangtze River basin due to the relatively larger differences in the averaged RZSC of the two products ($S_{R,CRU-SM}$: 135 mm, $S_{R,CHIRPS-CSM}$: 163 mm) in this basin. In the Ganges and Congo, the NDII time series show a two-humped structure, which the WAYS model can still capture, even though underestimations are detected in some years. The rank correlation coefficients in these two catchments are higher than 0.8. In the Yangtze, a suddenly high value of NDII is found on the 25th of August in 2008. By investigating the NDII values a few days before and after this date and the precipitation amount in this period, the unrealistic high value might be caused by the quality of satellite data MOD09A1 on that day, as it can be affected by many issues, including clouds, shadow, viewing angle, aerosol loading and so on (Vermote, 2015).

Relatively large differences between NDII and simulated RZWS are also found in some catchments. In the Mississippi, WAYS shows a good performance in large-value simulations, while it struggles to simulate low values, with considerable

overestimation of them. Therefore, the rank correlation is also relatively low in this catchment, with values of approximately 0.67. The Mississippi river basin is the northernmost catchment of our selected basins. The NDII here shows a totally different pattern compared to the others, while the WAYS-simulated RZWS can barely show a clear seasonal variation. There could be multiple reasons for this overestimation: our model has a relatively simple snowmelt module (degree-day method), which could consequently introduce biases into the simulation, especially in relatively cold regions. Additionally, the relatively uncertain forcing data could contribute to the mismatches between NDII and RZWS, as the largest uncertainties in precipitation occur mainly at the higher latitudes (Vinukollu et al., 2011). Some studies also reported that precipitation-induced spurious seasonal and interannual variations also exist in the soil moisture in this basin (Yang et al., 2015). In contrast, WAYS shows a trend of underestimation in the Murray Darling. A possible reason could be that deep rooted plants are widespread across the Murray Darling basin and can tap into groundwater (Runyan and D'Odorico, 2010; Lamontagne et al., 2014); thus, the NDII may not be the correct proxy for moisture stress in this region. A vast amount of groundwater drawing from the saturated zone to the root zone could explain such underestimation of RZWS (Leblanc et al., 2011). Other reasons behind these findings could be the underestimated RZSC in this region as well as the intensive human activities, including dam construction, a water diversion system and river management, which will impact both the RZSC estimation and RZWS simulation (Reid et al., 2002; Kingsford, 2000). In the Amazon, the model can only capture a few downward troughs but shows difficulty in representing the complete complex dynamics of NDII, resulting in the lowest value of the rank correlation (0.593 and 0.552) among all the selected basins. The primary reason for this low performance could be the inability of NDII to represent RZWS in relatively wet regions where water stress for plants is low (Sriwongsitanon et al., 2016). Among our selected basins, the Amazon has the highest averaged annual precipitation amount, with a value of 2201 mm/year in the validation period. In this case, the performance of WAYS in the RZWS simulation of such regions cannot be justified.

Overall, these model validation results over the ten selected river basins deliver generally good evaluated values that suggest the capability of the WAYS model for RZWS simulation, especially for interannual variability simulation. However, attention should also be paid to some regions, e.g., the basins at high latitudes in the Northern Hemisphere as well as the regions with plenty of precipitation where moisture stress might be low and NDII may not correctly reflect the RZWS dynamics (Sriwongsitanon et al., 2016).

### 4.3 Evaporation Evaluation

RZWS has a close link to the total evaporation, as RZWS represents the available water that plants can use. In this section, the performance of WAYS in evaporation simulation is evaluated against the FLUXNET2015 data. FLUXNET2015 is a global network of micrometeorological flux measurement sites that measure the exchange of $CO_2$, water vapor and energy between the biosphere and the atmosphere (Pastorello et al., 2017). The tower-measured latent heat flux (LF, $W/m^2$) is converted to ET ($mm/day$) using the proportionality parameter between energy and depth units of ET (Velpuri et al., 2013) as follows:

$$ET = \frac{LE}{\lambda} \tag{19}$$

where $\lambda$ is the latent heat of vaporization (2.45 $MJ/kg$). In total, 108 stations are selected based on the data availability in the period 1971-2010. The flux tower latent heat is converted to evaporation before the comparison. The correlation coefficients between simulated evaporation and the FLUXNET2015-derived evaporation are then calculated on the monthly scale.

The results are shown in Figure S15. The background is the annual averaged evaporation from WAYS for the period 1971-2010. The points indicate the comparison results between the flux tower and WAYS simulation. The locations of the points indicate the locations of the flux towers, and the colors indicate the correlation coefficient. WAYS is found to have relatively better performance in America, Europe and China than in Africa and Australia. However, a few stations near the boundary of America and Europe also show weak correlations between the simulations and flux tower data.

Figure S16 shows the percentage of data points within different intervals of the correlation coefficient. The calculated correlation coefficient is crowded in the interval of 0.6-0.8, while more than half of the stations (56%) show a correlation coefficient of more than 0.6. The relatively poor performance of the model in some regions could be partially explained by the following reason. FLUXNET2015 corresponds to point-based observation data, while WAYS simulates the evaporation on grid cells with a 0.5 degree spatial resolution. For the comparison, the model simulation in a certain pixel is selected based on the distance between the flux tower and the center of the pixel. The model simulation actually represents an averaged value for a 0.5 x 0.5-degree pixel. This averaging will inherently introduce errors when comparing the simulation to station-based data. Similar results are also found in other studies comparing FLUXNET2015 data to either model simulations or remote sensing-derived evaporations (Lorenz et al., 2014; Velpuri et al., 2013).

Furthermore, the average monthly evaporation is compared to the FLUXNET2015 data at each flux tower, and the results are shown in Figure 6. Good correspondence between the model simulation and flux tower data can be found by visual inspection. The points with a higher correlation coefficient show a better relationship between the model simulation and flux tower observation and are distributed closer to the diagonal. The evaluation results confirm the generally good performance of WAYS in monthly evaporation simulation. The detailed results on evaporation evaluation against FLUXNET2015 are provided in the SI as Excel files. In addition, an evaluation of the evaporation simulation is further conducted against LandFluxEVAL, a merged benchmark synthesis product of evaporation at the global scale (Mueller et al., 2013). The results can be found in the SI.

## 4.4 The Effect of Root Zone Storage Capacity on Hydrological Simulation

RZSC is a key parameter of the WAYS model. Therefore, it is important to investigate how RZSC could affect the model simulation. In addition to the model simulated with satellite data-derived RZSC products ($S_{R,CHIRPS-CSM}$ and $S_{R,CRU-SM}$), we have additionally conducted WAYS simulations with RZSC derived from uncertain root depth and soil data. The uncertain RZSC ($S_{R,LOOKUP-TABLE}$) is derived based on literature values of root depth and soil texture data (Müller Schmied et al., 2014; Wang-Erlandsson et al., 2016). Due to the global coverage of the RZSC data ($S_{R,CRU-SM}$), only the simulation with $S_{R,CRU-SM}$ is used for comparison. The spatial distribution of the uncertain RZSC is shown in Figure S17, and the differences between $S_{R,CRU-SM}$ and $S_{R,LOOKUP-TABLE}$ are shown in Figure S18. It can be seen that there are large differences between the two RZSC products. The simulation with uncertain RZSC $S_{R,LOOKUP-TABLE}$ shows overestimation globally

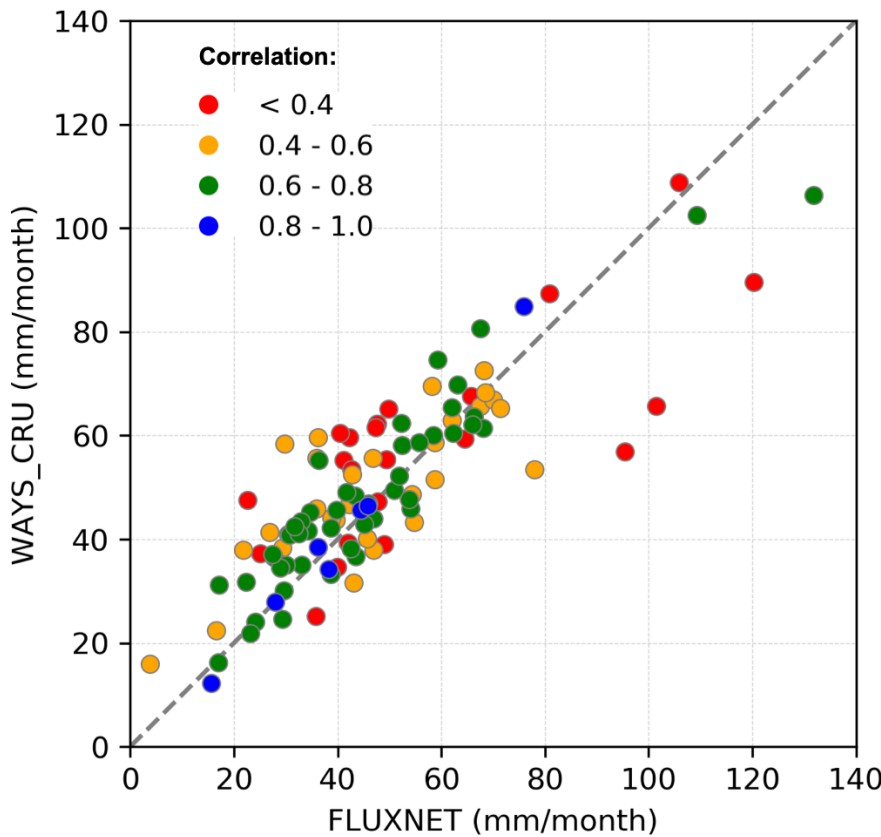

**Figure 6.** Averaged monthly evaporation of WAYS simulation (WAYS_CRU) against the FLUXNET data.

except for some regions around low-middle latitudes. The latitudinal averaged RZSC further confirms the overestimation of $S_{R,LOOKUP-TABLE}$ at middle-high latitudes (Figure S19).

The large differences between these two RZSC data sets also introduce differences in simulated hydrological elements. Figure S20 shows the impacts of RZSC on the model simulation, including runoff, evaporation and RZWS. A blue color

5  (decrease of RMSE and increase of the ranked correlation) indicates an improvement of the simulated results by replacing the uncertain RZSC ($S_{R,LOOKUP-TABLE}$) with satellite data-derived RZSC ($S_{R,CRU-SM}$), while a red color implies the opposite. For comparison, reference data are used for different variables. For runoff, evaporation and RZWS, the reference data are ERA-Interim/Land (2001-2010, monthly), LandFluxEVAL (1989-2005, monthly) and NDII (2001-2010, 8-days), respectively. Generally, the model simulations are improved by using the RZSC $S_{R,CRU-SM}$. This result emphasizes the

10  importance of an appropriate representation of RZSC in WAYS. A decline of the model performance is also found in some regions at high latitudes and low latitudes. This result can be partially explained by the inherent uncertainty in the $S_{R,CRU-SM}$ data, as they are derived from other data sets. The RZSC derivation method itself as well as the input data can also introduce biases (Wang-Erlandsson et al., 2016).

Figure 7 shows the RMSE improvements of simulated monthly evaporation for different land covers obtained by implementing the satellite data-derived RZSC ($S_{R,CRU-SM}$) instead of the uncertain RZSC ($S_{R,LOOKUP-TABLE}$). The analysis reveals that the satellite data-derived RZSC ($S_{R,CRU-SM}$) has great potential to improve the evaporation simulation for all kinds of land covers. The largest improvements are found in broadleaf forests. The improvements in the needleleaf forest, mixed forest and savanna are relatively low. The findings also resonate with another work that used a simple terrestrial evaporation to atmosphere model (STEAM) for evaporation simulation (Wang-Erlandsson et al., 2016).

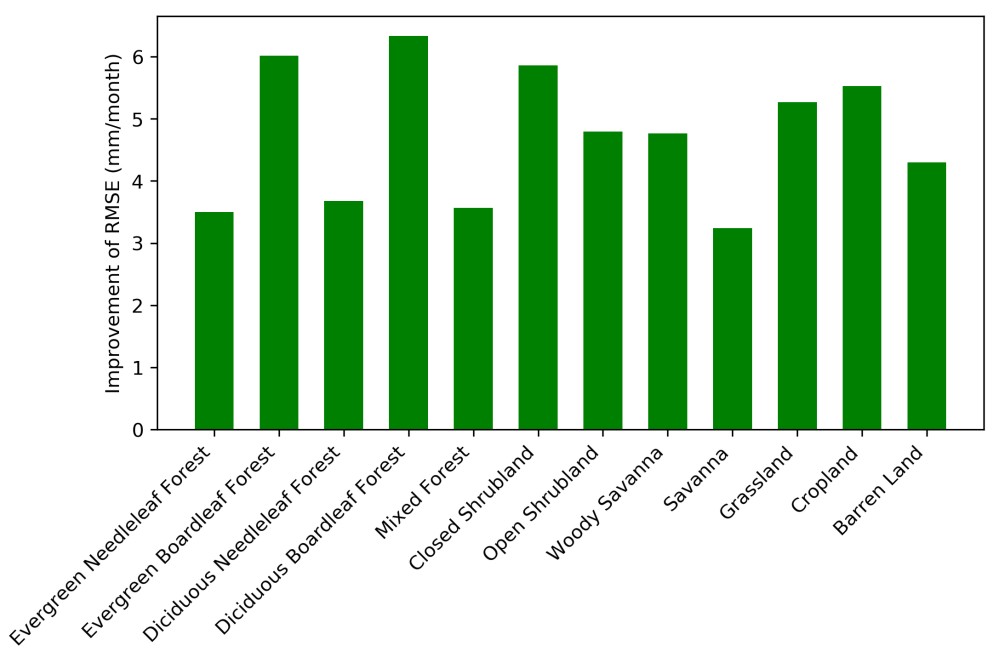

**Figure 7.** The improvement of RMSE in evaporation simulations for different land covers by using the satellite data-derived RZSC ($S_{R,CRU-SM}$) instead of the uncertain RZSC ($S_{R,LOOKUP-TABLE}$).

## 5   Discussion and conclusion

In this study, a global hydrological model has been developed that aims to simulate the soil water volume stored in the entire root zone, a critical variable for ecohydrology-related studies, by considering the global spatial heterogeneity of the plant rooting system. The primary motivation behind the development of WAYS is to improve the integrality of soil water simulation in hydrological models by acknowledging the key role played by RZWS in many applications, as it connects the climate, hydrology and earth surface systems (Savenije and Hrachowitz, 2017). Existing models represent the soil profile with different schemes (Devia et al., 2015). However, they still suffer from the structure limitations of the models in reflecting the soil water

dynamics for the entire rooting system (Bierkens, 2015; Sood and Smakhtin, 2015). A persistent weakness in the RZWS simulation in the hydrological models is the lack of direct observations for model evaluation (Sriwongsitanon et al., 2016).

Benefiting from recent progress made in the field of hydrology and remote sensing, the WAYS model is developed based on an advanced lumped model, FLEX (Fenicia et al., 2011; Gao et al., 2014a), and evaluated with a proxy of RZWS, the remote
sensing-based index NDII (Hardisky et al., 1983). NDII is not new, but strong linkage between NDII and RZWS found by Sriwongsitanon et al. (2016) enlightened our work. This potential candidate as a proxy of RZWS bridges the gaps in the field, where RZWS cannot be directly observed at large scales. The model FLEX is widely used and has been validated for root zone water dynamics simulation, but at the basin scale (Gao et al., 2014b; Nijzink et al., 2016; de Boer-Euser et al., 2016; Sriwongsitanon et al., 2016). A variety of modifications and extensions are made based on FLEX that allow WAYS to simulate
the hydrological cycles at the global scale with an advanced schema in the root zone system. Another key parameter that allows appropriate RZWS simulation in WAYS is the global RZSC recently produced by Wang-Erlandsson et al. (2016). Before that, it was usually obtained by look-up approaches with inherently large uncertainty. RZSC reveals the spatial heterogeneity of the plant rooting system and has a direct relation to RZWS. Moreover, RZSC is produced under the assumption that plants do not invest more in their roots than necessary to bridge a dry period. Thus, this assumption is also held by our work, and the root
zone reservoir (Section 2.3) actually defines the part of the unsaturated zone that determines the dynamics of the runoff regime (Sriwongsitanon et al., 2016; Savenije and Hrachowitz, 2017).

The major goal of this study is to test the feasibility of WAYS for reliable RZWS simulation. The newly developed model is first validated for runoff and RZWS simulation in ten major basins across the world and is then further evaluated against station observations, including flux tower and gauge data. Despite regional differences, general good performances are found
for runoff and evaporation simulation. In addition, the WAYS model also shows a good representation of RZWS, with high values of rank correlation in most of the validated regions. The evaluation results confirm the capacity of WAYS as a useful tool to simulate hydrological elements, particularly RZWS, at the global scale. However, we have to highlight that the model shows lower performance in some regions, e.g., the Amazon, in the RZWS simulation, where the reference data NDII may have shortcomings in reflecting RZWS. In these regions where NDII might not be a correct proxy for RZWS, an additional
data set could be helpful for evaluation, e.g., the solar-induced fluorescence (SIF), which reflects photosynthesis and thus has a close relationship to the available water in the root zone. A combination of vegetation index data, such as EVI and NDVI, could also be alternatives, as they represent different characteristics of plants. However, further investigations need to be performed before this combination can be applied. It is also important to note that the high latitude regions are not covered by one of the key parameters, i.e., root water storage capacity, used by the WAYS model, and only major river basins at middle and low
latitudes are investigated. Thus, the performance of the model in the other regions is not justified. This is one of the limitations of this work, and further investigations are needed.

It should also be noted that during the evaluation of RZWS, the reference data NDII represent a normalized index based on surface reflectance and can reflect only the dynamics of RZWS rather than the absolute value (Sriwongsitanon et al., 2016). Therefore, a real value-based evaluation could be much more helpful for the model application. This could be another limitation
of the work. However, this fact also emphasizes the importance and necessity of this work from the following two aspects: 1)

The remote sensing-based approach, e.g., NDII, is thus far one of the best available methods for root zone information retrieval (Tobin et al., 2017). However, it is still limited in its ability to reflect the real value, which urges model development, as the model has the ability for absolute value simulation. 2) The remote sensing-based approach works only for historical analysis, which limits its ability to be used in future impact studies. This issue also motivates model development, as the model can work for both past and future studies after appropriate evaluation.

Moreover, the current study does not consider the groundwater access and irrigation mainly due to the lack of global information. The groundwater table information is crucial for capillary rise simulation (Vergnes et al., 2014). Capillary rise simulation without proper water table information could significantly overestimate the evaporation. Thus, the capillary rise flux is ignored in this study. A similar strategy has also been applied by other works due to the absence of the information on the global water table (Döll et al., 2003; De Graaf et al., 2015; Hanasaki et al., 2018). Observations of irrigation on the global scale are also not available (Leng et al., 2015). Although there are simulated irrigation data available on the global scale, the inherent uncertainties could be propagated in our model simulation. Therefore, irrigation is also not considered at this time. However, this neglect could potentially introduce biases into the model simulation in irrigated areas and deep rooted plant-distributed regions, as both irrigation and capillary rise are an additional supply of soil water recharge. The biases may cause an underestimation of evaporation, especially in the dry summertime (Vergnes et al., 2014). This underestimation could consequently affect the simulation of RZWS and runoff because of the interlinkage of these three elements (Rockström et al., 1999). It is found that ignoring the capillary rise could reduce soil water content in the root zone (RZWS), while the runoff will also be reduced (Vergnes et al., 2014). However, these shortcomings can be simply overcome once the global data are available.

In summary, the newly developed global hydrological model WAYS improves the integrality of soil water simulation in hydrological models, as it simulates the water stored in the entire root zone. This added-value feature could benefit many applications related to the root zone processes. For instance, the correct representation of RZWS could help researchers in the investigation of land-vegetation-climate-water integration, where RZWS plays a key role. The capability for RZWS simulation could also benefit the field of agriculture, as RZWS represents the plant available water, which is closely linked to the crop yields. Moreover, this can also advance the hydrological model itself, as the water stored in the root zone controls the partitioning of the precipitation into evaporation, infiltration and runoff in the model (Liang et al., 1994). The precise simulation of variables in the root zone could benefit the simulation of other elements in the model, thus advancing the model simulation toward an advanced philosophy, i.e., obtaining the right answers for the right reasons rather than simply obtaining the right answers (Kirchner, 2006). In addition, the WAYS model can be further improved by integrating a more sophisticated evaporation module, e.g., the STEAM model developed by Wang-Erlandsson et al. (2014), which separates the evaporation fluxes in a more detailed way. Finally, a runoff generation module recently developed by Gao et al. (2019), HSC-MCT, could provide another possibility to improve the WAYS model, as it offers another venue for determining one of the key parameters in WAYS ($\beta$) independently without calibration. This calibration-free module could actually benefit any conceptual hydrological model.

*Code and data availability.* The model code is provided through a GitHub repository: https://github.com/argansos/WAYS. The meteorological data used in this work are available at the data center of the "Global Soil Wetness Project 3" (http://hydro.iis.u-tokyo.ac.jp/GSWP3/). The land use data are available at the Global Land Cover Facility (http://www.landcover.org). The root zone storage capacity is collected from the work of Wang-Erlandsson et al. (2016). The runoff data for model calibration are available at the Oak Ridge National Laboratory

5   Distributed Active Archive Center (ORNL DAAC) (https://daac.ornl.gov/cgi-bin/dsviewer.pl?ds_id=994). The runoff data for model evaluation are available at the European Centre for Medium-Range Weather Forecasts (ECMWF) website (http://apps.ecmwf.int/datasets/). The NDII data and simulated hydrological data are available upon request from the corresponding author.

*Author contributions.* GM and JL contributed equally to the paper. GM and JL designed the study, analyzed the data and wrote the paper. JL designed the model structure, and GM wrote the model code.

10   *Competing interests.* The authors declare that they have no conflict of interest.

*Acknowledgements.* This study was supported by the National Natural Science Foundation of China (Grant No. 41625001), the Strategic Priority Research Program of the Chinese Academy of Sciences (Grant No. XDA20060402), the National Natural Science Foundation of China (Grant No. 41571022), the Guangdong Provincial Key Laboratory of Soil and Groundwater Pollution Control (Grant No. 2017B030301012). Additional support was provided by State Environmental Protection Key Laboratory of Integrated Surface Water-Groundwater Pollution

15   Control. The model simulation work is supported by Center for Computational Science and Engineering of Southern University of Science and Technology. We would like to acknowledge the authors of the FLEX model for their great help during the development of WAYS.

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
