# Peer review of "WAYS v1: A hydrological model for root zone water storage simulation on a global scale"

_Geoscientific Model Development, 2019_

## Referee Comment (RC1) · Hubert H.G. Savenije (Referee) · 4 Apr 2019

Review of GMD-2019-52 by Mao and Liu.

First of all, I would like to mention that I find this an important paper. In my view, the authors have convincingly shown that their model is a very valuable addition to the set of existing global hydrological models. Its innovation is that it uses the root zone storage capacity (RZSC) determined independently by remote-sensing-based global products for precipitation and evaporation and does not calibrate obtain it by calibration. As far as I know all other global models either calibrate the RZSC or determine it on the basis of incomplete soil maps and inaccurate maps for rooting depth. The authors using climate-derived RZSC has the intuitive advantage that ecosystems apparently adjust

their RZSC to climate variability by creating a buffer against dry periods. In hydrological models the RZSC is the key variable determining the partitioning of precipitation into transpiration, recharge and surface runoff, making it the most important hydrological parameter for land-atmosphere interaction and runoff generation. The fact that the authors demonstrated that a remote-sensing-based estimate of the RZSC can be efficiently used in a global hydrological model is nothing less than a breakthrough.

On top of this, the authors demonstrated in their validation that the NDII, a simple remote sensing based proxy for root zone moisture stress, is a powerful tool to validate (and possibly calibrate) global hydrological models. Highly sophisticated satellites claim to monitor soil moisture, with limited success (e.g. SMOS, ERS, and AMSR-E). But NDII, a readily available remote sensing product observing the moisture content of vegetation, apparently can do this better, because it connects to the root zone moisture tension and not the moisture content of the surface. (Sriwongsitanong et al., 2016).

Of course this paper is a modelling paper, and should be treated as such. In that respect I think that the authors should make the code of the model freely available and not merely on request. The model builds on earlier work by Gao et al. (2014a and 2014b), and by Wang-Erlandsson et al. (2014 and 2016), and I think it is fair that the software is freely made available so that other people can advance this approach further. In fact, I think that a more sophisticated evaporation module as in Wang-Erlandsson et al. (2014) could improve the model even further.

Having said this, the paper requires some (major) revision. I shall highlight the major points.

1. The comparison in Figures 4 and 5 is not entirely fair. The models of the ISIMIP2a data set are not calibrated, whereas WAYS is. This is mentioned in the paper, but the comparison in these figures suggests otherwise. The caption should mention this.

2. It is important that the authors indicate which parameters are input independently (from what I can see: $S_{(rz,max)}$, $K_s$, $f_s$, $R_{(s,max)}$, $S_{(I,max)}$) and which are calibrated (I guess: Beta, K_ff, …). The fact that a number of these have been input as independently obtained parameters is crucial information, but we should also know which have been obtained by calibration. It is well known that there is equifinality between Beta and the RZSC, so this is not trivial. I would also want to see a Table with the calibrated values. There should be an openly shared data set with all parameters used, whether obtained by calibration or independently.

3. In my view, the Beta parameter is crucial. It affects the partitioning of precipitation into transpiration and runoff. The time scales K_s, K_ff and K_f merely affect hydrograph shape, but not the water balance. In this regard, it is interesting to know that Gao et al. (2019) developed a HAND-based method to determine Beta from independent topographical information. This method assumes that the dominant mechanism is Saturation Excess Overland Flow and therefore is not applicable on hillslopes. So it should be used with good judgement, but it offers another venue of estimating Beta independently without calibration.

4. I found a mistake in Equation (2). The correct equation should read: P_tf = MAX{0, P_r – (S_imax – S_i)/Delta t }. The Delta t is required to make the equation dimensionally correct and to prevent that if the model is used at another time step, no error is made. The MAX{0,x} operator is essential since P_tf is an overflow. Forgetting the MAX{0,x} operator can lead to negative P_tf values for small amounts of rainfall. This may trigger relatively small errors, but particularly in wet environments (the Amazon?) this can create errors. I fear that the authors have to rerun the models to correct this mistake.

5. In the validation against NDII, one should realise that some ecosystems (particularly Australian) tap into groundwater, so that in those ecosystems the NDII may not be the correct proxy for moisture stress in the root zone during dry periods. This may be another reason why the Murray Darling performs less well in the comparison with NDII.

6. This brings me to another point, that the FLEX model used apparently does not include capillary rise. In wetlands, this is a dominant mechanism, and also some dryland vegetation is known to tap water from deeper layers. A landscape-based model as developed by Gao et al. (2014a) could cater for this and could also distinguish between an independently derived Beta function for the wetland-terrace-plateau continuum and a calibrated Beta for hillslopes.

7. I don't understand the last sentence in the abstract. Indeed CHIRPS-CSM is limited to lower latitudes, but CRU-SM covers the entire globe. I think the sentence "Therefore, the performance etc." can be deleted.

8. There are many typos. I think the paper requires copyediting, which probably Copernicus can take care of.

So in summary, I think this is an important paper, but additional work needs to be done before the paper can be published.

The references used in this comment also occur in the discussion paper, except the following:

Gao, H., Birkel, C., Hrachowitz, M., Tetzlaff, D., Soulsby, C., and Savenije, H. H. G., 2019. A simple topography-driven and calibration-free runoff generation module, Hydrol. Earth Syst. Sci., 23, 787-809, https://doi.org/10.5194/hess-23-787-2019.

---

## Referee Comment (RC2) · Anonymous Referee #2 · 12 Apr 2019

A correct representation of root zone water storage is important for a robust hydrological modeling. However, in reality, obtaining reliable soil water information is difficult. In many previous studies, the root zone water storage has been quantified as the soil moisture in a certain depth rather than the water stored in the entire rooting system. This leads to an under- or over- estimation of root zone water storage depending on individual site conditions, including the types of vegetation covers on the land surface. The aim of the paper by Mao and Liu is to develop a hydrological model, WAYS, that is capable for simulating root zone water storage on a global scale, without constraining the quantification to a certain depth. Overall, I think the development of such a model is valuable to the hydrological community and can largely advance the eco-hydrological studies which tackle the interactions between the hydrological cycle

and vegetation dynamics on the land surface. I personally also think with the further development and improvement, WAYS has the potential to be applied in the investigation of land-vegetation-climate-water integrations which is very important for the global change impact assessments. Below I give some comments on the paper and hope the authors can address them in the revision. General comments 1. The model structure of WAYS is the core of this paper (Figure 1). Its scientific clarity is essential for others to understand the processes and also is important for possible future wider applications of the model beyond the authors' group. In the current version, the variables in the flow chat depicted in the small window and the ones in the schematic are not all matched. E.g., in the small window, Si, Pe, Rr, Sf, and Ss are used, but they are not indicated in the schematic. Even if their meanings are clear (some are not clear to me), the authors still need to denote them properly in the schematic. In addition, given the central role of Figure 1 in the entire paper, I suggest the authors to add some text elaborating the flow of the figure. This is different from the following sections describing individual processes in the model. 2. Table 1 gives all the equations concerning the water balance in WAYS. This is very useful for examining the processes and evaluating the robustness of the model. As all the equations are from the relevant literature (the authors give the references in the text), it would be good to provide the major references in the last column of Table 1. 3. Are there any other values used for Rx,max rather than 7, 4.5 and 2.5 for sandy soil, loamy soil and clayey soil? May be worth a checking for uncertainties stemmed from the use of Rx,max values for the mentioned soils. 4. In the captions of Figures 4 and 5, the ERA-Interim/Land represents the reference data. I think it is better to directly use ERA-Interim/Land here, because in the Figures, the ERA is used and no reference data is indicated. Also in Figure 4, WAYS-CHIRPS is not visible. Need to give a note for it, e.g., covered by …. The scale for Y axis for Murray Darling should be enlarged to show the simulated runoff more clearly. 5. The authors demonstrated the good performance of WAYS compared to ISIMIP2a models. However, no direct reasons are given to explain the better performances. I assume that the authors want to say that this is because of the better representation of the root zoon

water storage in WAYS. The authors should make this point clear. It justifies the effort for developing WAYS in this paper. Also, it seems to me not very convincing to state that the better performance is really from the better representation of the root zoon water storage. Could some other processes in the WAYS model be also influential for the better performance compared with the results from the models in ISIMIP2a? 6. In Figure 4, the authors stated that in the Murray Darling basin, WAYS performed very well in comparison to the ERA data for runoff. In Figure 7, the difference between simulated root zone water storage and the NDII values is quite large. The similar situation is also seen in Mississippi, Amazon and Yangtze. The correlation values provided in Table 2 are rather low for these river basins. The authors stated that this could be caused by either the uncertainty of WAYS or the problem of using NDII as a proxy of root zone water storage in the specific river basin. In general, I think this is reasonable. However, I still feel that some specific reasons should be highlighted with convincing evidence, instead of just saying this is either due to the problem of WAYS or the use of NDII. Besides, in the discussion, it would be good if the authors can give some suggestions on validation of root zone water storage simulations when the validity of using NDII for validation is not so suitable as shown in the above mentioned river basins. 7. Page 21, last paragraph. It is stated that 'this added value feature could benefit for many applications related to the root zone processes.' The authors should specify some of the potential benefits here. 8. The aim of the paper is to develop WAYS which is capable of simulating root zone water storage. In the model evaluation section, much text is about the validation of runoff. The elaboration of the importance to correctly represent root zone water storage and the good performance of WAYS in realizing this goal is relatively brief. It would be good if the authors can strengthen this part of the text to highlight the accomplishment of the paper. 9. I like the philosophy stated in the end of the paper, 'get the right answers for the right reasons rather than simply to get the right answers'. In this paper, I feel that the right results are clearly shown. But the right reasons, to me, are relatively weak. The good performance in runoff and root zone water storage simulations could be good results, but reasons for the good results

needs to be more clearly and explicitly explained and supported by evidence. Specific comments 10. The window in Figure 1 should be enlarged, as it is important to show components and their connections clearly. Anyway, there is space in Figure to accommodate the enlargement. 11. The manuscript contains many typos and grammatical mistakes. A professional editing of the manuscript is necessary, particularly because I think the paper has the potential to be an important paper in the field and could receive a high citation in the coming years.

---

## Short Comment (SC1) · 16 Apr 2019

This study tries to develop a 'global' hydrological model. The authors are lack of good understanding on hydrological processes, and the methodology they used are not appropriate at all. The authors overclaimed their contribution. The manuscript is poorly written. Some of the figures are not clear. The current manuscript cannot be accepted, and should be returned to the authors to make it a better work.

My comments are as below: 1. The methodology used are not appropriate at all. The authors compared their runoff simulation against some global model simulation and composite runoff data. We all know the global runoff simulation/composite runoff data are designed for global studies, and can have very large uncertainty on each river

basin. They cannot use these data to verify their simulation, especially for a study aiming to develop a 'new model'. Thus, the comparison between the authors' simulation and other runoff data that the authors used means nothing: the authors cannot claim their model is good. The authors should compare their simulation against hydrological gauge observation which is not difficult to collect at all. I doubt the authors' results. They may choose to avoid the comparison against hydrological gauge observation purposely because their model is suffering fatal flaws. For a paper developing a hydrological model, comparison against in situ gauge observation is extremely important. The authors should not skip this step. In addition, the authors compare their runoff simulation after calibrating their model, whereas the other models in ISIMIP are not calibrated. Thus, the comparisons are useless because the models in ISIMIP have large uncertainty which have already been unrevealed in several recent studies by the ISIMIP group (perhaps the authors missed these very important publications).

2. The authors are lack of basic knowledge about remote sensing. The root zone can be more than 10 meters in depth. The sensors used in the NDII studies cannot penetrate the earth ground up to 10 meters, and even one or two meters are suffering large uncertainty because the attenuations of signals with increase in depth. This is why the most state-of-the-art soil moisture products just provide data in the surface 5/10 cm. Thus, the comparison against NDII based data is not appropriate at all. Because this paper is to develop a model, the authors should use in situ observation which is not difficult to collect. I don't understand why the authors choose to skip the comparison against gauge observation.

3. The authors do not have a good understanding on hydrological processes. i). Vegetation plays a vital role in runoff variations especially in densely vegetated regions (e.g., the Amazon, Congon and some regions in the Yangtze, Mekong, Ganges, Mississippi Rivers et.al.) through the transpiration processes. At the leaf and canopy scales, the mechanisms of transpiration are also different. LAI, fPAR, $CO_2$, wind, solar radiation, stomatal conductance all are influencing transpiration. The authors did not consider the

stomatal influence at all (as shown in the Figure 1 and Table 1). Without comprehensively considering the transpiration processes, how the model developed can predict water resources availability, especially many recent studies have unravelled that the earth is greening and CO2 concentration is increasing. Thus, the model developed by the authors has fatal flaws, and this paper cannot be accepted.

ii). The infiltration capacity of soil plays an important role in controlling the volume of surface runoff and subsurface runoff, and also influences root zone water storage. The infiltration capacity of soil is related to soil type, and has clear physical meaning. The authors considered the infiltration as shown in the Figure 1. However, the authors did not report how they determine this important parameter value. If the authors used the values related each soil type, they did not report which soil map distribution data and which hydraulic property datasets of the soil types are used. If the authors calibrated the parameter values, the authors should be aware of that if it is appropriate to calibrate because the results may be wrong after calibrating some parameters with clear physical meaning. The authors are afraid of reporting the calibrated parameter values and the parameter ranges used in the calibration. The authors stated they calibrated their model for good runoff simulation. I am afraid that they calibrated their model for good runoff simulation with the cost of losing the physical meaning of important parameters. Perhaps the authors choose to not show the important information purposely in order to get their paper published. No, absolutely no. The authors have to show which parameters are calibrated, the parameter ranges used in calibration and calibrated parameter values.

4. The model developed is not a global scale model at all. Because the authors did not use soil map and related soil hydraulic parameter values, the use of the model must rely on calibration to determine some of its parameter values on river basin scales. Therefore, it cannot be a global scale model. It is still a river basin scale model, and the authors just applied the model in several large-scale river basins (without any river basins in most of the regions of Canada, Europe, Middle East, Russia, Mongolia). The

used river basins just cover a small proportion of global land surface.

5. The authors claimed they used 2000 iterations to calibrate their model. However, the authors did not explain the reason. Why 2000 iterations were used?

6. The root zone storage variations are related to ground water level dynamics. Did the model simulate the ground water level changes? Please show the simulation results.

7. Please use scientific languages. The sub-titles of Section 2.4 and 2.5 are not appropriate in such as a scientific paper. The statements 'Fast- and Slow-' are vague.

8. I agree with the reviewer 1 about the capillary mechanism which is missed by the authors. This indicates the authors are lack of good understanding on hydrological processes from another perspective. When we develop a new model, we try to incorporate new hydrological mechanism to advance our understanding on hydrological processes. However, the authors missed several very important hydrological processes which have already been recognised to be very important. Therefore, the 'developed' model cannot provide any new understanding on hydrology to us. I am afraid that the authors just copy other models' code, delete several important parts, replace a few equations and change computer language used in original code, and then the authors claim they develop a new model. No, this is not the right way to do research. I also wonder why the authors delete the capillary mechanism part from the original code. The authors should realize that they cannot just delete some codes of other's model, and make it look like a 'new model' in order to get the manuscript published. This is not real science. The authors must work hard to consider the capillary and vegetation transpiration mechanisms and using gauge data to validate their simulation. Otherwise, their model cannot be better (based on the physical processes considered) than other hundreds/thousands of models that already exist.

9. The manuscript is poorly written and needs to be largely reworked. There are many typos and grammar mistakes. Many sentences are vague and lack of support. The figures are not clear, e.g., Figure 4 and Figure 5, and one cannot distinguish the lines.

10. Figure 2 is not your result. Please remove Figure 2. Using related references in the manuscript to refer to the data is ok.

---

## Referee Comment (RC3) · Anonymous Referee #3 · 18 Apr 2019

**General comments**

The manuscript presents an interesting extension of the FLEX model with enhanced capability for root zone storage simulation at the global scale. Root zone storage capacity is an Achilles heel in global hydrological modelling that is crucial for determining water stress, but most often dependent on highly uncertain soil and rooting depth data. Thus, the authors are addressing an important issue of high relevance for the hydrological modelling community. However, among other improvement possibilities, I think that the analyses need to be more systematic and rigorous, and the manuscript need to better communicate the motivations underlying the developers' choices. I think the manuscript merits to be published after a major revision. My main concerns are the following:

- The manuscript could benefit from clearer descriptions of rationale and motivations for the model development, the analyses performed and other choices made. For example, why was runoff selected for evaluation against ERA-Interim/Land and the non-calibrated ISIMIP simulations? Why not use gauged data for a selection of basins and the ERA-Interim/Land and ISIMIP for global gridded comparisons? Were other variables and potentially better datasets considered and rejected for which reasons? How come capillary rise is disregarded on the basis of "lack of information at the global scale" is there are other models that take it into account? Why was Penman-Monteith FAO 56 PM method used (P6L25)? What were the considerations? Etc. Reviewers and readers will always have different views on preferred evaluation datasets and equations, but a clear description of the underlying rationale and motivation could help bridging differences in perspective if choices can be well-justified.

- The analyses could be better designed to facilitate understanding of how and why WAYS perform in certain ways, and thus, give more insight into how various components of the model affect the root zone storage and runoff simulation? For example, can the authors show how results are affected by e.g., use of root zone storage capacity derived from uncertain root depth and soil data versus the root zone storage capacity from Wang-Erlandsson et al., 2016? Can the authors perform some sensitivity analyses to highlight model structure and parameter sensitivity?

- The WAYS model is developed based on essential features of the FLEX model (P3L14), and as such I would (1) suggest the authors to present an overview of the similarities and differences between the two and (2) to retain "FLEX" in the model naming (e.g., FLEX-WAYS). Retaining FLEX in the name benefits the model developers that do not need to explain the model roots and will have an easier time communicating the new model developments that builds on an existing well-established mode, and would also be a nice acknowledgement of the

earlier FLEX model developments. The practice of name roots exists in the modelling community, and e.g., the models LPJmL and LPJ-GUESS show through their names that they share the same roots.

- The WAYS performance evaluation in terms of root zone storage moisture is highly dependent on the comparison with NDII, which weakens the conclusions, since also further work is still needed to robustly establish the relationship between NDII and soil moisture at the global scale. It is after all only recently suggested by Sriwongsitanon et al. (2016) – a study in a river basin in Thailand – that NDVII can have the potential to be used as a proxy for catchment scale root zone storage capacity. The authors could potentially strengthen their conclusions by evaluating model simulation outputs with additional sources of data/methods, such as FLUX-tower, evaporation, EVI etc. Summarizing evaluation figures can be shown in the main manuscript, and others could be included in Supplementary Information. A more detailed list of the equations and calibration process could also be included in Supplementary Information for transparency.

- Wang-Erlandsson et al., 2016 found that normalizing the root zone storage capacity using the Gumbel distribution by land cover type further improves performance, and recommended the use of Gumbel distribution. Please consider applying the Gumbel normalization to the root zone storage capacity data.

- Please consider discussing how and where the results might be influenced by groundwater access and irrigation, noting that the root zone storage capacity in Wang-Erlandsson et al., 2016 was adjusted for irrigation but not access to groundwater, while WAYS do not account for either groundwater or irrigation.

- Please provide the source code, and not only by request.

**Specific comments**

- P1L10: state what was used for evaluating root zone storage (i.e., NDII) in the abstract.

- P1L10: "many applications": please provide concrete examples.

- P1L11: "attention needs to also...": hardly the most important limitation, please consider rather listing the more pressing future model developments needs and emphasize the key contribution of this model in comparison to other existing global hydrological models.

- Please point out that Sriwongsitanon et al. (2016) is a study in a river basin in Thailand and not a global study.

- P6L27 "no information is available at the global scale": Please consider including a few more lines describing the issues related to capillary rise modelling in global scale models and include related references, such as (Vergnes, Decharme, and Habets 2014) and references within.

- P8L28, "it has been well-justified (de Boer-Euser et al., 2019)": please consider specifying what is justified and add other relevant sources, e.g. "the method has been shown to increase model performance at both basin and global scale (e.g., de Boer-Euser et al., 2016, 2019, Gao et al. 2014, Wang-Erlandsson et al., 2016, Nijzink et al., 2016)".

- P14L11, "reported in his work": please change to "reported in their work".

- P22L6 "DNII", should be NDII.

On a rather different note, please excuse me for taking the opportunity to promote constructive and supportive comments and reviews. Upon reading the short comment by William Chris, I felt urged to stand up for the view that reviews and comments can be both critical and constructive at the same time. Non-constructive comments attack

the person and could look like this "the authors lack basic knowledge about...", while equally critical, but constructive comment would simply address the issue "the authors neglect the fact that NDII only ..., and thus, does not provide an adequate... ". Surely, there are manuscripts that sometimes so lack in substance and show utter disregard for the reviewers' time, but this manuscript is not one of them. I would like to believe that we all - the authors, reviewers, and comments writer - all have poured time and efforts in this because we fundamentally share a love for the science and the science community, so please also let our writing reflect the fact that we are all in this together.

**References**

Vergnes, J.-P., B. Decharme, and F. Habets. 2014. "Introduction of Groundwater Capillary Rises Using Subgrid Spatial Variability of Topography into the ISBA Land Surface Model." Journal of Geophysical Research: Atmospheres 119(19): 11,065-11,086.

---

## Short Comment (SC2) · 24 Apr 2019

Dear authors,

in my role as Executive editor of GMD, I would like to bring to your attention our Editorial version 1.1:

http://www.geosci-model-dev.net/8/3487/2015/gmd-8-3487-2015.html

This highlights some requirements of papers published in GMD, which is also available on the GMD website in the 'Manuscript Types' section:

http://www.geoscientific-model-development.net/submission/manuscript_types.html

In particular, please note that for your paper, the following requirements have not been

met in the Discussions paper:

1. "The main paper must give the model name and version number (or other unique identifier) in the title."// Accordingly, add the acronym WAYS including a version number to the title.

2. GMD is encouraging authors to upload the program code of models (including relevant data sets) as a supplement or make the code and data of the exact model version described in the paper accessible through a DOI (digital object identifier). In case your institution does not provide the possibility to make electronic data accessible through a DOI you may consider other providers (eg. zenodo.org of CERN) to create a DOI. Please note that in the code accessibility section you can still point the reader how to obtain the newest version.

   If for some reason the code and/or data cannot be made available in this form (e.g. only via e-mail contact) the "Code Availability" section need to clearly state the reasons for why access is restricted (e.g. licensing reasons). Consequently, you need to provide a reason in the code availability section, why the code can not be made publicly available. Without a proper reason, it is not acceptable that the code is not made public available before the final publication of the article.

Yours,

Astrid Kerkweg (Executive Editor)

---

## Short Comment (SC3) · 30 Apr 2019

1. After reading the manuscript, I do not think the results can support the research objective. First, the WAYS is calibrated and the runoff simulation is compared with others after that. Therefore, the better performance of WAYS (let's assume it is better first, and actually I do not think so) could be due to the calibration, not because of the consideration of root zone water storage changes. Second, the NDII data were used as a surrogate of root zone water storage changes. The NDII is just suggested in a river basin in Thailand. However, it is still not clear if it is appropriate to do so on large scales with different climate and hydrological regimes. Therefore, the simulated root zone water storage is not actually verified.

[Figure]

2. I also feel it is not rigorous that without in situ hydrological gauge and flux tower data to verify simulations of ET, runoff, soil moisture etc. Comparison with ISIMIP model runoff simulation is not convincing. The spatial distribution of simulation is also important. Please show the spatial distribution.

3. As shown in Figure 1 and the manuscript descriptions, the soil layer is separated into vadose zone (includes the root zone) and saturated zone, which is similar to many existing models. In addition, in this manuscript, the NDII is not justified to represent root zone water storage changes on large scales with different climate and hydrological regimes. Therefore, the novelty of this manuscript is not enough.

4. Because the soil is separated into different zones, at every grid, each zone must have a certain depth (or a percentage value) at a moment and the depth or percentage will change with rainfall-runoff processes. The manuscript failed to report the changes of the depth of each zone or what percentage of soil is saturated/unsaturated at different time. Please also show the spatial distribution. This is important to see if the simulation is reasonable.

5. I don't think the WAYS is a new hydrological model because it only changed several equations and replaced a few parameters compared to the FLEX model. It is not an improvement of the FLEX model either, because it removed several important components of the original FLEX model and the manuscript failed to prove that the WAYS is better compared to FLEX after doing that.

6. When I saw the root zone water storage, I thought the manuscript would study vegetation. However, I did not find how they deal with vegetation transpiration. Because root zone water storage changes are largely controlled by vegetation transpiration, I don't believe the WAYS can simulate root zone water storage changes properly without considering vegetation transpiration. I share the similar concerns as other reviewers that WAYS has fatal flaws regarding this. In addition, WAYS means 'Water And ecoSystem Simulator' according to the manuscript. Without considering vegetation transpiration,

[Figure]

WAYS cannot represent ecosystem and cannot simulate ecosystem influence on water either. Thus, I believe that the manuscript title, the statement in the manuscript, and the model name are misleading and not suitable.

7. The manuscript failed to report how many parameters the model has, which parameters need to be calibrated, what are the calibrated parameter values, which parameters use default values. The physical meanings of the parameters should be reported. Some parameters have their physical meanings and cannot be calibrated.

In sum, I am not convinced by the methodology and results, and several key issues of the study objective are not solved. I feel that this manuscript should be rejected.
* * *

---

## Author Comment (AC1) · 26 Jul 2019

We would like to thank Referee Prof. Hubert H.G. Savenije for his interest in this topic and for the valuable comments to improve our manuscript. Based on the comments additional calculations have been performed in the revised manuscript. Our point-by-point response to the comments is given in the following (**Comments in black**, Answers in blue and the corresponding changes in the revised manuscript are marked in orange.):

**First of all, I would like to mention that I find this an important paper. In my view, the authors have convincingly shown that their model is a very valuable addition to the set of existing global hydrological models. Its innovation is that it uses the root zone storage capacity (RZSC) determined independently by remote-sensing-based global products for precipitation and evaporation and does not calibrate obtain it by calibration. As far as I know all other global models either calibrate the RZSC or determine it on the basis of incomplete soil maps and inaccurate maps for rooting depth. The authors using climate-derived RZSC has the intuitive advantage that ecosystems apparently adjust their RZSC to climate variability by creating a buffer against dry periods. In hydrological models the RZSC is the key variable determining the partitioning of precipitation into transpiration, recharge and surface runoff, making it the most important hydrological parameter for land-atmosphere interaction and runoff generation. The fact that the authors demonstrated that a remote-sensing-based estimate of the RZSC can be efficiently used in a global hydrological model is nothing less than a breakthrough.**

We thank the referee for this comment.

**On top of this, the authors demonstrated in their validation that the NDII, a simple remote sensing based proxy for root zone moisture stress, is a powerful tool to validate (and possibly calibrate) global hydrological models. Highly sophisticated satellites claim to monitor soil moisture, with limited success (e.g. SMOS, ERS, and AMSR-E). But NDII, a readily available remote sensing product observing the moisture content of vegetation, apparently can do this better, because it connects to the root zone moisture tension and not the moisture content of the surface. (Sriwongsitanong et al., 2016).**

We thank the referee for this comment.

**Of course this paper is a modelling paper, and should be treated as such. In that respect I think that the authors should make the code of the model freely available and not merely on request. The model builds on earlier work by Gao et al. (2014a and 2014b), and by Wang-Erlandsson et al. (2014 and 2016), and I think it is fair that the software is freely made available so that other people can advance this approach further. In fact, I think that a more sophisticated evaporation module as in Wang-Erlandsson et al. (2014) could improve the model even further.**

Thank you for the suggestion. We will make the code of WAYS model freely available after the manuscript is accepted. Moreover, we completely agree that the evaporation module in the STEAM model (Wang-Erlandsson et al., 2014) could further improve the WAYS model, because the former (Wang-Erlandsson et al., 2014) separates the evaporation fluxes in a

more detailed way. We are also very interested in coupling the STEAM model with WAYS in our future work by cooperating with the authors of the STEAM model.

**Authors' change in the manuscript.**

**Page 26, Line 9: (the changes is marked as blue)**
The precise simulation of variables in the root zone could benefit the simulation of other elements in the model, thus advancing the model simulation toward an advanced philosophy, i.e., obtaining the right answers for the right reasons rather than simply obtaining the right answers (Kirchner, 2006). In addition, the WAYS model can be further improved by integrating a more sophisticated evaporation module, e.g., the STEAM model developed by Wang-Erlandsson et al. (2014), which separates the evaporation fluxes in a more detailed way. Finally, a runoff generation module recently developed by Gao et al. (2019), HSC-MCT, could provide another possibility to improve the WAYS model, as it offers another venue for determining one of the key parameters in WAYS (β) independently without calibration. This calibration-free module could actually benefit any conceptual hydrological model.

**Having said this, the paper requires some (major) revision. I shall highlight the major points.**

We thank the referee for the constructive comments. Below, we give a point-to-point reply to the comments posted by the referee.

**1. The comparison in Figures 4 and 5 is not entirely fair. The models of the ISIMIP2a data set are not calibrated, whereas WAYS is. This is mentioned in the paper, but the comparison in these figures suggests otherwise. The caption should mention this.**

Thank you. Actually, the simulated runoff of the WAYS model is first compared with the reference data ERA-Interim/Land runoff. The performance of the WAYS model in runoff simulation is evaluated mainly based on the comparison between ERA-Interim/Land data and WAYS simulation. Since WAYS uses the same driving data as ISIMIP2a models and the ISIMIP2a simulations are widely studied and discussed in many studies, we believe the additional comparison between WAYS and ISIMIP2a models can provide added-value for examining our model. Therefore, ISIMIP2a simulations are also shown in the results section together with the ERA-Interim/Land data.

We agree with the referee and have revised the captions of Figures 4 and 5 accordingly. We would also like to note that because Referee #2 suggested the use of ERA-Interim/Land data instead of reference data in the captions of Figures 4 and 5, these changes are also shown in the captions. In addition, in short comment #1 (comment 10), the reviewer suggested us to move the Figures 2 and 3 from the main text of the manuscript. Thus, the figures in the revised manuscript are re-sorted.

**Authors' change in the manuscript.**

[Figure]

**Figure 2.** Time series of monthly runoff simulated by WAYS and the ISIMIP2a models, as well as the reference data. The basins highlighted in the world map indicate the selected catchments for model evaluation. The solid lines in blue and red indicate the WAYS simulations with two different RZSC products. The solid line in black indicates the ERA-Interim/Land data, and dashed lines represent the ISIMIP2a model simulations. In some plots, the red line is not visible and is covered by the blue line due to the small differences between the two WAYS runs. WAYS is calibrated using Composite Monthly Runoff data, while the ISIMIP2a models are not calibrated for the simulation.

**Page 15: Figure 3 caption is updated**

[Figure]

**Figure 3.** The probability of exceedance for monthly runoff simulated by different models as well as the reference data in ten selected basins. The solid lines in blue and red indicate WAYS simulations with two different RZSC products. The solid line in black indicates the ERA-Interim/Land data, and dashed lines represent the ISIMIP2a model simulations. In some plots, the red line is not visible and is covered by the blue line due to the small differences between the two WAYS runs. WAYS is calibrated using Composite Monthly Runoff data, while the ISIMIP2a models are not calibrated for the simulation.

**2. It is important that the authors indicate which parameters are input independently (from what I can see: S_(rz,max), K_s, f_s, R_(s,max), S_(I,max)) and which are calibrated (I guess: Beta, K_ff, . . .). The fact that a number of these have been input as independently obtained parameters is crucial information, but we should also know which have been obtained by calibration. It is well known that there is equifinality between Beta and the RZSC, so this is not trivial. I would also want to see a Table with the calibrated values. There should be an openly shared data set with all parameters used, whether obtained by calibration or independently.**

We agree with the referee's comments and have inserted a table to describe the parameters that are used in the WAYS model as well as their ranges. WAYS has 13 parameters in total, seven of which are obtained from the literature and the rest (six parameters) from the calibration (see page 11, table 2 in the revised manuscript). We will share all the model parameters after the manuscript is accepted. Since the calibrated parameters are spatially varied, it is not appropriate to show them in tables. Here we provide the spatial patterns of two key parameters ($\beta$, $C_e$) that are calibrated, as these two

parameters mostly affect the partitioning of precipitation (see Figure S21 and Figure S22). The rest of the calibrated parameters are uploaded to the response thread in terms of netCDF files (parameters.cn4) as Supplements.

[Figure]

*Figure S21. The spatial distribution of the model parameter $\beta$*

[Figure]

Parameter value ($C_e$)

*Figure S22. The spatial distribution of the model parameter $C_e$*

**Authors' change in the manuscript.**

**Page 12: A table is added**

**Table 2.** Parameter ranges of the WAYS model

| Parameter | Range | Literature | Parameter | Range |
|---|---|---|---|---|
| $S_{i,max}$ | distributed | Wang-Erlandsson et al. (2014) | $\beta$ | (0, 2) |
| $S_{rz,max}$ | distributed | Wang-Erlandsson et al. (2016) | $C_e$ | (0.1, 0.9) |
| $R_{s,max}$ | 7/4.5/2/5 (Sand/Loam/Clay) | Döll and Fiedler (2008) | $K_f$ | (1, 40) |
| $K_s$ | 100 | Döll et al. (2003) | $K_{ff}$ | (1, 9) |
| $f_s$ | distributed | Döll and Fiedler (2008) | $S_{ftr}$ | (10, 200) |
| $F_{DD}$ | distributed | Müller Schmied et al. (2014) | $T_{lag}$ | (0, 5) |
| $T_t$ | 0 | Müller Schmied et al. (2014) | | |

**3. In my view, the Beta parameter is crucial. It affects the partitioning of precipitation into transpiration and runoff. The time scales K_s, K_ff and K_f merely affect hydrograph shape, but not the water balance. In this regard, it is interesting to know that Gao et al. (2019) developed a HAND-based method to determine Beta from independent topographical information. This method assumes that the dominant mechanism is Saturation Excess Overland Flow and therefore is not applicable on hillslopes. So it should be used with good judgement, but it offers another venue of estimating Beta independently without calibration.**

We completely agree that parameter $\beta$ is crucial as it controls the precipitation partitioning, thus mostly affecting the water balance. In addition, parameter $C_e$ plays an important role in water balance control as it affects the evaporation and consequently influences the root zone water storage, which determines the precipitation partitioning. Indeed, the rest of the parameters (e.g., $K_s$, $K_{ff}$ and $K_f$) are more important for hydrograph shape adjustment rather than the water balance.

We would like to thank the referee for sharing the recently published paper by Gao et al. (2019). The authors developed a calibration-free module (HSC-MCT) for runoff generation based on the Height Above Nearest Drainage (HAND) data. They found that the runoff coefficient can be ingeniously linked with the HAND-based area fraction, and thus $\beta$ can be determined accordingly. This finding offers another venue for determining $\beta$ independently without calibration. The HSC-MCT module can provide added-value for any conceptual hydrological model. We agree that it would be very interesting to integrate HSC-MCT into WAYS and that it can further improve the WAYS model. However, at the current stage, the main purpose of our work is to demonstrate the value of integrating a remote-sensing-based RZSC into a global hydrological model, especially for RZWS simulation. The integration of HSC-MCT could potentially introduce uncertainties into the model as HSC-MCT is currently only applicable to regions dominated by Saturation Excess Overland Flow. Moreover, HSC-MCT is heavily dependent on HAND data, and there are currently no available HAND data at 0.5 degrees. Self-derived HAND data could further introduce uncertainties as HAND is very sensitive to the drainage threshold and the open water elevation (Nobre et al., 2016). A thorough validation of HAND on a global scale is necessary before it can be applied in subsequent analyses. In these regards, we would like to skip the integration of the HSC-MCT module into WAYS at the current stage, but we will examine its inclusion in our future works.

**Authors' change in the manuscript.**

**Page 26, Line 9: (the changes is marked as blue)**
The precise simulation of variables in the root zone could benefit the simulation of other elements in the model, thus advancing the model simulation toward an advanced philosophy, i.e., obtaining the right answers for the right reasons rather than simply obtaining the right answers (Kirchner, 2006). In addition, the WAYS model can be further improved by integrating a more sophisticated evaporation module, e.g., the STEAM model developed by Wang-Erlandsson et al. (2014), which separates the evaporation fluxes in a more detailed way. Finally, a runoff generation module recently developed by Gao et al. (2019), HSC-MCT, could provide another possibility to improve the WAYS model, as it offers another venue for determining one of the key parameters in WAYS ($\beta$) independently without calibration. This calibration-free module could actually benefit any conceptual hydrological model.

**4. I found a mistake in Equation (2). The correct equation should read: P_tf = MAX{0, P_r – (S_imax – S_i)/Delta t }. The Delta t is required to make the equation dimensionally correct and to prevent that if the model is used at another time step, no error is made. The MAX{0,x} operator is essential since P_tf is an overflow. Forgetting the MAX{0,x}**

**operator can lead to negative P_tf values for small amounts of rainfall. This may trigger relatively small errors, but particularly in wet environments (the Amazon?) this can create errors. I fear that the authors have to rerun the models to correct this mistake.**

We thank the referee for pointing out the mistake in Equation (2), which has been corrected, and the codes have been changed accordingly (see Figure 1). As a result, the model has been rerun and the results updated. We would like to state that before rerunning of the model, parameter RZSC ($S_{rz,max}$) was also updated in the model based on the comment of Referee #3 (comment 6). Referee #3 suggested that RZSC should be updated by applying the Gumbel normalization, as Wang-Erlandsson et al., (2016) found that normalizing the RZSC using the Gumbel distribution by land cover type further improves performance. The model results are updated in the revised manuscript. In comparison to the previously simulated results, both simulations of runoff and RZWS are only slightly altered due to the correction of the precipitation throughfall equation as well as the update of the RZSC data. In the Amazon basin, the rank correlation between simulated RZWS and NDII are improved from 0.533 (WAYS_CRU) and 0.506 (WAYS_CHIRPS) to 0.593 (WAYS_CRU) and 0.552 (WAYS_CHIRPS).

WAYS is currently run on a daily scale, and the **Delta t** suggested by the referee is necessary for running the model at other time scales. **Delta t** is required not only for the precipitation throughfall equation but also for all the time scale-related parameters. Therefore, we have updated the table of model equations and stated the following at the end of the table: "Note: all time scale-dependent parameters need to be divided by $\Delta t$ to make the equation dimensionally correct and suitable for any other time scales".

```
202   def intercept(pr, si, simax):
203       """interception"""
204       # ptf: precipitation throughfall
205       if simax == 0:
206           si = 0
207           ptf = pr
208       else:
209           if pr + si > simax:
210               ptf = pr - (simax - si)
211               si = simax
212           else:
213               ptf = 0
214               si += pr
215       return ptf, si
```

*Figure 1. the codes for the precipitation throughfall ($P_{tf}$) calculation*

**Authors' change in the manuscript.**

**Page 8: Table 1 is updated. (the changes are marked as blue)**

**Table 1.** Water balance and constitutive equations used in WAYS

| Reservoirs | Water balance equations | | Constitutive equations | | Reference |
|---|---|---|---|---|---|
| | | | $P_{tf} = max(0, P_r - (S_{i,max} - S_i))$ | (2) | - |
| Interception reservoir | $\dfrac{dS_i}{dt} = P_r - E_i - P_{tf}$ | (1) | $E_i = E_p \left(\dfrac{S_i}{S_{i,max}}\right)^{2/3}$ | (3) | Deardorff (1978) |
| | | | $S_{i,max} = m_c L$ | (4) | Wang-Erlandsson et al. (2014) |
| Snow reservoir | $\dfrac{dS_w}{dt} = \begin{cases} -M & \text{if } T > T_t \\ P_s & \text{if } T \leq T_t \end{cases}$ | (5) | $M = \begin{cases} min(S_w, F_{DD}(T - T_t)) & \text{if } T > T_t \\ 0 & \text{if } T \leq T_t \end{cases}$ | (6) | Rango and Martinec (1995) |
| | | | $P_e = P_{tf} + M$ | (8) | - |
| Root zone reservoir | $\dfrac{dS_{rz}}{dt} = P_e - R - E_a$ | (7) | $\dfrac{R}{P_e} = 1 - \left(1 - \dfrac{S_{rz}}{(1+\beta)S_{rz,max}}\right)^{\beta}$ | (9) | Sriwongsitanon et al. (2016) |
| | | | $E_a = (E_0 - E_i) \cdot min\left(1, \dfrac{S_{rz}}{C_e S_{rz,max}(1+\beta)}\right)$ | (10) | Sriwongsitanon et al. (2016) |
| Slow response reservoir | $\dfrac{dS_s}{dt} = R_s - Q_s$ | (11) | $R_s = min(f_s R, R_{s,max})$ | (12) | Döll and Fiedler (2008) |
| | | | $Q_s = S_s / K_s$ | (13) | Döll et al. (2003) |
| | | | $R_f = R - R_s$ | (15) | - |
| Fast response reservoir | $\dfrac{dS_f}{dt} = R_f - Q_{ff} - Q_f$ | (14) | $Q_{ff} = max(0, S_f - S_{ftr})/K_{ff}$ | (16) | - |
| | | | $Q_f = S_f / K_f$ | (17) | - |

Note: all the time scale-dependent parameters need to be divided by $\Delta t$ to make the equations dimensionally correct and suitable for any other time scales.

- in the reference column indicates that the formula is taken from the FLEX model.

**Page 14: Figure 2 is updated**
To avoid repetition, please to see the changes in response to comment 1

**Page 15: Figure 3 is updated**
To avoid repetition, please to see the changes in response to comment 1

**Page 17: Figure 4 is updated**

[Figure]

**Figure 4.** The catchment clockwise pole plot according to different metrics, (a) 1-NSE, (b) RMSE, and (c) PBIAS. Colored markers indicate the score for the WAYS model with two different simulations, and black markers represent the score for ISIMIP2a models. For all the metrics, the value of 0 is the benchmark.

**Page 18: Figure 5 is updated**

[Figure]

**Figure 5.** Time series of 8-day normalized RZWS simulated by the WAYS model and NDII value.

**Table 3.** The rank correlation of NDII and WAYS-simulated RZWS in ten selected basins

| Selected River Basins | Models | |
|---|---|---|
| | WAYS_CRU | WAYS_CHIRPS |
| Congo | 0.872 | 0.871 |
| Nile | 0.951 | 0.967 |
| Niger | 0.975 | 0.975 |
| Yangtze | 0.713 | 0.764 |
| Ganges | 0.803 | 0.817 |
| Parana | 0.931 | 0.934 |
| Amazon | 0.593 | 0.552 |
| Mississippi | 0.689 | 0.677 |
| Murray Darling | 0.614 | 0.636 |
| Mekong | 0.936 | 0.938 |

**5. In the validation against NDII, one should realise that some ecosystems (particularly Australian) tap into groundwater, so that in those ecosystems the NDII may not be the correct proxy for moisture stress in the root zone during dry periods. This may be another reason why the Murray Darling performs less well in the comparison with NDII.**

Thanks for the comment. We agree with the referee and have mention this in the manuscript.

**Authors' change in the manuscript.**

In contrast, WAYS shows a trend of underestimation in the Murray Darling. A possible reason could be that deep rooted plants are widespread across the Murray Darling basin and can tap into groundwater (Runyan and D'Odorico, 2010; Lamontagne et al., 2014); thus, the NDII may not be the correct proxy for moisture stress in this region. A vast amount of groundwater drawing from the saturated zone to the root zone could explain such underestimation of RZWS (Leblanc et al., 2011). Other reasons behind these findings could be the underestimated RZSC in this region as well as the intensive human activities, including dam construction, a water diversion system and river management, which will impact both the RZSC estimation and RZWS simulation (Reid et al., 2002; Kingsford, 2000).

**6. This brings me to another point, that the FLEX model used apparently does not include capillary rise. In wetlands, this is a dominant mechanism, and also some dryland vegetation is known to tap water from deeper layers. A landscape-based model as developed by Gao et al. (2014a) could cater for this and could also distinguish between an independently derived Beta function for the wetland-terrace-plateau continuum and a calibrated Beta for hillslopes**

Thank you for the comment. We agree that the capillary rise is important in regions in which surface water and groundwater exchanges are intense, e.g., in wetlands and regions with deep root plants. The upward capillary fluxes can impact the root zone water storage and thus the water budget between the surface and the lower atmosphere (Vergnes et al., 2014). In fact, WAYS does include the capillary module from Gao et al. (2014a), a key publication of the FLEX model. At the current stage, it is, however, disabled due to the lack of global information on the groundwater table. Of course, this will affect the simulated results in this work, e.g., the evaporation and RZWS. However, we disabled the capillary module based on our analysis.

We set up two experimental runs for the WAYS model, for which one run with the capillary module was active and the other with the capillary module inactive. For both runs, the RZSC data ($S_{R,CRU-SM}$) that are derived based on CRU, SSEBop and MOD16 are used. The model is calibrated before running for each simulation run, and the parameter range of $C_{Rmax}$ is set to (0.01, 2), where $C_{Rmax}$ is the key parameter for the capillary module that controls the maximum capillary rise. The simulation period is set to 1989-2005 for both runs, as we compared the simulated evaporation with the LandFluxEVAL data set, which is only available in this period. The LandFluxEVAL data are a merged benchmark synthesis product of evaporation on the global scale and comprise a combination of land-surface model simulations, remote sensing products, reanalysis data and ground observation data (Mueller et al., 2013). The LandFluxEVAL data are used in many studies as refence data for evaporation evaluations (Lorenz et al., 2014; Martens et al., 2017; Wartenburger et al., 2018). The results show that the simulated evaporation is significantly overestimated for the run in which the capillary module is switched on. The global averaged annual evaporation is estimated as 513 mm/year by WAYS with the capillary module switched off, while the global averaged annual evaporation is simulated as 697 mm/year by switching on the capillary module. In the LandFluxEVAL data, the global averaged annual evaporation is 491 mm/year. The significant overestimation of evaporation by WAYS with capillary module switched on.

This is mainly because there is no observed groundwater table information to constrain the capillary rise amount. Therefore, we decided to disable the capillary module in the current version of WAYS; it can be active once the global groundwater table information is available. To clarify this information in the manuscript, we have revised the text accordingly by stating our analysis-based decision regarding capillary module deactivation as well as discussing the impacts of ignoring the capillary rise. The revisions can be found in "Authors' change in the manuscript" below.

Regarding the issue of "using an independently derived Beta function for the wetland-terrace-plateau continuum and a calibrated Beta for hillslopes in the model.", we agree with the referee that this would be an interesting experiment and the derivation of parameter $\beta$ from HSC-MCT without calibration can benefit any conceptual hydrological models. However, the HSC-MCT module is heavily dependent on the HAND data, and a verified global data set on HAND is not currently available. The landscape classification for wetland, terrace, plateau and hillslopes is also based on the HAND data. Indeed, HAND can be simply derived from DEM data. However, HAND data that are not well verified could potentially introduce large uncertainties because HAND is very sensitive to the drainage threshold and the open water elevation (Nobre et al., 2016).

Given that the manuscript is already quite extensive, including these results would not necessarily contribute to improving the manuscript clarity. In addition, it is not the main objective of the paper to focus on deriving the beta function independently from the HAND data. Therefore, we prefer not to include these results in the paper.

**Authors' change in the manuscript.**

**Page 25, Line 4: The following paragraph is inserted in the discussion part**
Moreover, the current study does not consider the groundwater access and irrigation mainly due to the lack of global information. The groundwater table information is crucial for capillary rise simulation (Vergnes et al., 2014). Capillary rise simulation without proper water table information could significantly overestimate the evaporation. Thus, the capillary rise flux is ignored in this study. A similar strategy has also been applied by other works due to the absence of the information on the global water table (Döll et al., 2003; De Graaf et al., 2015; Hanasaki et al., 2018). Observations of irrigation on the global scale are also not available (Leng et al., 2015). Although there are simulated irrigation data available on the global scale, the inherent uncertainties could be propagated in our model simulation. Therefore, irrigation is also not considered at this time. However, this neglect could potentially introduce biases into the model simulation in irrigated areas and deep rooted plant-distributed regions, as both irrigation and capillary rise are an additional supply of soil water recharge. The biases may cause an underestimation of evaporation, especially in the dry summertime (Vergnes et al., 2014). This underestimation could consequently affect the simulation of RZWS and runoff because of the interlinkage of these three elements (Rockström et al., 1999). It is found that ignoring the capillary rise could reduce soil water content in the root zone (RZWS), while the runoff will also be reduced (Vergnes et al., 2014). However, these shortcomings can be simply overcome once the global data are available.

**7. I don't understand the last sentence in the abstract. Indeed CHIRPS-CSM is limited to lower latitudes, but CRU-SM covers the entire globe. I think the sentence "Therefore, the performance etc." can be deleted.**

Thanks for the comment. We have deleted this sentence in the manuscript.

Authors' change in the manuscript.

Page 1, Line 14:

The following sentence is deleted in page 1, line 12.

**8. There are many typos. I think the paper requires copyediting, which probably Copernicus can take care of.**

Thank you for the comment. The revised manuscript has now been edited by a professional academic language and manuscript service company.

**So in summary, I think this is an important paper, but additional work needs to be done before the paper can be published.**

**The references used in this comment also occur in the discussion paper, except the following:**
**Gao, H., Birkel, C., Hrachowitz, M., Tetzlaff, D., Soulsby, C., and Savenije, H. H. G., 2019. A simple topography-driven and calibration-free runoff generation module, Hydrol. Earth Syst. Sci., 23, 787-809, https://doi.org/10.5194/hess-23-787-2019.**

We would like to express our sincere thanks again to Referee Prof. Hubert H.G. Savenije for his time reviewing our manuscript and for the valuable comments to improve our manuscript.

**Reference:**
Nobre, A. D., Cuartas, L. A., Momo, M. R., Severo, D. L., Pinheiro, A. and Nobre, C. A.: HAND contour: A new proxy predictor of inundation extent, Hydrol. Process., 30(2), 320–333, doi:10.1002/hyp.10581, 2016.

---

## Author Comment (AC2) · 26 Jul 2019

We would like to thank Referee #2 for his/her interest in this topic and for the valuable comments to improve our manuscript. Based on the comments some calculations have been performed. Our point-by-point response to the comments is given in the following (**Comments in black**, Answers in blue and the content related to the changes in the revised manuscript are marked in orange.):

**A correct representation of root zone water storage is important for a robust hydrological modeling. However, in reality, obtaining reliable soil water information is difficult. In many previous studies, the root zone water storage has been quantified as the soil moisture in a certain depth rather than the water stored in the entire rooting system. This leads to an under- or over- estimation of root zone water storage depending on individual site conditions, including the types of vegetation covers on the land surface. The aim of the paper by Mao and Liu is to develop a hydrological model, WAYS, that is capable for simulating root zone water storage on a global scale, without constraining the quantification to a certain depth. Overall, I think the development of such a model is valuable to the hydrological community and can largely advance the eco-hydrological studies which tackle the interactions between the hydrological cycle and vegetation dynamics on the land surface. I personally also think with the further development and improvement, WAYS has the potential to be applied in the investigation of land-vegetation-climate-water integrations which is very important for the global change impact assessments. Below I give some comments on the paper and hope the authors can address them in the revision.**

We would like to thank the referee for this comment. Based on the referee's comments, our point-to-point reply to the comments is given in the following.

**General comments**

**1. The model structure of WAYS is the core of this paper (Figure 1). Its scientific clarity is essential for others to understand the processes and also is important for possible future wider applications of the model beyond the authors' group. In the current version, the variables in the flow chat depicted in the small window and the ones in the schematic are not all matched. E.g., in the small window, $S_i$, $P_e$, $R_r$, $S_f$, and $S_s$ are used, but they are not indicated in the schematic. Even if their meanings are clear (some are not clear to me), the authors still need to denote them properly in the schematic. In addition, given the central role of Figure 1 in the entire paper, I suggest the authors to add some text elaborating the flow of the figure. This is different from the following sections describing individual processes in the model.**

Thank you for the comment. We have updated the figure accordingly. $S_f$ and $S_s$ are the two conceptual reservoirs in the model representing the fast and slow response in hydrological cycling. They are slightly different from the other three conceptual reservoirs, $S_i$, $S_{rz}$ and $S_w$, which represent the actual water storages (Gao et al., 2014). Therefore, we have marked these response reservoirs with a dashed line in the schematic. Moreover, some of the fluxes are intermediate variables, e.g., $R_f$ is the generated preferential runoff in the root zone layer before the split of runoff into surface runoff and subsurface runoff. The effective precipitation $P_e$ is the sum of snowmelt and precipitation throughfall. These fluxes are

shown in the flowchart but cannot be properly visualized in the schematic drawing. We have explained this mis-match issue in the text in the revised manuscript.

In addition, we have added a few sentences to further elaborate the flow of the figure in the revised manuscript. The changes can be found in the following section "Authors' change in the manuscript."

**Authors' change in the manuscript.**

**Page 4: Figure 1 is updated**

[Figure]

**Figure 1.** Model structure of WAYS

**Page 3, Line 25: The following text is inserted.**
In Figure 1, the flowchart represents the conceptualized hydrological cycle in the model, and the schematic drawing shows the corresponding water fluxes and stocks in the real world. Since some of the fluxes are intermediate variables, they are shown in the flowchart but not visualized in the schematic drawing. For instance, Rf is the generated preferential runoff in the root zone layer before the split of the runoff into surface runoff and subsurface runoff. The effective precipitation Pe is the sum of snowmelt and precipitation throughfall. The conceptualized hydrological cycle of the model can be briefly described as follows. The precipitation that can drop as rainfall or snowfall depends on the temperature. The snowfall will be stored in the snow reservoir, and the rainfall will be intercepted by the canopy before it reaches the surface. After the interception, the rainfall penetrates the canopy and reaches the surface as precipitation throughfall. The effective precipitation that consists of the throughfall and the snowmelt will partially infiltrate into the soil, and the rest runs away as runoff. The runoff is then split into surface runoff and subsurface runoff depending on the texture. A part of the infiltration will be stored in the soil for plants, and the rest will percolate into the deep soil and reach the groundwater table as groundwater recharge.

**2. Table 1 gives all the equations concerning the water balance in WAYS. This is very useful for examining the processes and evaluating the robustness of the model. As all the equations are from the relevant literature (the authors give the references in the text), it would be good to provide the major references in the last column of Table 1.**

Thank you for the comment. We have updated the Table 1 accordingly. The updated table can be found in the following section "Authors' change in the manuscript."

**Authors' change in the manuscript.**

**Page 8: Table 1 is updated. (the changes are marked as blue)**

**Table 1.** Water balance and constitutive equations used in WAYS

| Reservoirs | Water balance equations | Constitutive equations | | Reference |
|---|---|---|---|---|
| | | $P_{tf} = max(0, P_r - (S_{i,max} - S_i))$ | (2) | - |
| Interception reservoir | $\frac{dS_i}{dt} = P_r - E_i - P_{tf}$ (1) | $E_i = E_p \left(\frac{S_i}{S_{i,max}}\right)^{2/3}$ | (3) | Deardorff (1978) |
| | | $S_{i,max} = m_c L$ | (4) | Wang-Erlandsson et al. (2014) |
| Snow reservoir | $\frac{dS_w}{dt} = \begin{cases} -M & \text{if } T > T_t \\ P_s & \text{if } T \leq T_t \end{cases}$ (5) | $M = \begin{cases} min(S_w, F_{DD}(T - T_t)) & \text{if } T > T_t \\ 0 & \text{if } T \leq T_t \end{cases}$ | (6) | Rango and Martinec (1995) |
| | | $P_e = P_{tf} + M$ | (8) | - |
| Root zone reservoir | $\frac{dS_{rz}}{dt} = P_e - R - E_a$ (7) | $\frac{R}{P_e} = 1 - \left(1 - \frac{S_{rz}}{(1+\beta)S_{rz,max}}\right)^\beta$ | (9) | Sriwongsitanon et al. (2016) |
| | | $E_a = (E_0 - E_i) \cdot min\left(1, \frac{S_{rz}}{C_e S_{rz,max}(1+\beta)}\right)$ | (10) | Sriwongsitanon et al. (2016) |
| Slow response reservoir | $\frac{dS_s}{dt} = R_s - Q_s$ (11) | $R_s = min(f_s R, R_{s,max})$ | (12) | Döll and Fiedler (2008) |
| | | $Q_s = S_s / K_s$ | (13) | Döll et al. (2003) |
| Fast response reservoir | $\frac{dS_f}{dt} = R_f - Q_{ff} - Q_f$ (14) | $R_f = R - R_s$ | (15) | - |
| | | $Q_{ff} = max(0, S_f - S_{ftr})/K_{ff}$ | (16) | - |
| | | $Q_f = S_f / K_f$ | (17) | - |

Note: all the time scale-dependent parameters need to be divided by $\Delta t$ to make the equations dimensionally correct and suitable for any other time scales.

- in the reference column indicates that the formula is taken from the FLEX model.

**3. Are there any other values used for Rx,max rather than 7, 4.5 and 2.5 for sandy soil, loamy soil and clayey soil? May be worth a checking for uncertainties stemmed from the use of Rx,max values for the mentioned soils.**

Thank you. Since the groundwater recharge module in WAYS is based on the work of Döll and Fiedler (2008), the values are directly taken from that publication. These values are also used for other global groundwater recharge simulation-related works, e.g., Müller Schmied et al. (2014). However, these are indeed empirical parameter values. We agree with the

referee that the uncertainty/sensitivity analysis is necessary for $R_{s,max}$. We have performed the uncertainty/sensitivity analysis, and the results are shown in Supplementary Information (SI).

**The following part is put in the SI.**
Since the groundwater recharge module in WAYS is based on the work of Döll and Fiedler (2008), the values are taken directly from it. These values are also used for other global groundwater recharge simulation-related works, e.g., Müller Schmied et al. (2014). However, as these are indeed empirical parameter values, uncertainty/sensitivity analysis is necessary for $R_{s,max}$. Three pixels from different soil types are selected for the $R_{s,max}$-induced uncertainty investigation. Figure S1 shows the grouped soil texture classes for this study based on the FAO Harmonized World Soil Database and the selected pixels for the uncertainty analysis. Pixel 1, pixel 2 and pixel 3 represent the soil type of clay, loam and sand, respectively.

$R_{s,max}$ (mm/day) directly influences the matrix flow (contributes 100% to groundwater recharge with a certain time lag) based on equation 12 in Table 1 in the manuscript, as it controls the maximum groundwater recharge for different soil types. Consequently, it will also impact the preferential flow, as the runoff is partially split into matrix flow and the rest to the preferential flow. Therefore, parameter $R_{s,max}$ will have light effects on the runoff generation but could have considerable impacts on the matrix flow and preferential flow. Thus, the sensitivity of the simulated preferential flow and matrix flow to the maximum groundwater recharge $R_{s,max}$ is investigated, and the results are shown in Figure S2 (pixel with clayey soil), Figure S3 (pixel with loamy soil) and Figure S4 (pixel with sandy soil). The sensitivities of WAYS to $R_{s,max}$ are checked by perturbing the parameter. We set the simulation with soil texture-specified $R_{s,max}$ as the control run, and perturbed $R_{s,max}$ by -80%, -50%, -20%, 20%, 50% and 80% for the sensitivity test. Figure S2 (bottom plot) shows the impacts of the values of $R_{s,max}$ on the simulated daily matrix flow with the soil type of clay. With the increase in $R_{s,max}$, the simulated daily matrix flow has a higher peak, while the opposite is observed with the decrease in $R_{s,max}$. It changes the scale of the simulated matrix flow but not its shape at the daily scale. Moreover, due to the change in daily simulation, the monthly simulation of the matrix and preferential flow are affected accordingly, as seen in Figure S2 (top and middle plots). The results show that parameter $R_{s,max}$ has opposite impacts on preferential flow and matrix flow, which is logical because both are part of the runoff. A similar phenomenon is found in daily simulated time series. Thus, for pixel 2 and pixel 3, only monthly simulated matrix and preferential flow are shown to visualize the uncertainties stemming from $R_{s,max}$ for loamy and sandy soil. The simulated matrix flow time series with a decreased value of $R_{s,max}$ shows are found to have larger uncertainties than time series with an increased value of $R_{s,max}$, because the maximum value of matrix flow is not only determined by $R_{s,max}$ but also the groundwater recharge factor $f_s$.

[Figure]

*Figure S1. Grouped soil texture classes for the study based on the FAO Harmonized World Soil Database.*

*Figure S2. Sensitivity of the simulated preferential flow and matrix flow to the maximum groundwater recharge $R_{s,max}$ for the pixel with clayey soil.*

[Figure]

[Figure]

*Figure S3. Sensitivity of the simulated preferential flow and matrix flow to the maximum groundwater recharge $R_{s,max}$ for the pixel with loamy soil.*

[Figure]

[Figure]

*Figure S4. Sensitivity of the simulated preferential flow and matrix flow to the maximum groundwater recharge $R_{s,max}$ for the pixel with sandy soil.*

**4. In the captions of Figures 4 and 5, the ERA-Interim/Land represents the reference data. I think it is better to directly use ERA-Interim/Land here, because in the Figures, the ERA is used and no reference data is indicated. Also in Figure 4, WAYS-CHIRPS is not visible. Need to give a note for it, e.g., covered by . . .. The scale for Y axis for Murray Darling should be enlarged to show the simulated runoff more clearly.**

Thank you for the comment. Figure 4 and Figure 5 are updated accordingly. The scale for the Y axis for Murray Darling is slightly enlarged because the uncertainty range from the other model is quite large. A big enlargement will erase the simulations from ISIMIP2a models. We would also like to note that because Referee #1 suggested us to mention the non-calibration issue of ISIMIP2a model simulations in the captions of Figures 4 and 5, these changes are also shown in the captions. In addition, in short comment #1 (comment 10), the reviewer suggested us to move the Figures 2 and 3 from the main text of the manuscript. Thus, the figures in the revised manuscript are re-sorted.

**Authors' change in the manuscript.**

**Page 14: Figure 2 caption is updated**

[Figure]

**Figure 2.** Time series of monthly runoff simulated by WAYS and the ISIMIP2a models, as well as the reference data. The basins highlighted in the world map indicate the selected catchments for model evaluation. The solid lines in blue and red indicate the WAYS simulations with two different RZSC products. The solid line in black indicates the ERA-Interim/Land data, and dashed lines represent the ISIMIP2a model simulations. In some plots, the red line is not visible and is covered by the blue line due to the small differences between the two WAYS runs. WAYS is calibrated using Composite Monthly Runoff data, while the ISIMIP2a models are not calibrated for the simulation.

**Page 15: Figure 3 caption is updated**

[Figure]

**Figure 3.** The probability of exceedance for monthly runoff simulated by different models as well as the reference data in ten selected basins. The solid lines in blue and red indicate WAYS simulations with two different RZSC products. The solid line in black indicates the ERA-Interim/Land data, and dashed lines represent the ISIMIP2a model simulations. In some plots, the red line is not visible and is covered by the blue line due to the small differences between the two WAYS runs. WAYS is calibrated using Composite Monthly Runoff data, while the ISIMIP2a models are not calibrated for the simulation.

**5. The authors demonstrated the good performance of WAYS compared to ISIMIP2a models. However, no direct reasons are given to explain the better performances. I assume that the authors want to say that this is because of the better representation of the root zoon water storage in WAYS. The authors should make this point clear. It justifies the effort for developing WAYS in this paper. Also, it seems to me not very convincing to state that the better performance is really from the better representation of the root zoon water storage. Could some other processes in the WAYS model be also influential for the better performance compared with the results from the models in ISIMIP2a?**

Thank you for the comment. In fact, the simulated runoff from WAYS is first compared with the reference data ERA-Interim/Land runoff. The performance of WAYS in runoff simulation is evaluated mainly based on the comparison between ERA-Interim/Land data and WAYS simulation.

Since WAYS uses the same driving data as the ISIMIP2a models and the ISIMIP2a simulations are widely studied and discussed in many studies, we believe the additional comparison

between WAYS and ISIMIP2a models can provide added-value for examining our model. Therefore, ISIMIP2a simulations are also shown in the results together with the ERA-Interim/Land data. It is also important to state that the ISIMIP2a models are not calibrated for global simulation, which could explain the better performance of WAYS than the ISIMIP2a models.

However, it is not easy to justify the impacts of better representation of root zone water storage without the comparative experiment. In this regard, we have performed an experimental test to investigate whether the better performance is truly due to the better representation of the root zone water storage. Since this part could support the conclusion of this paper regarding the importance of correct representation of RZSC in models, we would like to include this it in the main text of the manuscript by including some figures in the Supplementary Information (SI). Please refer to the changes in the revised manuscript.

**Authors' change in the manuscript.**

**Page 21, Line 26: (the following paragraphs are added)**
RZSC is a key parameter of the WAYS model. Therefore, it is important to investigate how RZSC could affect the model simulation. In addition to the model simulated with satellite data-derived RZSC products ($S_{R,CHIRPS-CSM}$ and $S_{R,CRU-SM}$), we have additionally conducted WAYS simulations with RZSC derived from uncertain root depth and soil data. The uncertain RZSC ($S_{R,LOOKUP-TABLE}$) is derived based on literature values of root depth and soil texture data (Müller Schmied et al., 2014; Wang-Erlandsson et al., 2016). Due to the global coverage of the RZSC data ($S_{R,CRU-SM}$), only the simulation with $S_{R,CRU-SM}$ is used for comparison. The spatial distribution of the uncertain RZSC is shown in Figure S17, and the differences between $S_{R,CRU-SM}$ and $S_{R,LOOKUP-TABLE}$ are shown in Figure S18. It can be seen that there are large differences between the two RZSC products. The simulation with uncertain RZSC $S_{R,LOOKUP-TABLE}$ shows overestimation globally except for some regions around low-middle latitudes. The latitudinal averaged RZSC further confirms the overestimation of $S_{R,LOOKUP-TABLE}$ at middle-high latitudes (Figure S19).

The large differences between these two RZSC data sets also introduce differences in simulated hydrological elements. Figure S20 shows the impacts of RZSC on the model simulation, including runoff, evaporation and RZWS. A blue color (decrease of RMSE and increase of the ranked correlation) indicates an improvement of the simulated results by replacing the uncertain RZSC ($S_{R,LOOKUP-TABLE}$) with satellite data-derived RZSC ($S_{R,CRU-SM}$), while a red color implies the opposite. For comparison, reference data are used for different variables. For runoff, evaporation and RZWS, the reference data are ERA-Interim/Land (2001-2010, monthly), LandFluxEVAL (1989-2005, monthly) and NDII (2001-2010, 8-days), respectively. Generally, the model simulations are improved by using the RZSC $S_{R,CRU-SM}$. This result emphasizes the importance of an appropriate representation of RZSC in WAYS. A decline of the model performance is also found in some
regions at high latitudes and low latitudes. This result can be partially explained by the inherent uncertainty in the $S_{R,CRU-SM}$ data, as they are derived from other data sets. The RZSC derivation method itself as well as the input data can also introduce biases (Wang-Erlandsson et al., 2016).

[Figure]

**Figure 7.** The improvement of RMSE in evaporation simulations for different land covers by using the satellite data-derived RZSC ($S_{R,CRU-SM}$) instead of the uncertain RZSC ($S_{R,LOOKUP-TABLE}$).

Figure 7 shows the RMSE improvements of simulated monthly evaporation for different land covers obtained by implementing the satellite data-derived RZSC ($S_{R,CRU–SM}$) instead of the uncertain RZSC ($S_{R,LOOKUP–TABLE}$). The analysis reveals that the satellite data-derived RZSC ($S_{R,CRU–SM}$) has great potential to improve the evaporation simulation for all kinds of land covers. The largest improvements are found in broadleaf forests. The improvements in the needleleaf forest, mixed forest and savanna are relatively low. The findings also resonate with another work that used a simple terrestrial evaporation to atmosphere model (STEAM) for evaporation simulation (Wang-Erlandsson et al., 2016).

**The following figures can be found in the SI of this paper.**

[Figure]

Root Zone Storage Capacity (mm)

*Figure S5. Spatial distribution of uncertain RZSC (SR,LOOKUP-TABLE).*

[Figure]

Root Zone Storage Capacity (mm)

*Figure S6. The difference between $S_{R,CRU-SM}$ and $S_{R,LOOKUP-TABLE}$ ($S_{R,LOOKUP-TABLE}$ - $S_{R,CRU-SM}$).*

[Figure]

*Figure S7. Latitudinal averaged RZSC of different products.*

[Figure]

*Figure S8. The impacts of RZSC on the model simulation. Blue color indicates the improvement of the simulated results by replacing the uncertain RZSC ($S_{R,LOOKUP-TABLE}$) with satellite data-derived RZSC ($S_{R,CRU-SM}$), while red color implies the opposite. (a) The result for runoff and the reference data for comparison is ERA-Interim/Land data (2001-2010, monthly), (b) the result for evaporation and the reference data for comparison is*

*LandFluxEVAL data (1989-2005, monthly), and (c) the result for RZWS and the reference data is NDII data (2001-2010, 8 days).*

**6. In Figure 4, the authors stated that in the Murray Darling basin, WAYS performed very well in comparison to the ERA data for runoff. In Figure 7, the difference between simulated root zone water storage and the NDII values is quite large. The similar situation is also seen in Mississippi, Amazon and Yangtze. The correlation values provided in Table 2 are rather low for these river basins. The authors stated that this could be caused by either the uncertainty of WAYS or the problem of using NDII as a proxy of root zone water storage in the specific river basin. In general, I think this is reasonable. However, I still feel that some specific reasons should be highlighted with convincing evidence, instead of just saying this is either due to the problem of WAYS or the use of NDII. Besides, in the discussion, it would be good if the authors can give some suggestions on validation of root zone water storage simulations when the validity of using NDII for validation is not so suitable as shown in the above mentioned river basins.**

Thank you for the comment. Indeed, the runoff simulation in the Murray Darling basin is much better than RZWS. Since the model is calibrated to the runoff, the performance of the runoff generation could potentially surpass the other variables, e.g., RZWS and evaporation. However, we also agree with the referee that the large difference between NDII and simulated RZWS in some basins could imply other potential issues that may affect the simulation in addition to the model structure itself. We have further strengthened this part of the manuscript, and the corresponding changes can be found in the revised version (see "Authors' change in the manuscript.")

Moreover, we have also discussed the validation issue regarding the possible inappropriate representation of NDII in some basins.

**Authors' change in the manuscript.**

**Page 20, Line 8: (the changes is marked as blue)**
In contrast, WAYS shows a trend of underestimation in the Murray Darling. A possible reason could be that deep rooted plants are widespread across the Murray Darling basin and can tap into groundwater (Runyan and D'Odorico, 2010; Lamontagne et al., 2014); thus, the NDII may not be the correct proxy for moisture stress in this region. A vast amount of groundwater drawing from the saturated zone to the root zone could explain such underestimation of RZWS (Leblanc et al., 2011). Other reasons behind these findings could be the underestimated RZSC in this region as well as the intensive human activities, including dam construction, a water diversion system and river management, which will impact both the RZSC estimation and RZWS simulation (Reid et al., 2002; Kingsford, 2000).

**Page 19, Line 20: (the changes is marked as blue)**
In the Mississippi,
WAYS shows a good performance in large-value simulations, while it struggles to simulate low values, with considerable overestimation of them. Therefore, the rank correlation is also relatively low in this catchment, with values of approximately 0.67. The Mississippi river

basin is the northernmost catchment of our selected basins. The NDII here shows a totally different pattern compared to the others, while the WAYS-simulated RZWS can barely show a clear seasonal variation. There could be multiple reasons for this overestimation: our model has a relatively simple snowmelt module (degree-day method), which could consequently introduce biases into the simulation, especially in relatively cold regions. Additionally, the relatively uncertain forcing data could contribute to the mismatches between NDII and RZWS, as the largest uncertainties in precipitation occur mainly at the higher latitudes (Vinukollu et al., 2011). Some studies also reported that precipitation-induced spurious seasonal and interannual variations also exist in the soil moisture in this basin (Yang et al., 2015).

**Page 24, Line 22: (the changes is marked as blue)**
However, we have to highlight that the model shows lower performance in some regions, e.g., the Amazon, in the RZWS simulation, where the reference data NDII may have shortcomings in reflecting RZWS. In these regions where NDII might not be a correct proxy for RZWS, an additional data set could be helpful for evaluation, e.g., the solar-induced fluorescence (SIF), which reflects photosynthesis and thus has a close relationship to the available water in the root zone. A combination of vegetation index data, such as EVI and NDVI, could also be alternatives, as they represent different characteristics of plants. However, further investigations need to be performed before this combination can be applied.

**7. Page 21, last paragraph. It is stated that 'this added value feature could benefit for many applications related to the root zone processes.' The authors should specify some of the potential benefits here.**

Thank you. We have specified the potential benefits of the developed model accordingly. The updates can be found in the revised manuscript.

**Authors' change in the manuscript.**

**Page 28, Line 32: (the changes is marked as blue)**
This added-value feature could benefit for many applications related to the root zone processes. For instance, the correct representation of RZWS could help the researchers in the investigation of land-vegetation-climate-water integrations, where RZWS plays a key role. The capacity of RZWS simulation could also bring benefit to the field of agriculture, as RZWS represents the plant available water that closely linked with the crop yields.

**8. The aim of the paper is to develop WAYS which is capable of simulating root zone water storage. In the model evaluation section, much text is about the validation of runoff. The elaboration of the importance to correctly represent root zone water storage and the good performance of WAYS in realizing this goal is relatively brief. It would be good if the authors can strengthen this part of the text to highlight the accomplishment of the paper.**

Thank you. We agree with the referee that the elaboration of the importance of correct representation of root zone water storage is relatively brief. Therefore, we have performed additional work on the model evaluation to strength this portion of the manuscript. We provide a detailed reply regarding this issue in the previous comment section (comment 5). To avoid repetition, we would like to refer to the response to comment 5. The corresponding revision in the manuscript can also be found there.

**9. I like the philosophy stated in the end of the paper, 'get the right answers for the right reasons rather than simply to get the right answers'. In this paper, I feel that the right results are clearly shown. But the right reasons, to me, are relatively weak. The good performance in runoff and root zone water storage simulations could be good results, but reasons for the good results needs to be more clearly and explicitly explained and supported by evidence.**

Thank you for the comment. We agree with the referee that in the current version of the paper, the model evaluation part is relatively weak. Hence, we have performed an additional evaluation to demonstrate the importance of proper representation of the RZSC in hydrological models. We provide a detailed reply regarding this issue in the previous section (comment 5). To avoid repetition, we would like to refer to the response to comment 5. A comparative experiment is established to observe the impacts of RZSC. The results reveal that correct representation of the RZSC could significantly improve the model simulation, including runoff, evaporation and RZWS. This finding also confirms one of the objectives of our paper regarding the advanced hydrological philosophy to "get the right answers for the right reasons rather than simply to get the right answers".

**Specific comments**

**10. The window in Figure 1 should be enlarged, as it is important to show components and their connections clearly. Anyway, there is space in Figure to accommodate the enlargement.**

Thank you for the comment. It is addressed together with the general comment 1 by enlarging the flowchart in Figure 1.

**11. The manuscript contains many typos and grammatical mistakes. A professional editing of the manuscript is necessary, particularly because I think the paper has the potential to be an important paper in the field and could receive a high citation in the coming years.**

Thank you for the comment. The revised manuscript has now been edited by a professional academic language and manuscript service company.

**All the references are included in the manuscript.**

---

## Author Comment (AC3) · 26 Jul 2019

We would like to thank Referee #3 for his interest in this topic and for the valuable comments to improve our manuscript. Based on the comments additional calculations have been performed. Our point-by-point response to the comments is given in the following (**Comments in black**, Answers in blue and the content related to the changes in the revised manuscript are marked in orange.):

**General comments**
**#1 The manuscript presents an interesting extension of the FLEX model with enhanced capability for root zone storage simulation at the global scale. Root zone storage capacity is an Achilles heel in global hydrological modelling that is crucial for determining water stress, but most often dependent on highly uncertain soil and rooting depth data. Thus, the authors are addressing an important issue of high relevance for the hydrological modelling community. However, among other improvement possibilities, I think that the analyses need to be more systematic and rigorous, and the manuscript need to better communicate the motivations underlying the developers' choices. I think the manuscript merits to be published after a major revision. My main concerns are the following:**

We would like to thank the referee for assessing the quality of the paper and for providing very constructive and valuable comments. Indeed, the root zone storage capacity (RZSC) is a persistent weakness in global hydrological modeling, while it is crucial for water fluxes partitioning. This is the primary motivation of our work to integrate an advanced RZSC dataset into a hydrological model and to test the capacity of the model for root zone water storage (RZWS) simulation. Based on the referee's comments, we have performed additional computation and analyses to make the results more systematic and rigorous. A point-to-point reply to each specific comment is provided below.

**#2 The manuscript could benefit from clearer descriptions of rationale and motivations for the model development, the analyses performed and other choices made. For example, why was runoff selected for evaluation against ERA-Interim/Land and the non-calibrated ISIMIP simulations? Why not use gauged data for a selection of basins and the ERA-Interim/Land and ISIMIP for global gridded comparisons? Were other variables and potentially better datasets considered and rejected for which reasons? How come capillary rise is disregarded on the basis of "lack of information at the global scale" is there are other models that take it into account? Why was Penman-Monteith FAO 56 PM method used (P6L25)? What were the considerations? Etc. Reviewers and readers will always have different views on preferred evaluation datasets and equations, but a clear description of the underlying rationale and motivation could help bridging differences in perspective if choices can be well-justified.**

Thank you for the comments. The corresponding responses are provided below. Since there are a numerous questions in this comment, we have repeated the specific comment before the detailed response.

**#2-1 For example, why was runoff selected for evaluation against ERA-Interim/Land and the non-calibrated ISIMIP simulations?**

The gridded data set ERA-Interim/Land is selected for model evaluation mainly because the current version of the WAYS model does not include a runoff routing module on the global scale. Therefore, the results are not comparable to the observed gauged data. The ERA-Interim/Land data set is a global land surface reanalysis data. It is well assessed and has been used as reference data for many studies (Alfieri et al., 2013; Orth and Seneviratne, 2015; Reichle et al., 2017). Thus, the evaluation of WAYS against ERA-Interim/Land is well-justified.

Since WAYS uses the same driving data as the ISIMIP2a models and the ISIMIP2a simulations are widely discussed in many studies, we believe that the additional comparison between WAYS and the ISIMIP2a models can provide added-value for evaluating our model. Therefore, the ISIMIP2a simulations are also shown in the results section together with the ERA-Interim/Land data. We did mention the purpose of inclusion of the ISIMIP2a simulations in the results (page 13, line 15: "Since WAYS uses the same driving data as the ISIMIP2a models and the ISIMIP2a simulations have been widely discussed in many studies (Schewe et al., 2014; Müller Schmied et al., 2016; Gernaat et al., 2017; Zaherpour et al., 2018), we also perform a comparison between WAYS and the ISIMIP2a models to further evaluate our model."). However, we did not mention this in the validation strategy section in the manuscript. We have now further clarified this issue in the revised manuscript (please see "Authors' change in the manuscript.").

Indeed, the ISIMIP2a models are not calibrated. We have mentioned this issue in the manuscript (page 13, line 30: This result occurs partly because some of the ISIMIP2a models are not calibrated at all (Zaherpour et al., 2018), whereas WAYS is calibrated to a Composite Monthly Runoff data set that assimilates the monitored river discharge (Fekete et al., 2011).). We have now revised the captions of related figures (Figures 2 and 3 in the revised manuscript) to note this issue.

**Authors' change in the manuscript.**

**Page 11, Line 5: (the changes is marked as blue)**
In this study, the ERA-Interim/Land runoff data are used for validation of the runoff simulation, and the Normalized Difference Infrared Index (NDII) is used for the validation of the WAYS model for root zone water storage simulation. Considering the time period of coverage of both data sets (ERA-Interim/Land: 1979-2010, NDII: 2000-present) and the study period (1971-2010) of this work, the period 2001-2010 is selected as the validation period. For runoff evaluation, ISIMIP2a simulations are also included, as they use the same climate forcing as our study in the same period. The purpose of inclusion of the ISIMIP2a simulations for comparison can be found in the model evaluation section (see Section 4).

**Page 11, Line 9: (the changes is marked as blue)**
ERA-Interim/Land is a global land surface reanalysis data set produced by the European Centre for Medium-Range Weather Forecasts (ECMWF) (Balsamo et al., 2015). The gridded data set ERA-Interim/Land is selected for model evaluation mainly because the current version of the WAYS model does not include a runoff routing model on the global scale. Therefore, the results are not comparable with observed gauge data. Since the ERA-

Interim/Land data set is well assessed with a quality check through comparison with ground-based and remote sensing observations, it has been used as reference data for many studies (Xia et al., 2014; Dorigo et al., 2017).

**Page 13, Line 15: (the changes is marked as blue)**
"Since the ISIMIP2a simulations are widely discussed in many studies (Schewe et al., 2014; Müller Schmied et al., 2016; Gernaat et al., 2017; Zaherpour et al., 2018), the comparison between WAYS and the ISIMIP2a models can provide added-value for evaluation in addition to examine only with the reference data." is changed to "Since WAYS uses the same driving data as the ISIMIP2a models and the ISIMIP2a simulations have been widely discussed in many studies (Schewe et al., 2014; Müller Schmied et al., 2016; Gernaat et al., 2017; Zaherpour et al., 2018), we also perform a comparison between WAYS and the ISIMIP2a models to further evaluate our model."

**#2-2 Why not use gauged data for a selection of basins and the ERA-Interim/Land and ISIMIP for global gridded comparisons?**

The simulated results are not compared to the gauged data because the current version of WAYS does not include a runoff routing module on the global scale. Therefore, the results are not comparable to the observed gauged data.

The evaluation of runoff is performed on the basin scale rather because it is difficult to show the global gridded comparisons for the time series simulated runoff. Thus, ten major basins are selected for the validation based on the coverage of the two RZSC datasets ($S_{R,CRU-SM}$ and $S_{R,CHIRPS-CSM}$). However, we agree with the reviewer that a comparison with gauged data is important. Thus, additional calculations have been performed. In the revised manuscript, we have evaluated our results with observed discharges from the Global Runoff Data Centre (GRDC). Since WAYS does not have a native runoff routing module at the moment, a third-part runoff routing tool CaMa-flood is applied to route the WAYS simulated runoff (Yamazaki et al., 2011). Given that the manuscript is already quite extensive, the discharge comparison is not a direct evaluation to WAYS but an evaluation of both WAYS and CaMa-Flood. We have included this information in Supplementary Information (SI).

**Authors' change in the manuscript.**

**Page 18, Line 4: (the following paragraph is inserted in the end of runoff evaluation section)**
The performance of WAYS is further evaluated against the gauge observations. Since WAYS does not have a native runoff routing module at the moment, a third-part runoff routing tool, CaMa-flood, is applied to route the WAYS simulated runoff (Yamazaki et al., 2011). The evaluation results can be found in the Supplementary Information (SI).

**The following part is put in the SI.**
To further evaluate the model performance, we have evaluated our results with observed discharges from the Global Runoff Data Centre (GRDC). The CaMa-flood model is the only available open-source global runoff routing model (http://hydro.iis.utokyo.ac.jp/~yamadai/cama-flood/) that is capable of simulating backwater effects, which is important for plain regions, making it a popular choice for many studies (Hirabayashi et al., 2013; Mateo et al., 2014; Pappenberger et al., 2012).

The GRDC stations along a river were selected with interstation areas larger than 7000 km$^2$ to omit catchments with hydrological processes that are not properly represented by global hydrological models operating at a 0.5° resolution (Hunger and Döll, 2008). In total, 154 stations are selected for major river basins worldwide. For discharge simulation, the CaMa-flood is run at a 0.5° resolution to maintain consistency with the WAYS simulated runoff. The WAYS_CRU simulation is used for routing due to global coverage of the data. The discharge is simulated for the 1971-2010 period.

For the evaluation, the simulated discharge is compared with the GRDC data at each selected station depending on the data availability. Since the observations provided by GRDC are on a monthly time scale, the simulated data are also aggregated to the monthly scale for the comparison. The correlation coefficient and Nash-Sutcliffe efficiency coefficient are calculated, while the correlation coefficient between the simulated discharges and GRDC station records are visualized in Figure S1.

[Figure]

*Figure S1. The evaluation of simulated discharge by comparison with the GRDC observations. The discharge is simulated by the CaMa-flood model, and the WAYS simulated runoff based on the RZSC data ($S_{R,CRU-SM}$) is used as the input data for routing. The background of the figure is the annual averaged discharge for the 1971-2010 period. The point indicates the correlation coefficient between simulated discharge and GRDC observations. The location of the points*

*implies the location of the GRDC station. Different colors at the points represent the magnitudes of the correlation coefficient.*

The simulation shows a generally good correlation with the GRDC observations, while poor performance in the discharge simulation is also found in a few stations. The errors between the simulated discharge and observations could be caused by both the WAYS model for runoff simulation as well as the CaMa-Flood model for runoff routing, as the CaMa-Flood model itself also shows different performances in basins across the world (Yamazaki et al., 2011). The relatively low performance of WAYS is found in middle-high latitudes compared with low-middle latitude regions. This result could be explained by the relatively simple snow-melt module in the WAYS model, which thus could consequently produce low-quality runoff for river routing in cold regions. In Australia, only two GRDC stations in the Murray Darling basin are selected for the evaluation, and the correlation coefficient between simulated discharge and GRDC station is less than 0.5, indicating the large difference between them.

Figure S2 shows the histogram of the data points within different intervals of the correlation coefficient. Only in 7.2% of the stations are the correlations between simulation and observation less than 0.5. For more than half the stations, the correlations are higher than 0.7. The results show a generally good correspondence between the simulated and observed discharge. The generally good performance in the discharge simulation confirms the strong capacity of WAYS for runoff generation.

[Figure]

*Figure S2. Histogram showing the percentage of data points within different intervals of the correlation coefficient.*

**#2-3 Were other variables and potentially better datasets considered and rejected for which reasons?**

In addition to ERA-Interim/Land, other reanalysis runoff data are available, such as ERA-Interim, GLDAS, and NECP, among others. However, they show low robustness based on the available research results. For instance, GLDAS v1.0-CLM is found to overestimate runoff globally, and GLDAS v1.0-Noah generated more surface runoff over northern middle-high latitudes (Lv et al., 2018). GLDAS v2.0-Noah showed a significant underestimation trend in exorheic basins (Wang et al., 2016). The snowmelt-runoff peak magnitude simulated by GLDAS v2.1-Noah was found to be excessively high in June and July (Lv et al., 2018). NECP runoff is found to be too high during the winter and too low during the summer in the Mississippi River Basin (Roads and Betts, 2000). ERA-Interim is found to be less close to the observed stream flows compared with ERA-Interim/Land (Balsamo et al., 2015).

The ERA-Interim/Land is well assessed with quality checks by comparison with ground-based observations (GRDC observation) and is widely used as benchmark data (Alfieri et al. 2013; Balsamo et al. 2015; Orth and Seneviratne 2015; Reichle et al. 2017; Wang-Erlandsson et al. 2014). Therefore, it is selected as the reference data for this study for runoff comparison.

**Authors' change in the manuscript.**

**Page 11, Line 20: (the following paragraph is inserted in the end of section "3.3.1 ERA-Interim/Land Runoff Data")**
It should be noted that there are other reanalysis runoff data available, such as ERA-Interim, GLDAS and NECP. However, they show low robustness based on the available research results. For instance, GLDAS v1.0-CLM was found to overestimate runoff globally, and GLDAS v1.0-Noah generated more surface runoff over the northern middle-high latitudes (Lv et al., 2018). GLDAS v2.0-Noah showed a significant underestimation trend in exorheic basins (Wang et al., 2016). The snowmelt-runoff peak magnitude simulated by GLDAS v2.1-Noah was found to be excessively high in June and July (Lv et al., 2018). NECP runoff was found to be too high during the winter and too low during the summer in the Mississippi River Basin (Roads and Betts, 2000). ERA-Interim was found to be less close to the observed stream flows compared with ERA-Interim/Land data (Balsamo et al., 2015).

**#2-4 How come capillary rise is disregarded on the basis of "lack of information at the global scale" is there are other models that take it into account?**

WAYS contains the capillary module, which is adopted from the FLEX model. At the current stage, it is, however, disabled due to the lack of global information on the groundwater table, which could affect the simulated results in this work, e.g., evaporation and RZWS. We decided to disable the capillary module based on our experimental analysis.

We set up two experimental runs for WAYS to check the impact of the capillary module in the current version by switching it on/off. Since there is no observed groundwater table information to constrain the capillary rise amount, switching on the capillary module significantly overestimates the evaporation globally. The global averaged annual evaporation reaches 697 mm/year. Switching off the capillary module reduces the evaporation to 513 mm/year. A merged benchmark synthesis product of evaporation, i.e.,

LandFluxEVAL data, shows only 491 mm/year, which is much closer to the value without the capillary module. Thus, the capillary module is temporary disabled in WAYS until the global information on groundwater table is available. We have mentioned this issue in the revised manuscript. A response to the similar comment can be found in "the response letter to Referee #1" (Comment 6).

In fact, many models ignore the capillary at the global scale due to the absence of groundwater table information (Döll et al., 2003; De Graaf et al., 2015; Hanasaki et al., 2018). Consideration of the capillary in hydrological simulation is more popular in regional studies, mainly due to the local groundwater data availability (Gao et al., 2014; Vergnes et al., 2014).

**Authors' change in the manuscript.**

**Page 25, Line 4: The following paragraph is inserted in the discussion part**
Moreover, the current study does not consider the groundwater access and irrigation mainly due to the lack of global information. The groundwater table information is crucial for capillary rise simulation (Vergnes et al., 2014). Capillary rise simulation without proper water table information could significantly overestimate the evaporation. Thus, the capillary rise flux is ignored in this study. A similar strategy has also been applied by other works due to the absence of the information on the global water table (Döll et al., 2003; De Graaf et al., 2015; Hanasaki et al., 2018). Observations of irrigation on the global scale are also not available (Leng et al., 2015). Although there are simulated irrigation data available on the global scale, the inherent uncertainties could be propagated in our model simulation. Therefore, irrigation is also not considered at this time. However, this neglect could potentially introduce biases into the model simulation in irrigated areas and deep rooted plant-distributed regions, as both irrigation and capillary rise are an additional supply of soil water recharge. The biases may cause an underestimation of evaporation, especially in the dry summertime (Vergnes et al., 2014). This underestimation could consequently affect the simulation of RZWS and runoff because of the interlinkage of these three elements (Rockström et al., 1999). It is found that ignoring the capillary rise could reduce soil water content in the root zone (RZWS), while the runoff will also be reduced (Vergnes et al., 2014). However, these shortcomings can be simply overcome once the global data are available.

**#2-5 Why was Penman-Monteith FAO 56 PM method used (P6L25)? What were the considerations?**

Indeed, many methods are available to estimate potential evapotranspiration (PET) from standard meteorological observations. The Penman-Monteith FAO 56 PM method is recommended by FAO and other studies based on their thorough analysis in PET method intercomparisons (Allen et al., 1998; Lu et al., 2005; Vörösmarty et al., 1998). The Penman-Monteith FAO 56 PM method is based on fundamental physical principles and is found to be the most reliable method for potential evapotranspiration estimation where sufficient meteorological data exist (Chen et al., 2005; Kingston et al., 2009).

We would like to mention that FLEX uses the Hamon method for PET estimation. However, the Hamon method is found to have less robustness in different climatic conditions as well as drawbacks in terms of the daily variability of PET simulation (Bai et al., 2016; Droogers and Allen, 2002). Therefore, we have used the Penman-Monteith FAO 56 PM method in our study.

**Authors' change in the manuscript.**

**Page 7, Line 4: (the changes is marked as blue)**
Potential evapotranspiration is derived by the Hamon equation (Hamon, 1961) in the FLEX model, and it is now replaced by the using the Penman-Monteith FAO 56 PM method (Allen et al., 1998) for the following reason. The Hamon method is found to have less robustness in different climatic conditions as well as drawbacks in the daily variability of the PET simulation due mainly to the relatively simple equation in the Hamon method, as it only employs the average air temperature as an input (Bai et al., 2016; Droogers and Allen, 2002). In contrast, the Penman-Monteith FAO 56 PM method is based on fundamental physical principles and is found to be the most reliable method for potential evapotranspiration estimation when sufficient meteorological data exist (Chen et al., 2005; Kingston et al., 2009). The Penman-Monteith FAO 56 PM method is recommended by FAO and other studies based on thorough analyses of PET method intercomparisons (Allen et al., 1998; Jian biao et al., 2005; Vörösmarty et al., 1998).

**#3 The analyses could be better designed to facilitate understanding of how and why WAYS perform in certain ways, and thus, give more insight into how various components of the model affect the root zone storage and runoff simulation? For example, can the authors show how results are affected by e.g., use of root zone storage capacity derived from uncertain root depth and soil data versus the root zone storage capacity from Wang-Erlandsson et al., 2016? Can the authors perform some sensitivity analyses to highlight model structure and parameter sensitivity?**

Thank you for the comment. We agree with the referee that our study could benefit from a better design of the experiment. To facilitate understanding of how RZSC could affect the model simulation, we have additionally conducted a simulation of WAYS with RZSC derived from an uncertain root depth and soil data. The simulated results are then compared between two runs, i.e., one with RZSC from Wang-Erlandsson (2016) and the other with uncertain RZSC. Since this part could support the conclusion of this paper regarding the importance of correct representation of RZSC in models, we include it in the revised manuscript by adding some figures in the SI.

In addition, we have also performed a sensitivity test to highlight the model structure and parameter sensitivity. Since this part is not directly related to the main conclusion of the manuscript but important to demonstrate the model robustness, we have included it in the SI.

**Authors' change in the manuscript.**

**Page 21, Line 26: (the following paragraphs are added)**

RZSC is a key parameter of the WAYS model. Therefore, it is important to investigate how RZSC could affect the model simulation. In addition to the model simulated with satellite data-derived RZSC products ($S_{R,CHIRPS-CSM}$ and $S_{R,CRU-SM}$), we have additionally conducted WAYS simulations with RZSC derived from uncertain root depth and soil data. The uncertain RZSC ($S_{R,LOOKUP-TABLE}$) is derived based on literature values of root depth and soil texture data (Müller Schmied et al., 2014; Wang-Erlandsson et al., 2016). Due to the global coverage of the RZSC data ($S_{R,CRU-SM}$), only the simulation with $S_{R,CRU-SM}$ is used for comparison. The spatial distribution of the uncertain RZSC is shown in Figure S17, and the differences between $S_{R,CRU-SM}$ and $S_{R,LOOKUP-TABLE}$ are shown in Figure S18. It can be seen that there are large differences between the two RZSC products. The simulation with uncertain RZSC $S_{R,LOOKUP-TABLE}$ shows overestimation globally except for some regions around low-middle latitudes. The latitudinal averaged RZSC further confirms the overestimation of $S_{R,LOOKUP-TABLE}$ at middle-high latitudes (Figure S19).

The large differences between these two RZSC data sets also introduce differences in simulated hydrological elements. Figure S20 shows the impacts of RZSC on the model simulation, including runoff, evaporation and RZWS. A blue color (decrease of RMSE and increase of the ranked correlation) indicates an improvement of the simulated results by replacing the uncertain RZSC ($S_{R,LOOKUP-TABLE}$) with satellite data-derived RZSC ($S_{R,CRU-SM}$), while a red color implies the opposite. For comparison, reference data are used for different variables. For runoff, evaporation and RZWS, the reference data are ERA-Interim/Land (2001-2010, monthly), LandFluxEVAL (1989-2005, monthly) and NDII (2001-2010, 8-days), respectively. Generally, the model simulations are improved by using the RZSC $S_{R,CRU-SM}$. This result emphasizes the importance of an appropriate representation of RZSC in WAYS. A decline of the model performance is also found in some
regions at high latitudes and low latitudes. This result can be partially explained by the inherent uncertainty in the $S_{R,CRU-SM}$ data, as they are derived from other data sets. The RZSC derivation method itself as well as the input data can also introduce biases (Wang-Erlandsson et al., 2016).

[Figure]

**Figure 7.** The improvement of RMSE in evaporation simulations for different land covers by using the satellite data-derived RZSC ($S_{R,CRU-SM}$) instead of the uncertain RZSC ($S_{R,LOOKUP-TABLE}$).

Figure 7 shows the RMSE improvements of simulated monthly evaporation for different land covers obtained by implementing the satellite data-derived RZSC ($S_{R,CRU–SM}$) instead of the uncertain RZSC ($S_{R,LOOKUP–TABLE}$). The analysis reveals that the satellite data-derived RZSC ($S_{R,CRU–SM}$) has great potential to improve the evaporation simulation for all kinds of land covers. The largest improvements are found in broadleaf forests. The improvements in the needleleaf forest, mixed forest and savanna are relatively low. The findings also resonate with another work that used a simple terrestrial evaporation to atmosphere model (STEAM) for evaporation simulation (Wang-Erlandsson et al., 2016).

**The following figures can be found in the SI of this paper.**

[Figure]

*Figure S3. Spatial distribution of uncertain RZSC (SR,LOOKUP-TABLE).*

[Figure]

*Figure S4. The difference between $S_{R,CRU-SM}$ and $S_{R,LOOKUP-TABLE}$ ($S_{R,LOOKUP-TABLE}$ - $S_{R,CRU-SM}$).*

[Figure]

*Figure S5. Latitudinal averaged RZSC of different products.*

[Figure]

*Figure S6. The impacts of RZSC on the model simulation. Blue color indicates the improvement of the simulated results by replacing the uncertain RZSC ($S_{R,LOOKUP-TABLE}$) with satellite data-derived RZSC ($S_{R,CRU-SM}$), while red color implies the opposite. (a) The result for runoff and the reference data for comparison is ERA-Interim/Land data (2001-2010, monthly), (b) the result for evaporation and the reference data for comparison is*

*LandFluxEVAL data (1989-2005, monthly), and (c) the result for RZWS and the reference data is NDII data (2001-2010, 8 days).*

**The following part is put in the SI.**

In the WAYS mode, $\beta$ and $C_e$ are two crucial parameters that control the partitioning of precipitation into evaporation and runoff, thus affecting the water balance. Due to the incredibly high computation cost, only the sensitivity of the model simulation to these two parameters are tested. First, a pixel is selected randomly from the domain to demonstrate the impacts of parameter perturbation on simulated evaporation, runoff and RZWS. For each experiment, only one parameter is perturbed, and the other one is set to the calibrated value. The calibrated value for $\beta$ and $C_e$ is 0.17 and 1.67, respectively. The parameter is perturbed within the range randomly 1000 times during the experiment. Simulations are executed from 2009 to 2010 on a daily scale, while the results are shown on a monthly scale (see Figure S7). The model is more sensitive to parameter $C_e$ than parameter $\beta$. The uncertainties caused by the parameter $C_e$ are generally larger than those caused by the parameter $\beta$, especially for RZWS. These two parameters also have complementary effects on the model simulation, causing larger uncertainties for the simulation than one parameter.

To further investigate the uncertainties stemming from parameters on a global scale, a Monte Carlo simulation of 1000 samples is performed by perturbing the two parameters simultaneously. For both parameters, the normal distribution is used for the Monte Carlo perturbation. Simulations are executed from 2001 to 2010 on a daily scale. The coefficient of variation (CV) for each pixel is then calculated, which reflects the uncertainties (De Graaf et al., 2015). A high value of CV indicates relatively higher uncertainty caused by the parameters, while a low value of CV implies the opposite. Figure S8 shows that parameter-induced uncertainties of evaporation and runoff have similar patterns, while the magnitude is slightly higher for the runoff globally. This finding is consistent with the pixel-based sensitivity test (see Figure S7). The simulated RZWS has the largest uncertainties with the Monte Carlo simulation. Additionally, the uncertainties of RZWS show the opposite trend to the uncertainties of evaporation and runoff. In the northern part of Africa, the Arabian Peninsula, northwest of China and southern part of Australia, the uncertainties in evaporation and runoff are low. However, the uncertainties in RZWS are quite large in these regions.

[Figure]

*Figure S7. Sensitivity of simulated evaporation (top), runoff (middle) and RZWS (bottom) to parameter β and Ce in a randomly selected pixel within the domain. The black solid line represents the simulation based on the calibrated value. The blue area indicates the uncertainties induced from the perturbation of the parameter 1000 times.*

[Figure]

*Figure S8. Coefficient of variation of model-simulated evaporation (top), runoff (middle) and RZWS (bottom) from 1000 Monte Carlo simulations with different parameter settings for β and Ce.*

**#4 The WAYS model is developed based on essential features of the FLEX model (P3L14), and as such I would (1) suggest the authors to present an overview of the similarities and differences between the two and (2) to retain "FLEX" in the model naming (e.g., FLEX-WAYS). Retaining FLEX in the name benefits the model developers that do not need to explain the model roots and will have an easier time communicating the new model developments that builds on an existing well-established mode, and would also be a nice acknowledgement of the earlier FLEX model developments. The practice of name roots**

**exists in the modelling community, and e.g., the models LPJmL and LPJ-GUESS show through their names that they share the same roots.**

Thank you for the comment. To present the similarities and differences between FLEX and WAYS, we have updated Table 1 in the revised manuscript. The last column named "Reference" in Table 1 highlights the sources of equations that are adopted from FLEX or from the literature.

Regarding the model naming, however, we would prefer to keep it as WAYS for the following reasons: (1) WAYS is expected to be further developed by integrating new features, such as a water quality module that allows for environmental impact studies or an economic module to connect the physics of water and virtual water. In this case, WAYS needs its own postfix to identify different features, e.g., WAYS-WQ or WAYS-ECO. A prefix of FLEX will make the model name too complicated. (2) The FLEX model itself actually has different branches, e.g., FLEX-Topo indicates a topography-driven FLEX model, and FLEX$^D$ represents a semidistributed FLEX model while FLEX$^{TO}$ stands for FLEX-Topo without constraints (Gao et al., 2014). All the different types of FELX have the same equations for all hydrological processes but with different model structures during the application. WAYS has replaced many equations from FLEX to enhance the capacity of the model for global simulation. A prefix of FLEX could cause confusion as the FLEX branches seldom change the equations. (3) The application of name roots to models is a good strategy for the models that share the core structure and equations but with different features as added functions. For instance, the VIC model is developed based on a small-scale distributed model Xinanjiang (Zhao, 1992). It has its own name and has been further developed by including additional features, e.g., VIC-CropSyst-v2, that simulate the nexus of climate, hydrology, cropping systems, and human decisions.

**Authors' change in the manuscript.**

**Page 8: Table 1 is updated. (the changes are marked as blue)**

**Table 1.** Water balance and constitutive equations used in WAYS

| Reservoirs | Water balance equations | | Constitutive equations | | Reference |
|---|---|---|---|---|---|
| | | | $P_{tf} = max(0, P_r - (S_{i,max} - S_i))$ | (2) | - |
| Interception reservoir | $\dfrac{dS_i}{dt} = P_r - E_i - P_{tf}$ | (1) | $E_i = E_p \left(\dfrac{S_i}{S_{i,max}}\right)^{2/3}$ | (3) | Deardorff (1978) |
| | | | $S_{i,max} = m_c L$ | (4) | Wang-Erlandsson et al. (2014) |
| Snow reservoir | $\dfrac{dS_w}{dt} = \begin{cases} -M & \text{if } T > T_t \\ P_s & \text{if } T \leq T_t \end{cases}$ | (5) | $M = \begin{cases} min(S_w, F_{DD}(T - T_t)) & \text{if } T > T_t \\ 0 & \text{if } T \leq T_t \end{cases}$ | (6) | Rango and Martinec (1995) |
| | | | $P_e = P_{tf} + M$ | (8) | - |
| Root zone reservoir | $\dfrac{dS_{rz}}{dt} = P_e - R - E_a$ | (7) | $\dfrac{R}{P_e} = 1 - \left(1 - \dfrac{S_{rz}}{(1+\beta)S_{rz,max}}\right)^{\beta}$ | (9) | Sriwongsitanon et al. (2016) |
| | | | $E_a = (E_0 - E_i) \cdot min\left(1, \dfrac{S_{rz}}{C_e S_{rz,max}(1+\beta)}\right)$ | (10) | Sriwongsitanon et al. (2016) |
| Slow response reservoir | $\dfrac{dS_s}{dt} = R_s - Q_s$ | (11) | $R_s = min(f_s R, R_{s,max})$ | (12) | Döll and Fiedler (2008) |
| | | | $Q_s = S_s / K_s$ | (13) | Döll et al. (2003) |
| Fast response reservoir | $\dfrac{dS_f}{dt} = R_f - Q_{ff} - Q_f$ | (14) | $R_f = R - R_s$ | (15) | - |
| | | | $Q_{ff} = max(0, S_f - S_{ftr})/K_{ff}$ | (16) | - |
| | | | $Q_f = S_f / K_f$ | (17) | - |

Note: all the time scale-dependent parameters need to be divided by $\Delta t$ to make the equations dimensionally correct and suitable for any other time scales.

- in the reference column indicates that the formula is taken from the FLEX model.

**Page 26, Line 9: (the changes are marked as blue)**

This study was supported by the National Natural Science Foundation of China (Grant No. 41625001), the Strategic Priority Research Program of the Chinese Academy of Sciences (Grant No. XDA20060402) and the National Natural Science Foundation of China (Grant No. 41571022). We would like to acknowledge the authors of the FLEX model for their great help during the development of WAYS.

**#5 The WAYS performance evaluation in terms of root zone storage moisture is highly dependent on the comparison with NDII, which weakens the conclusions, since also further work is still needed to robustly establish the relationship between NDII and soil moisture at the global scale. It is after all only recently suggested by Sriwongsitanon et al. (2016) – a study in a river basin in Thailand – that NDII can have the potential to be used as a proxy for catchment scale root zone storage capacity. The authors could potentially strengthen their conclusions by evaluating model simulation outputs with additional sources of data/methods, such as FLUX-tower, evaporation, EVI etc. Summarizing evaluation figures can be shown in the main manuscript, and others could be included in Supplementary Information. A more detailed list of the equations and calibration process could also be included in Supplementary Information for transparency.**

Thank you for the comment. Based on the referee's suggestion, we have performed an additional evaluation on our model simulation to strengthen the conclusions. Since RZWS has close links to the total evaporation, we have compared the WAYS simulated evaporation to FLUX-tower observations (FLUXNET2015) as well as a merged benchmark synthesis product of evaporation (LandFluxEVAL).

The referee also suggested that we compare our model simulation to EVI. However, as we stated in our manuscript (page 2, line 22), EVI as well as NDVI are the most widely used vegetation indices, which have strong links to root zone soil moisture but cannot reflect the dynamics of the water content in the root zone layer (Santos et al. 2014). However, NDII determines the water stress of plants in the root zone by taking advantage of the property of shortwave infrared reflectance, thus possessing the intuitive advantage of reflecting the dynamics of RZWS than EVI (Sriwongsitanon et al. 2016). As we have already compared our model results to NDII, we prefer to skip the comparison between our model simulation with EVI.

Since this evaporation evaluation can support the conclusion of this paper regarding the capacity for hydrological cycle simulation, we include the work of evaporation evaluation against FLUX-tower data in the main text of the revised manuscript and the rest in SI. Please refer to the changes in the revised manuscript.

Regarding "A more detailed list of the equations and calibration process could also be included in Supplementary Information for transparency.", we have further updated the table with model equations by adding necessary references to each equation and one more table to illustrate the model parameters as well as the parameter ranges. Since the model equations and parameters are important to a study on model development, we have retained these two tables in the main text of the manuscript. In addition, the calibrated parameters will be uploaded as supplements in terms of netCDF files.

**Authors' change in the manuscript.**

**Page 20, Line 26: The following paragraphs are inserted**
RZWS has a close link to the total evaporation, as RZWS represents the available water that plants can use. In this section, the performance of WAYS in evaporation simulation is evaluated against the FLUXNET2015 data. FLUXNET2015 is a global network of micrometeorological flux measurement sites that measure the exchange of $CO_2$, water vapor and energy between the biosphere and the atmosphere (Pastorello et al., 2017). The tower-measured latent heat flux (LF, $W/m^2$) is converted to ET (mm/day) using the proportionality parameter between energy and depth units of ET (Velpuri et al., 2013) as follows:

$$ET = \frac{LE}{\lambda}$$

The results are shown in Figure S15. The background is the annual averaged evaporation from WAYS for the period 1971-2010. The points indicate the comparison results between the flux tower and WAYS simulation. The locations of the points indicate the locations of the flux towers, and the colors indicate the correlation coefficient. WAYS is found to have relatively better performance in America, Europe and China than in Africa and Australia.

However, a few stations near the boundary of America and Europe also show weak correlations between the simulations and flux tower data.

Figure S16 shows the percentage of data points within different intervals of the correlation coefficient. The calculated correlation coefficient is crowded in the interval of 0.6-0.8, while more than half of the stations (56%) show a correlation coefficient of more than 0.6. The relatively poor performance of the model in some regions could be partially explained by the following reason. FLUXNET2015 corresponds to point-based observation data, while WAYS simulates the evaporation on grid cells with a 0.5 degree spatial resolution. For the comparison, the model simulation in a certain pixel is selected based on the distance between the flux tower and the center of the pixel. The model simulation actually represents an averaged value for a 0.5 x 0.5-degree pixel. This averaging will inherently introduce errors when comparing the simulation to station-based data. Similar results are also found in other studies comparing FLUXNET2015 data to either model simulations or remote sensing-derived evaporations (Lorenz et al., 2014; Velpuri et al., 2013).

[Figure]

**Figure 6.** Averaged monthly evaporation of WAYS simulation (WAYS_CRU) against the FLUXNET data.

Furthermore, the average monthly evaporation is compared to the FLUXNET2015 data at each flux tower, and the results are shown in Figure 6. Good correspondence between the model simulation and flux tower data can be found by visual inspection. The points with a higher correlation coefficient show a better relationship between the model simulation and flux tower observation and are distributed closer to the diagonal. The evaluation results confirm the generally good performance of WAYS in monthly evaporation simulation. The detailed results on evaporation evaluation against FLUXNET2015 are provided in the SI as Excel files. In addition, an evaluation of the evaporation simulation is further conducted

against LandFluxEVAL, a merged benchmark synthesis product of evaporation at the global scale (Mueller et al., 2013). The results can be found in the SI.

**The following part is put in the SI.**

The model simulation is further compared to a gridded data set, LandFluxEVAL data, for evaporation evaluation. The LandFluxEVAL data are a merged benchmark synthesis product of evaporation on a global scale and a combination of land-surface model simulations, remote sensing products, reanalysis data and ground observation data (Mueller et al., 2013). The LandFluxEVAL data are used in many studies as reference data for evaporation evaluations (Lorenz et al., 2014; Martens et al., 2017; Wartenburger et al., 2018). Since the LandFluxEVAL data are only available at 1-degree spatial resolution, the WAYS simulated evaporation is aggregated to 1 degree to match the resolution of the reference data. The evaluation is executed for 1989-2005 based on the availability of the LandFluxEVAL data. For the spatial evaluation, the WAYS simulation based on RZSC ($S_{R,CRU-SM}$) is used due to the global coverage of the RZSC product. For latitudinal comparison, both runs of WAYS simulated evaporation are used.

A promising relationship between WAYS simulated evaporation and LandFluxEVAL evaporation is found both in spatial pattern and in latitudinal average (see Figure S9). The generally high correlation coefficient (Figure S9, a) confirms the good performance of the WAYS model. However, relatively poor performance is also found in some regions in Europe, North America and South America (Amazon basin). It can also be seen that the spatial pattern of WAYS simulated annual averaged evaporation follows that of LandFluxEVAL data, while overestimations are found in regions, e.g., the Amazon basin and southeast Asia. The latitudinal evaluation shows that both WAYS simulations (WAYS_CRU and WAYS_CHIRPS) display a slight overestimation.

[Figure]

*Figure S9. Validation results of the evaporation of WAYS simulation against the LandFluxEVAL data (1989-2005). (a) The calculated correlation coefficient between LandFluxEVAL data and WAYS simulation, (b) the annual averaged evaporation of LandFluxEVAL data, (c) the annual averaged evaporation of WAYS simulation based on RZSC $S_{R,CRU-SM}$, and (d) the comparison of the averaged latitudinal evaporation for WAYS model runs as well as the LandFluxEVAL data.*

**#6 Wang-Erlandsson et al., 2016 found that normalizing the root zone storage capacity using the Gumbel distribution by land cover type further improves performance, and recommended the use of Gumbel distribution. Please consider applying the Gumbel normalization to the root zone storage capacity data.**

We incorporate the suggestion in the revised manuscript. We have now updated the RZSC data based on the Gumbel normalization. The optimized RZSC is calculated based on the suggested return period for each land cover by Wang-Erlandsson et al. (2016). Figure shows the flow chart of updating the RZSC data based on the Gumbel normalization with a different optimized return period. Since RZSC is a key parameter in the WAYS model, the model is recalibrated, and the simulations are also updated accordingly due to the change of RZSC.

[Figure]

*Figure 1. Flow chart of updating RZSC based on the Gumbel normalization*

**Authors' change in the manuscript.**

**Page 10, Line 15: (the changes are marked as blue)**
Since Wang-Erlandsson et al. (2016) suggested that a Gumbel normalization of RZSC by land cover types with different return periods could further improve the model performance, we have accordingly adjusted the RZSC in this study. The two selected global root zone storage capacity products are shown in Figure S13, and their mean latitudinal values are shown in Figure S14.

**#7 Please consider discussing how and where the results might be influenced by groundwater access and irrigation, noting that the root zone storage capacity in Wang-Erlandsson et al., 2016 was adjusted for irrigation but not access to groundwater, while WAYS do not account for either groundwater or irrigation.**

Thank you for the comment. Indeed, the WAYS model does not consider the groundwater access and irrigation at the current stage. Although the capillary module is included in

WAYS, it is currently disabled due to the lack of information on the global groundwater table. The same strategy is also applied in other works, especially for hydrological simulation on the global scale (Döll et al., 2003; De Graaf et al., 2015; Hanasaki et al., 2018). Based on the experimental test on the capillary module, activation of the capillary module without groundwater information for capillary flux constrain is found to significantly increase the simulated evaporation. Nevertheless, ignorance of the capillary rise and irrigation are shortcomings of our study. This issue is now discussed in the revised manuscript.

**Authors' change in the manuscript.**

**Page 25, Line 4: The following paragraph is inserted in the discussion part**
Moreover, the current study does not consider the groundwater access and irrigation mainly due to the lack of global information. The groundwater table information is crucial for capillary rise simulation (Vergnes et al., 2014). Capillary rise simulation without proper water table information could significantly overestimate the evaporation. Thus, the capillary rise flux is ignored in this study. A similar strategy has also been applied by other works due to the absence of the information on the global water table (Döll et al., 2003; De Graaf et al., 2015; Hanasaki et al., 2018). Observations of irrigation on the global scale are also not available (Leng et al., 2015). Although there are simulated irrigation data available on the global scale, the inherent uncertainties could be propagated in our model simulation. Therefore, irrigation is also not considered at this time. However, this neglect could potentially introduce biases into the model simulation in irrigated areas and deep rooted plant-distributed regions, as both irrigation and capillary rise are an additional supply of soil water recharge. The biases may cause an underestimation of evaporation, especially in the dry summertime (Vergnes et al., 2014). This underestimation could consequently affect the simulation of RZWS and runoff because of the interlinkage of these three elements (Rockström et al., 1999). It is found that ignoring the capillary rise could reduce soil water content in the root zone (RZWS), while the runoff will also be reduced (Vergnes et al., 2014). However, these shortcomings can be simply overcome once the global data are available.

**#8 Please provide the source code, and not only by request.**

Thank you for the suggestion. We will make the code of the WAYS model, including all the parameters, freely available after the manuscript is accepted.

**Specific comments**

1.      **P1L10: state what was used for evaluating root zone storage (i.e., NDII) in the abstract.**

Thank you. The reference data for RZWS evaluation is stated in the abstract.

**Authors' change in the manuscript.**
**Page 1, Line 10: (the changes is marked as blue)**

The results show the ability of the model to mimic RZWS dynamics in most of the regions through comparison with proxy data, the Normalized Difference Infrared Index (NDII).

2. **P1L10: "many applications": please provide concrete examples.**

Thank you. It is addressed accordingly.

Authors' change in the manuscript.
Page 1, Line 11: (the changes is marked as blue)
Compared to existing hydrological models, WAYS's ability to resolve the field-scale spatial heterogeneity of RZSC and simulate RZWS may offer benefits for many applications, e.g., agriculture and land-vegetation-climate interaction investigations.

3. **P1L11: "attention needs to also. . .": hardly the most important limitation, please consider rather listing the more pressing future model developments needs and emphasize the key contribution of this model in comparison to other existing global hydrological models.**

Thank you. We have incorporated the suggestion of the referee and have emphasized the key contribution of this model in comparison to other existing global hydrological models. We have further listed the pressing future research needs in the abstract accordingly.

Authors' change in the manuscript.
Page 1, Line 11: (the changes is marked as blue)
Compared to existing hydrological models, WAYS's ability to resolve the field-scale spatial heterogeneity of RZSC and simulate RZWS may offer benefits for many applications, e.g., agriculture and land-vegetation-climate interaction investigations. However, the results from this study suggest an additional evaluation of RZWS is required for the regions where the NDII might not be the correct proxy.

4. **Please point out that Sriwongsitanon et al. (2016) is a study in a river basin in Thailand and not a global study.**

Thank you. It is addressed accordingly.

Authors' change in the manuscript.
Page 2, Line 26: (the changes is marked as blue)
Recently, Sriwongsitanon et al. (2016) investigated the relation between root zone water storage and the Normalized Difference Infrared Index and found a promising correspondence between them in a river basin in Thailand, especially in the dry seasons, where water stress exists. However, a global scale study has been absent in the literature.

5. **P6L27 "no information is available at the global scale": Please consider including a few more lines describing the issues related to capillary rise modelling in global scale models and include related references, such as (Vergnes, Decharme, and Habets 2014) and references within.**

Thank you. It is addressed accordingly.

**Authors' change in the manuscript.**
To avoid repetition, please to see the changes in response to comment 7

6.   **P8L28, "it has been well-justified (de Boer-Euser et al., 2019)": please consider specifying what is justified and add other relevant sources, e.g. "the method has been shown to increase model performance at both basin and global scale (e.g., de Boer-Euser et al., 2016, 2019, Gao et al. 2014, Wang-Erlandsson et al., 2016, Nijzink et al., 2016)".**

Thank you. It is addressed accordingly.

**Authors' change in the manuscript.**

**Page 9, Line 28: (The changes are marked in blue)**
This method has been well justified (de Boer-Euser et al., 2019) and overcomes the shortcomings of the traditional methods (look-up table approach; field observation-based approach) at the global scale, such as data scarcity, location bias, and risks of unlikely vegetation and soil combinations due to data uncertainty (Feddes et al., 2001). The method has been shown to increase the model performance at both the basin and global scales (Gao et al., 2014b; Nijzink et al., 2016; Wang-Erlandsson et al., 2016). Moreover, it has been proven to be able to produce plausible root zone storage capacity in boreal regions by investigating the relationship between RZSC and numerous environmental factors, including climate variables, vegetation characteristics, and catchment characteristics (de Boer-Euser et al., 2019).

7.   **P14L11, "reported in his work": please change to "reported in their work".**

Thank you. It is corrected.

8.   **P22L6 "DNII", should be NDII.**

Thank you. It is corrected.

**References**
Vergnes, J.-P., B. Decharme, and F. Habets. 2014. "Introduction of Groundwater Capillary Rises Using Subgrid Spatial Variability of Topography into the ISBA Land Surface Model." Journal of Geophysical Research: Atmospheres 119(19): 11,065-11,086.

Cited in the revised manuscript.

**All the references are included in the manuscript.**

---

## Author Comment (AC4) · 26 Jul 2019

We would like to thank William Chris for his interest in this topic and for the comments to improve our manuscript. Based on the comments some calculations have been performed. Our point-by-point response to the comments is given in the following (**Comments in black**, Answers in blue and the content related to the changes in the revised manuscript are marked in orange.):

This study tries to develop a 'global' hydrological model. The authors are lack of good understanding on hydrological processes, and the methodology they used are not appropriate at all. The authors overclaimed their contribution. The manuscript is poorly written. Some of the figures are not clear. The current manuscript cannot be accepted, and should be returned to the authors to make it a better work.

Thank you for the comment. In this work, we have extended a widely used lumped model, FLEX, into a distributed model that can be used on a global scale. In addition, a climatederived root zone storage capacity (RZSC) is integrated into the developed model WAYS to capture the spatial heterogeneity of the rooting systems. We demonstrate the benefit of a climate-derived RZSC to the hydrological model for simulation, especially the capacity of root zone water storage (RZWS) simulation. Thus, we believe the methodology we have used is appropriate and that the hydrological processes conceptualized in WAYS are proper. Based on the comments from the three referees as well as from the short comments, we have further improved the manuscript text as well as the figures and tables.

My comments are as below: 1. The methodology used are not appropriate at all. The authors compared their runoff simulation against some global model simulation and composite runoff data. We all know the global runoff simulation/composite runoff data are designed for global studies, and can have very large uncertainty on each river basin. They cannot use these data to verify their simulation, especially for a study aiming to develop a 'new model'. Thus, the comparison between the authors' simulation and other runoff data that the authors used means nothing: the authors cannot claim their model is good. The authors should compare their simulation against hydrological gauge observation which is not difficult to collect at all. I doubt the authors' results. They may choose to avoid the comparison against hydrological gauge observation purposely because their model is suffering fatal flaws. For a paper developing a hydrological model, comparison against in situ gauge observation is extremely important. The authors should not skip this step. In addition, the authors compare their runoff simulation after calibrating their model, whereas the other models in ISIMIP are not calibrated. Thus, the comparisons are useless because the models in ISIMIP have large uncertainty which have already been unrevealed in several recent studies by the ISIMIP group (perhaps the authors missed these very important publications).

Thank you. In fact, the simulated runoff of WAYS is first compared to the reference data ERA-Interim/Land runoff. The performance of WAYS in runoff simulation is evaluated based on the comparison between ERA-Interim/Land data and WAYS simulation.

Since WAYS uses the same driving data as the ISIMIP2a models and the ISIMIP2a simulations are widely discussed in many studies, an additional comparison between WAYS and the

ISIMIP2a models can provide added-value for evaluating our model. Therefore, the ISIMIP2a simulations are also shown in the results section together with the ERA-Interim/Land data. We do mention the purpose of inclusion of the ISIMIP2a simulations in the results (page 13, line 15: "Since WAYS uses the same driving data as the ISIMIP2a models and the ISIMIP2a simulations have been widely discussed in many studies (Schewe et al., 2014; Müller Schmied et al., 2016; Gernaat et al., 2017; Zaherpour et al., 2018), we also perform a comparison between WAYS and the ISIMIP2a models to further evaluate our model."). However, this is not mentioned in the validation strategy section of the manuscript. We have now further clarified this issue in the revised manuscript (please see "Authors' change in the manuscript.").

Indeed, the ISIMIP2a models are not calibrated. We have mentioned this issue in the manuscript (page 13, line 30: This result occurs partly because some of the ISIMIP2a models are not calibrated at all (Zaherpour et al., 2018), whereas WAYS is calibrated to a Composite Monthly Runoff data set that assimilates the monitored river discharge (Fekete et al., 2011).). We have revised the captions of the related figures (Figures 2 and 3 in the revised manuscript) to note this issue.

To enhance the validation part of this manuscript, we have additionally evaluated our results with observed discharge from the Global Runoff Data Centre (GRDC). Since WAYS does not currently have a native runoff routing module, a third-part runoff routing tool, CaMa-flood, is applied to route the WAYS simulated runoff (Yamazaki et al. 2011). Given that the manuscript is already quite extensive, the discharge comparison is not a direct evaluation of WAYS but of both WAYS and CaMa-Flood. This information has been added in Supplementary Information (SI). Since this comment shares the similar opinion with Referee #3 (comment 2), we would like to refer to the responses to the comments of referee #3 to avoid repetition, as the response is long. The corresponding revision to the manuscript can also be found there.

Authors' change in the manuscript.

**Page 11, Line 5: (the changes is marked as blue)**

In this study, the ERA-Interim/Land runoff data are used for validation of the runoff simulation, and the Normalized Difference Infrared Index (NDII) is used for the validation of the WAYS model for root zone water storage simulation. Considering the time period of coverage of both data sets (ERA-Interim/Land: 1979-2010, NDII: 2000-present) and the study period (1971-2010) of this work, the period 2001-2010 is selected as the validation period. For runoff evaluation, ISIMIP2a simulations are also included, as they use the same climate forcing as our study in the same period. The purpose of inclusion of the ISIMIP2a simulations for comparison can be found in the model evaluation section (see Section 4).

2. The authors are lack of basic knowledge about remote sensing. The root zone can be more than 10 meters in depth. The sensors used in the NDII studies cannot penetrate the earth ground up to 10 meters, and even one or two meters are suffering large uncertainty because the attenuations of signals with increase in depth. This is why the most state-ofthe-art soil moisture products just provide data in the surface 5/10 cm. Thus, the comparison against NDII based data is not appropriate at all. Because this paper is to develop a model, the authors should use in situ observation which is not difficult to collect. I don't understand why the authors choose to skip the comparison against gauge observation.

Thank you for the comment. As we stated in the manuscript (page 2, line 8 in the revised manuscript), remote sensing itself can only detect the soil water in the surface layer. However, NDII is not a direct observation from the satellite but a Normalized Difference Index, similar to NDWI, NDVI, and so on. It is calculated based on infrared reflectance (NIR) and shortwave infrared reflectance, and it reflects the water stress in the root zone layer and, thus, can be used as proxy data for RZWS rather than RZWS itself. The NDII-related information is interpreted in detail in Section 3.3.2 in the manuscript.

We do not compare the situ observations because there are no observations available for RZWS (Sriwongsitanon et al., 2016). The observation you mentioned is probably the soil moisture at a certain depth, which differs from RZWS.

3. The authors do not have a good understanding on hydrological processes. i). Vegetation plays a vital role in runoff variations especially in densely vegetated regions (e.g., the Amazon, Congon and some regions in the Yangtze, Mekong, Ganges, Mississippi Rivers et.al.) through the transpiration processes. At the leaf and canopy scales, the mechanisms of transpiration are also different. LAI, fPAR, CO2, wind, solar radiation, stomatal conductance all are influencing transpiration. The authors did not consider the stomatal influence at all (as shown in the Figure 1 and Table 1). Without comprehensively considering the transpiration processes, how the model developed can predict water resources availability, especially many recent studies have unravelled that the earth is greening and CO2 concentration is increasing. Thus, the model developed by the authors has fatal flaws, and this paper cannot be accepted.

Thank you for the comment. We agree with the reviewer that vegetation plays a vital role in runoff variations, especially in densely vegetated regions, and the mechanisms of transpiration are also different at the leaf and canopy scales. However, the model we developed in this study is a conceptual hydrological model with a conceptualized structure to mimic the hydrological cycle. This design differs from land surface models, dynamic vegetation models or physically based hydrological models, which could have more functions with physical meanings (Bierkens, 2015). The conceptual hydrological model, however, has its own advantages in practicability and computation efficiency (Devia et al., 2015). The transpiration is of course considered by conceptual models, while some of them calculated the total evaporation without separating the evaporation into different fluxes. Moreover, conceptual models are widely applied for water related applications, e.g., runoff simulation and water scarcity analysis, especially on a global scale (Döll et al., 2003; Döll and Fiedler, 2008; Hanasaki et al., 2008; Wang-Erlandsson et al., 2014). Thus, a well-developed conceptual model, such as WAYS, should be proper for predicting water resource availability.

In addition, the continuous greening of the earth as well as the increased  $CO_2$  concentration are indeed important issues. However, they are beyond the scope of this study as they are more related to climate change analysis.

ii). The infiltration capacity of soil plays an important role in controlling the volume of surface runoff and subsurface runoff, and also influences root zone water storage. The infiltration capacity of soil is related to soil type, and has clear physical meaning. The authors considered the infiltration as shown in the Figure 1. However, the authors did not report how they determine this important parameter value. If the authors used the values related each soil type, they did not report which soil map distribution data and which hydraulic property datasets of the soil types are used. If the authors calibrated the parameter values, the authors should be aware of that if it is appropriate to calibrate because the results may be wrong after calibrating some parameters with clear physical meaning. The authors are afraid of reporting the calibrated parameter values and the parameter ranges used in the calibration. The authors stated they calibrated their model for good runoff simulation. I am afraid that they calibrated their model for good runoff simulation with the cost of losing the physical meaning of important parameters. Perhaps the authors choose to not show the important information purposely in order to get their paper published. No, absolutely no. The authors have to show which parameters are calibrated, the parameter ranges used in calibration and calibrated parameter values.

Thank you for the comment. The precipitation partitioning function in the WAYS model is based on a widely used beta function of the Xinanjiang model (Zhao, 1992). It is a conceptualized runoff generation function that consists of empirical parameters. The model is a conceptual model and is different from the physically based model. The model parameter must be calibrated before simulation. Indeed, the physically based model is usually run without calibration as the parameters it uses have corresponding physical meanings. However, the physical model and conceptual model are two different methods without any conflicts between each other.

Moreover, we have added a table to illustrate the parameters used as well as the parameter ranges if calibration is needed. Please refer to the changes in the revised manuscript (page 11, Table 2). We will share the calibrated parameters together with the code for the model after the paper is accepted for publication.

Since the calibrated parameters are spatially distributed and are not appropriate to show in tables, we provide the spatial patterns of two key parameters ( $\beta$ ,  $C_e$ ) that are calibrated, as these two parameters mostly affect the partitioning of precipitation (see Figure S21 and Figure S22). The rest of the calibrated parameters are uploaded to the response thread in a netCDF file as a supplementary document.

---

## Author Comment (AC5) · 26 Jul 2019

We would like to thank Astrid Kerkweg, the Executive editor of GMD, for his constructive comments on our manuscript regarding to the basic requirements of papers submitted in GMD. Our point-by-point response to the comments is given in the following (**Comments in black**, Answers in blue and the content related to the changes in the revised manuscript are marked in orange.):

**Dear authors,**
**in my role as Executive editor of GMD, I would like to bring to your attention our Editorial version 1.1:**
**http://www.geosci-model-dev.net/8/3487/2015/gmd-8-3487-2015.html**
**This highlights some requirements of papers published in GMD, which is also available on the GMD website in the 'Manuscript Types' section:**
**http://www.geoscientific-model-development.net/submission/manuscript_types.html**
**In particular, please note that for your paper, the following requirements have not been met in the Discussions paper:**

Thank you for the notes. We have carefully read the information of Editorial version 1.1. We will follow the requirements of papers published in GMD.

**1 "The main paper must give the model name and version number (or other unique identifier) in the title."// Accordingly, add the acronym WAYS including a version number to the title.**

Thank you. We have changed the title accordingly.

**Authors' change in the manuscript.**
The title of the manuscript is changed from
"A hydrological model for root zone water storage simulation on a global scale"
to "WAYS v1: A hydrological model for root zone water storage simulation on a global scale"

**2 GMD is encouraging authors to upload the program code of models (including relevant data sets) as a supplement or make the code and data of the exact model version described in the paper accessible through a DOI (digital object identifier). In case your institution does not provide the possibility to make electronic data accessible through a DOI you may consider other providers (eg. zenodo.org of CERN) to create a DOI. Please note that in the code accessibility section you can still point the reader how to obtain the newest version.**

**If for some reason the code and/or data cannot be made available in this form (e.g. only via e-mail contact) the "Code Availability" section need to clearly state the reasons for why access is restricted (e.g. licensing reasons). Consequently, you need to provide a reason in the code availability section, why the code can not be made publicly available. Without a proper reason, it is not acceptable that the code is not made public available before the final publication of the article.**

Thank you for the comment, we will make the code including the parameters public accessible.

---

## Author Comment (AC6) · 26 Jul 2019

We would like to thank H.M. Jones for his interest in this topic and for the comments to improve our manuscript. Based on the comments some calculations have been performed. Our point-by-point response to the comments is given in the following (**Comments in black**, Answers in blue and the content related to the changes in the revised manuscript are marked in orange.):

**1. After reading the manuscript, I do not think the results can support the research objective. First, the WAYS is calibrated and the runoff simulation is compared with others after that. Therefore, the better performance of WAYS (let's assume it is better first, and actually I do not think so) could be due to the calibration, not because of the consideration of root zone water storage changes. Second, the NDII data were used as a surrogate of root zone water storage changes. The NDII is just suggested in a river basin in Thailand. However, it is still not clear if it is appropriate to do so on large scales with different climate and hydrological regimes. Therefore, the simulated root zone water storage is not actually verified.**

Thank you. In fact, the simulated runoff of WAYS is first compared to the reference data ERA-Interim/Land runoff. The performance of WAYS in runoff simulation is evaluated based on the comparison between ERA-Interim/Land data and WAYS simulation. Since WAYS uses the same driving data as the ISIMIP2a models and the ISIMIP2a simulations are widely discussed in many studies, an additional comparison between WAYS and the ISIMIP2a models can provide added-value for evaluating our model. Therefore, the ISIMIP2a simulations are also shown in the results section together with the ERA-Interim/Land data.

Moreover, to investigate the impacts of RZSC on model simulation, we have additionally conducted the simulation of WAYS with the root zone storage capacity derived from uncertain root depth and soil data. The simulated results are then compared with the two runs, i.e., one with RZSC from Wang-Erlandsson (2016) and the other with uncertain RZSC. The uncertain RZSC ($S_{R,UNCERTAIN}$) is derived based on the literature values of root depth and soil texture data (Müller Schmied et al. 2014; Wang-Erlandsson et al. 2016). For details of related changes, we refer to the response to the comments of Referee #2 (comment 5) to avoid repetition, as the response is long. The corresponding revision in the manuscript can also be found there.

We agree with the reviewer that the NDII is just investigated in a river basin in Thailand, thus the evaluation could weaken the conclusions. Thus, additional evaluations covering large areas have been conducted to further strengthen conclusions. Since this comment shares the similar opinion with Referee #3 (comment 5), we would like to refer to the responses to the comments of referee #3 to avoid repetition, as the response is long. The corresponding revision to the manuscript can also be found there.

**2. I also feel it is not rigorous that without in situ hydrological gauge and flux tower data to verify simulations of ET, runoff, soil moisture etc. Comparison with ISIMIP model runoff simulation is not convincing. The spatial distribution of simulation is also important. Please show the spatial distribution.**

Thank you. In the revised manuscript, we have substantially improved the model evaluation by conducting additional simulations, validations and comparisons, following the comments by reviewers and the short comments. In brief, we have compared the discharge with GRDC observations and evaluated evaporation simulation against FLUXNET2015 and LandFluxEVAL data. Please refer to our responses to comments 2 and 5 of Reviewer 3 for details.

Regarding the comment on comparing our model simulation with soil moisture, we have stated in the manuscript (page 2, line 16: However, all these studies estimated the root zone soil moisture up to a certain depth, e.g., 100 cm, thus still retaining drawbacks to the accurate calculation of the water stored in the entire root zone layer. Since the root depth is location-dependent and could reach a depth of more than 30 meters (Fan et al. 2017).). The soil moisture in the top 100 cm cannot reflect the water content in the entire root zone layer. Evaluating the model performance against soil moisture would not necessarily contribute to improving the model robustness. Therefore, we did not verify the model with the soil moisture.

**3. As shown in Figure 1 and the manuscript descriptions, the soil layer is separated into vadose zone (includes the root zone) and saturated zone, which is similar to many existing models. In addition, in this manuscript, the NDII is not justified to represent root zone water storage changes on large scales with different climate and hydrological regimes. Therefore, the novelty of this manuscript is not enough.**

Thank you. We agree with the reviewer that NDII is not justified to represent root zone water storage changes on large scales, which could weaken the conclusions. Referee #3 shares a similar opinion in his comments. We have addressed this issue in detail in the response to the comments of Referee #3 (comment 5). The corresponding revision in the manuscript can also be found there.

**4. Because the soil is separated into different zones, at every grid, each zone must have a certain depth (or a percentage value) at a moment and the depth or percentage will change with rainfall-runoff processes. The manuscript failed to report the changes of the depth of each zone or what percentage of soil is saturated/unsaturated at different time. Please also show the spatial distribution. This is important to see if the simulation is reasonable.**

Thank you. One of the motivations of our work is to simulate root zone water storage on a global scale, without constraining the quantification to a certain depth. In fact, RZWS is simulated in terms of the water equivalent depth. Based on the comment, we have analyzed the spatial pattern of the simulated RZWS as well as the dynamics of RZWS in different latitudes in different months. Due to the length of the manuscript, this information has been included in Supplementary Information (SI).

**The following part is put in the SI.**
Figure S1 shows the spatial distribution of the annual averaged RZWS simulated by WAYS in the 1971-2010 period. It shows that RZWS is high in low-middle latitudes, while RZWS in

middle-high regions is relatively low. RZWS represents the water content that is stored in the root zone as well as the available water for plants. Therefore, for lower or middle latitudes, the available water for plants is relative higher than for high-latitude areas.

To further investigate the soil water condition, we have calculated the root zone soil wetness by dividing RZWS by RZSC, and the results are shown in Figure S2. The root zone soil wetness follows the spatial patterns of RZWS in general. However, differences can also be found in regions such as Europe, South America and the eastern part of North America.

[Figure]

*Figure S1. The spatial distribution of the annual averaged RZWS simulated by WAYS in the 1971-2010 period.*

[Figure]

*Figure S2. The spatial distribution of the annual averaged root zone soil wetness in the 1971-2010 period.*

The simulated RZWS is shown in Figure S3 to reveal dynamics in different latitudes and in different months. The latitudinal averaged RZWS again confirms that the RZWS are relatively plentiful compared with the high latitudes. However, a decreasing trend can also be found by moving from a low latitude to the equator. Two simulations of the WAYS model show similar fluctuation along the latitudes, while the simulation with $S_{R,CHIRPS}$ is slightly higher. Figure S3 shows that RZWS in low-middle latitudes has a larger monthly variation than in other regions, while the Northern Hemisphere and Southern Hemisphere show opposite changing trends. In the low latitudes in Northern Hemisphere, the RZWS peak occurs in May-June and the off-peak in October-November. In the Southern Hemisphere, the RZWS off-peak occurs in May-June and the peak in October-November.

[Figure]

*Figure S3. The dynamics of the simulated RZWS for different latitudes in different months.*

**5. I don't think the WAYS is a new hydrological model because it only changed several equations and replaced a few parameters compared to the FLEX model. It is not an improvement of the FLEX model either, because it removed several important components of the original FLEX model and the manuscript failed to prove that the WAYS is better compared to FLEX after doing that.**

Thank you. In this work, we have extended a widely used lumped model, FLEX, into a distributed model that can be applied on a global scale. In addition, a climate-derived root zone storage capacity (RZSC) is integrated into WAYS to capture the spatial heterogeneity of the rooting systems. We have demonstrated the benefits of a climate-derived RZSC to the hydrological model for simulation, especially the capacity of root zone water storage (RZWS) simulation.

In fact, we did not remove any important components from the original FLEX model. Only the capillary module is disabled due to the lack of global information on the groundwater table. A detailed explanation can be found in the responses to the comments of Referee #1 (comment 6) and Referee #3 (comment 2). The corresponding revision in the manuscript can also be found there.

**6. When I saw the root zone water storage, I thought the manuscript would study vegetation. However, I did not find how they deal with vegetation transpiration. Because root zone water storage changes are largely controlled by vegetation transpiration, I don't believe the WAYS can simulate root zone water storage changes properly without considering vegetation transpiration. I share the similar concerns as other reviewers that WAYS has fatal flaws regarding this. In addition, WAYS means 'Water And ecosYstem Simulator' according to the manuscript. Without considering vegetation transpiration, WAYS cannot represent ecosystem and cannot simulate ecosystem influence on water**

**either. Thus, I believe that the manuscript title, the statement in the manuscript, and the model name are misleading and not suitable.**

Thank you. Like most of the conceptual models, WAYS considers the vegetation transpiration. It simulates the total evaporation, which consists of interception, soil evaporation and transpiration. A detailed list of all the model equations is shown in Table 1 in the manuscript. WAYS simulates the water stored in the root zone, which is a critical variable connecting the hydrology and ecology. Thus, the extension name of WAYS, i.e., 'Water And ecosYstem Simulator', will not mislead readers.

**7. The manuscript failed to report how many parameters the model has, which parameters need to be calibrated, what are the calibrated parameter values, which parameters use default values. The physical meanings of the parameters should be reported. Some parameters have their physical meanings and cannot be calibrated.**

Thank you for the comment. This comment shares the same opinion with comment 2 of Referee #1, to which a detailed interpretation and response to the comment can be found. The corresponding revision in the manuscript can also be found there.

**In sum, I am not convinced by the methodology and results, and several key issues of the study objective are not solved. I feel that this manuscript should be rejected.**

We have further improved our manuscript based on all the comments from referees and the short commenters. We believe that the manuscript has now been greatly improved.

**All the references are included in the manuscript.**

---

## Author Comment (AC7) · 26 Jul 2019

We would like to thank William Chris for his interest in this topic and for the comments to improve our manuscript. Based on the comments some calculations have been performed. Our point-by-point response to the comments is given in the following (**Comments in black**, Answers in blue and the content related to the changes in the revised manuscript are marked in orange.):

**This study tries to develop a 'global' hydrological model. The authors are lack of good understanding on hydrological processes, and the methodology they used are not appropriate at all. The authors overclaimed their contribution. The manuscript is poorly written. Some of the figures are not clear. The current manuscript cannot be accepted, and should be returned to the authors to make it a better work.**

Thank you for the comment. In this work, we have extended a widely used lumped model, FLEX, into a distributed model that can be used on a global scale. In addition, a climate-derived root zone storage capacity (RZSC) is integrated into the developed model WAYS to capture the spatial heterogeneity of the rooting systems. We demonstrate the benefit of a climate-derived RZSC to the hydrological model for simulation, especially the capacity of root zone water storage (RZWS) simulation. Thus, we believe the methodology we have used is appropriate and that the hydrological processes conceptualized in WAYS are proper. Based on the comments from the three referees as well as from the short comments, we have further improved the manuscript text as well as the figures and tables.

**My comments are as below: 1. The methodology used are not appropriate at all. The authors compared their runoff simulation against some global model simulation and composite runoff data. We all know the global runoff simulation/composite runoff data are designed for global studies, and can have very large uncertainty on each river basin. They cannot use these data to verify their simulation, especially for a study aiming to develop a 'new model'. Thus, the comparison between the authors' simulation and other runoff data that the authors used means nothing: the authors cannot claim their model is good. The authors should compare their simulation against hydrological gauge observation which is not difficult to collect at all. I doubt the authors' results. They may choose to avoid the comparison against hydrological gauge observation purposely because their model is suffering fatal flaws. For a paper developing a hydrological model, comparison against in situ gauge observation is extremely important. The authors should not skip this step. In addition, the authors compare their runoff simulation after calibrating their model, whereas the other models in ISIMIP are not calibrated. Thus, the comparisons are useless because the models in ISIMIP have large uncertainty which have already been unrevealed in several recent studies by the ISIMIP group (perhaps the authors missed these very important publications).**

Thank you. In fact, the simulated runoff of WAYS is first compared to the reference data ERA-Interim/Land runoff. The performance of WAYS in runoff simulation is evaluated based on the comparison between ERA-Interim/Land data and WAYS simulation.

Since WAYS uses the same driving data as the ISIMIP2a models and the ISIMIP2a simulations are widely discussed in many studies, an additional comparison between WAYS and the

ISIMIP2a models can provide added-value for evaluating our model. Therefore, the ISIMIP2a simulations are also shown in the results section together with the ERA-Interim/Land data. We do mention the purpose of inclusion of the ISIMIP2a simulations in the results (page 13, line 15: "Since WAYS uses the same driving data as the ISIMIP2a models and the ISIMIP2a simulations have been widely discussed in many studies (Schewe et al., 2014; Müller Schmied et al., 2016; Gernaat et al., 2017; Zaherpour et al., 2018), we also perform a comparison between WAYS and the ISIMIP2a models to further evaluate our model."). However, this is not mentioned in the validation strategy section of the manuscript. We have now further clarified this issue in the revised manuscript (please see "Authors' change in the manuscript.").

Indeed, the ISIMIP2a models are not calibrated. We have mentioned this issue in the manuscript (page 13, line 30: This result occurs partly because some of the ISIMIP2a models are not calibrated at all (Zaherpour et al., 2018), whereas WAYS is calibrated to a Composite Monthly Runoff data set that assimilates the monitored river discharge (Fekete et al., 2011).). We have revised the captions of the related figures (Figures 2 and 3 in the revised manuscript) to note this issue.

To enhance the validation part of this manuscript, we have additionally evaluated our results with observed discharge from the Global Runoff Data Centre (GRDC). Since WAYS does not currently have a native runoff routing module, a third-part runoff routing tool, CaMa-flood, is applied to route the WAYS simulated runoff (Yamazaki et al. 2011). Given that the manuscript is already quite extensive, the discharge comparison is not a direct evaluation of WAYS but of both WAYS and CaMa-Flood. This information has been added in Supplementary Information (SI). Since this comment shares the similar opinion with Referee #3 (comment 2), we would like to refer to the responses to the comments of referee #3 to avoid repetition, as the response is long. The corresponding revision to the manuscript can also be found there.

**Authors' change in the manuscript.**

**Page 11, Line 5: (the changes is marked as blue)**
In this study, the ERA-Interim/Land runoff data are used for validation of the runoff simulation, and the Normalized Difference Infrared Index (NDII) is used for the validation of the WAYS model for root zone water storage simulation. Considering the time period of coverage of both data sets (ERA-Interim/Land: 1979-2010, NDII: 2000-present) and the study period (1971-2010) of this work, the period 2001-2010 is selected as the validation period. For runoff evaluation, ISIMIP2a simulations are also included, as they use the same climate forcing as our study in the same period. The purpose of inclusion of the ISIMIP2a simulations for comparison can be found in the model evaluation section (see Section 4).

**2. The authors are lack of basic knowledge about remote sensing. The root zone can be more than 10 meters in depth. The sensors used in the NDII studies cannot penetrate the earth ground up to 10 meters, and even one or two meters are suffering large uncertainty because the attenuations of signals with increase in depth. This is why the most state-of-**

the-art soil moisture products just provide data in the surface 5/10 cm. Thus, the comparison against NDII based data is not appropriate at all. Because this paper is to develop a model, the authors should use in situ observation which is not difficult to collect. I don't understand why the authors choose to skip the comparison against gauge observation.

Thank you for the comment. As we stated in the manuscript (page 2, line 8 in the revised manuscript), remote sensing itself can only detect the soil water in the surface layer. However, NDII is not a direct observation from the satellite but a Normalized Difference Index, similar to NDWI, NDVI, and so on. It is calculated based on infrared reflectance (NIR) and shortwave infrared reflectance, and it reflects the water stress in the root zone layer and, thus, can be used as proxy data for RZWS rather than RZWS itself. The NDII-related information is interpreted in detail in Section 3.3.2 in the manuscript.

We do not compare the situ observations because there are no observations available for RZWS (Sriwongsitanon et al., 2016). The observation you mentioned is probably the soil moisture at a certain depth, which differs from RZWS.

3. The authors do not have a good understanding on hydrological processes. i). Vegetation plays a vital role in runoff variations especially in densely vegetated regions (e.g., the Amazon, Congon and some regions in the Yangtze, Mekong, Ganges, Mississippi Rivers et.al.) through the transpiration processes. At the leaf and canopy scales, the mechanisms of transpiration are also different. LAI, fPAR, CO2, wind, solar radiation, stomatal conductance all are influencing transpiration. The authors did not consider the stomatal influence at all (as shown in the Figure 1 and Table 1). Without comprehensively considering the transpiration processes, how the model developed can predict water resources availability, especially many recent studies have unravelled that the earth is greening and CO2 concentration is increasing. Thus, the model developed by the authors has fatal flaws, and this paper cannot be accepted.

Thank you for the comment. We agree with the reviewer that vegetation plays a vital role in runoff variations, especially in densely vegetated regions, and the mechanisms of transpiration are also different at the leaf and canopy scales. However, the model we developed in this study is a conceptual hydrological model with a conceptualized structure to mimic the hydrological cycle. This design differs from land surface models, dynamic vegetation models or physically based hydrological models, which could have more functions with physical meanings (Bierkens, 2015). The conceptual hydrological model, however, has its own advantages in practicability and computation efficiency (Devia et al., 2015). The transpiration is of course considered by conceptual models, while some of them calculated the total evaporation without separating the evaporation into different fluxes. Moreover, conceptual models are widely applied for water related applications, e.g., runoff simulation and water scarcity analysis, especially on a global scale (Döll et al., 2003; Döll and Fiedler, 2008; Hanasaki et al., 2008; Wang-Erlandsson et al., 2014). Thus, a well-developed conceptual model, such as WAYS, should be proper for predicting water resource availability.

In addition, the continuous greening of the earth as well as the increased $CO_2$ concentration are indeed important issues. However, they are beyond the scope of this study as they are more related to climate change analysis.

**ii). The infiltration capacity of soil plays an important role in controlling the volume of surface runoff and subsurface runoff, and also influences root zone water storage. The infiltration capacity of soil is related to soil type, and has clear physical meaning. The authors considered the infiltration as shown in the Figure 1. However, the authors did not report how they determine this important parameter value. If the authors used the values related each soil type, they did not report which soil map distribution data and which hydraulic property datasets of the soil types are used. If the authors calibrated the parameter values, the authors should be aware of that if it is appropriate to calibrate because the results may be wrong after calibrating some parameters with clear physical meaning. The authors are afraid of reporting the calibrated parameter values and the parameter ranges used in the calibration. The authors stated they calibrated their model for good runoff simulation. I am afraid that they calibrated their model for good runoff simulation with the cost of losing the physical meaning of important parameters. Perhaps the authors choose to not show the important information purposely in order to get their paper published. No, absolutely no. The authors have to show which parameters are calibrated, the parameter ranges used in calibration and calibrated parameter values.**

Thank you for the comment. The precipitation partitioning function in the WAYS model is based on a widely used beta function of the Xinanjiang model (Zhao, 1992). It is a conceptualized runoff generation function that consists of empirical parameters. The model is a conceptual model and is different from the physically based model. The model parameter must be calibrated before simulation. Indeed, the physically based model is usually run without calibration as the parameters it uses have corresponding physical meanings. However, the physical model and conceptual model are two different methods without any conflicts between each other.

Moreover, we have added a table to illustrate the parameters used as well as the parameter ranges if calibration is needed. Please refer to the changes in the revised manuscript (page 11, Table 2). We will share the calibrated parameters together with the code for the model after the paper is accepted for publication.

Since the calibrated parameters are spatially distributed and are not appropriate to show in tables, we provide the spatial patterns of two key parameters ($\beta$, $C_e$) that are calibrated, as these two parameters mostly affect the partitioning of precipitation (see Figure S21 and Figure S22). The rest of the calibrated parameters are uploaded to the response thread in a netCDF file as a supplementary document.

[Figure]

*Figure S21. The spatial distribution of the model parameter $\beta$*

[Figure]

*Figure S22. The spatial distribution of the model parameter $C_e$*

**Authors' change in the manuscript.**

**Page 12: (the following table is added)**

**Table 2.** Parameter ranges of the WAYS model

| Parameter | Range | Literature | Parameter | Range |
|---|---|---|---|---|
| $S_{i,max}$ | distributed | Wang-Erlandsson et al. (2014) | $\beta$ | (0, 2) |
| $S_{rz,max}$ | distributed | Wang-Erlandsson et al. (2016) | $C_e$ | (0.1, 0.9) |
| $R_{s,max}$ | 7/4.5/2/5 (Sand/Loam/Clay) | Döll and Fiedler (2008) | $K_f$ | (1, 40) |
| $K_s$ | 100 | Döll et al. (2003) | $K_{ff}$ | (1, 9) |
| $f_s$ | distributed | Döll and Fiedler (2008) | $S_{ftr}$ | (10, 200) |
| $F_{DD}$ | distributed | Müller Schmied et al. (2014) | $T_{lag}$ | (0, 5) |
| $T_t$ | 0 | Müller Schmied et al. (2014) | | |

**4. The model developed is not a global scale model at all. Because the authors did not use soil map and related soil hydraulic parameter values, the use of the model must rely on calibration to determine some of its parameter values on river basin scales. Therefore, it cannot be a global scale model. It is still a river basin scale model, and the authors just applied the model in several large-scale river basins (without any river basins in most of the regions of Canada, Europe, Middle East, Russia, Mongolia). The used river basins just cover a small proportion of global land surface.**

Thank you for the comment. The WAYS model actually uses many global parameters for hydrological simulation, e.g., RZSC, land cover, DEM, digital maps of the slope, soil texture, geology and permafrost information (see page 6, line 5 in the revised manuscript).

Based on the comments from the referees as well as from the short comments, additional evaluations covering large areas are included in the revised manuscript. These include discharge comparison to GRDC observation, evaporation comparison to FLUXNET2015 and LandFluxEVAL data. The results of the discharge evaluation can be found in responses to comments of referee #3 (comment 2). For the evaporation evaluation, we would like to refer to the responses to the comments of referee #3 (comment 5) to avoid repetition, as the response is long. The corresponding revision to the manuscript can also be found there.

**5. The authors claimed they used 2000 iterations to calibrate their model. However, the authors did not explain the reason. Why 2000 iterations were used?**

Thank you. In fact, the number of iterations is recommended by the author of Dynamically Dimensioned Search (DDS) algorithm (Tolson and Shoemaker, 2009). We have clarified this in the revised manuscript.

**Authors' change in the manuscript.**

**Page 13, Line 6: (the changes is marked as blue)**

The criterion of fit for calibration is the Nash-Sutcliffe efficiency coefficient (NSE), and the DDS optimization algorithm is run with 2000 iterations for each grid cell for parameter estimation, as suggested by the author of DDS (Tolson and Shoemaker, 2007).

**6. The root zone storage variations are related to ground water level dynamics. Did the model simulate the ground water level changes? Please show the simulation results.**

Thank you. The current work did not consider the groundwater level changes as well as the capillary rise due to the lack of groundwater table information. We have discussed this issue in the revise manuscript.

**Authors' change in the manuscript.**

**Page 25, Line 4: The following paragraph is inserted in the discussion part**
Moreover, the current study does not consider the groundwater access and irrigation mainly due to the lack of global information. The groundwater table information is crucial for capillary rise simulation (Vergnes et al., 2014). Capillary rise simulation without proper water table information could significantly overestimate the evaporation. Thus, the capillary rise flux is ignored in this study. A similar strategy has also been applied by other works due to the absence of the information on the global water table (Döll et al., 2003; De Graaf et al., 2015; Hanasaki et al., 2018). Observations of irrigation on the global scale are also not available (Leng et al., 2015). Although there are simulated irrigation data available on the global scale, the inherent uncertainties could be propagated in our model simulation. Therefore, irrigation is also not considered at this time. However, this neglect could potentially introduce biases into the model simulation in irrigated areas and deep rooted plant-distributed regions, as both irrigation and capillary rise are an additional supply of soil water recharge. The biases may cause an underestimation of evaporation, especially in the dry summertime (Vergnes et al., 2014). This underestimation could consequently affect the simulation of RZWS and runoff because of the interlinkage of these three elements (Rockström et al., 1999). It is found that ignoring the capillary rise could reduce soil water content in the root zone (RZWS), while the runoff will also be reduced (Vergnes et al., 2014). However, these shortcomings can be simply overcome once the global data are available.

**7. Please use scientific languages. The sub-titles of Section 2.4 and 2.5 are not appropriate in such as a scientific paper. The statements 'Fast- and Slow-' are vague.**

Thank you. We have now changed the Fast- and Slow- flow to preferential flow and matrix flow based on the related literature (Ali et al., 2018; Gao et al., 2019).

**Authors' change in the manuscript.**
The fast flow and slow flow are replaced by preferential flow and matrix flow in the entire manuscript.

**8. I agree with the reviewer 1 about the capillary mechanism which is missed by the authors. This indicates the authors are lack of good understanding on hydrological processes from another perspective. When we develop a new model, we try to incorporate new hydrological mechanism to advance our understanding on hydrological processes. However, the authors missed several very important hydrological processes which have already been recognised to be very important. Therefore, the 'developed' model cannot provide any new understanding on hydrology to us. I am afraid that the authors just copy other models' code, delete several important parts, replace a few equations and change computer language used in original code, and then the authors claim they develop a new model. No, this is not the right way to do research. I also wonder why the authors delete the capillary mechanism part from the original code. The authors should realize that they cannot just delete some codes of other's model, and make it look like a 'new model' in order to get the manuscript published. This is not real science. The authors must work hard to consider the capillary and vegetation transpiration mechanisms and using gauge data to validate their simulation. Otherwise, their model cannot be better (based on the physical processes considered) than other hundreds/thousands of models that already exist.**

Thank you. In fact, WAYS does include the capillary module from Gao et al. (2014a), a key publication on the FLEX model. At the current stage, it is, however, disabled due to the lack of global information on the groundwater table. A detailed explanation could be found in the responses to comment 6 of referee #1. The corresponding revision in the manuscript can also be found there.

**9. The manuscript is poorly written and needs to be largely reworked. There are many typos and grammar mistakes. Many sentences are vague and lack of support. The figures are not clear, e.g., Figure 4 and Figure 5, and one cannot distinguish the lines.**

Thank you. We have carefully checked the manuscript and corrected typos and grammatical mistakes. The revised manuscript has been edited by a professional academic language and manuscript service company. We have also further improved the manuscript as well as the figures and tables. Figure 4 and Figure 5 are reproduced with high resolution, and the lines are clearer now. Since short comment 10 suggested us to move some figures from the main text of the manuscript. Thus, the figures in the revised manuscript are re-sorted.

**10. Figure 2 is not your result. Please remove Figure 2. Using related references in the manuscript to refer to the data is ok.**

Thank you. Indeed, Figure 2 shows the spatial distribution of RZSC, which is obtained from Wang-Erlandsson et al. (2016). Since it a key parameter for the model we developed and spatial distribution information would be useful, we have move them to SI rather just placed them in the references. In addition, Figure 3, which shows the latitudinal averaged RZSC, has also been moved from the main text of the manuscript to SI, as it is also based on the results from Wang-Erlandsson et al. (2016).

It is also important to note that the RZSC is now updated based on the comment of Referee #3. Referee #3 suggested that RZSC should be updated by applying the Gumbel normalization, as Wang-Erlandsson et al. (2016) found that normalizing the RZSC using the Gumbel distribution by land cover type further improves performance.

**Authors' change in the manuscript.**
The Figures 2 and 3 are moved to the SI.

**All the references are included in the manuscript.**

---

## Author Comment (AC9) · 26 Jul 2019

**Supplementary Information**

**Supplementary Section: Parameter Sensitivity Analysis**

In the WAYS mode, $\beta$ and $C_e$ are two crucial parameters that control the partitioning of precipitation into evaporation and runoff, thus affecting the water balance. Due to the incredibly high computation cost, only the sensitivity of the model simulation to these two parameters are tested. First, a pixel is selected randomly from the domain to demonstrate the impacts of parameter perturbation on simulated evaporation, runoff and RZWS. For each experiment, only one parameter is perturbed, and the other one is set to the calibrated value. The calibrated value for $\beta$ and $C_e$ is 0.17 and 1.67, respectively. The parameter is perturbed within the range randomly 1000 times during the experiment. Simulations are executed from 2009 to 2010 on a daily scale, while the results are shown on a monthly scale (see Figure S1). The model is more sensitive to parameter $C_e$ than parameter $\beta$. The uncertainties caused by the parameter $C_e$ are generally larger than those caused by the parameter $\beta$, especially for RZWS. These two parameters also have complementary effects on the model simulation, causing larger uncertainties for the simulation than one parameter.

To further investigate the uncertainties stemming from parameters on a global scale, a Monte Carlo simulation of 1000 samples is performed by perturbing the two parameters simultaneously. For both parameters, the normal distribution is used for the Monte Carlo perturbation. Simulations are executed from 2001 to 2010 on a daily scale. The coefficient of variation (CV) for each pixel is then calculated, which reflects the uncertainties (De Graaf et al., 2015). A high value of CV indicates relatively higher uncertainty caused by the parameters, while a low value of CV implies the opposite. Figure S2 shows that parameter-induced uncertainties of evaporation and runoff have similar patterns, while the magnitude is slightly higher for the runoff globally. This finding is consistent with the pixel-based sensitivity test (see Figure S1). The simulated RZWS has the largest uncertainties with the Monte Carlo simulation. Additionally, the uncertainties of RZWS show the opposite trend to the uncertainties of evaporation and runoff. In the northern part of Africa, the Arabian Peninsula, northwest of China and southern part of Australia, the uncertainties in evaporation and runoff are low. However, the uncertainties in RZWS are quite large in these regions.

[Figure]

*Figure S1. Sensitivity of simulated evaporation (top), runoff (middle) and RZWS (bottom) to parameter β and Ce in a randomly selected pixel within the domain. The black solid line represents the simulation based on the calibrated value. The blue area indicates the uncertainties induced from the perturbation of the parameter 1000 times.*

[Figure]

*Figure S2. Coefficient of variation of model-simulated evaporation (top), runoff (middle) and RZWS (bottom) from 1000 Monte Carlo simulations with different parameter settings for $\beta$ and Ce.*

Since the groundwater recharge module in WAYS is based on the work of Döll and Fiedler (2008), the values are taken directly from it. These values are also used for other global groundwater recharge simulation-related works, e.g., Müller Schmied et al. (2014). However, as these are indeed empirical parameter values, uncertainty/sensitivity analysis is necessary for $R_{s,max}$. Three pixels from different soil types are selected for the $R_{s,max}$-induced uncertainty investigation. Figure S3 shows the grouped soil texture classes for this study based on the FAO Harmonized World Soil Database and the selected pixels for the

uncertainty analysis. Pixel 1, pixel 2 and pixel 3 represent the soil type of clay, loam and sand, respectively.

$R_{s,max}$ (mm/day) directly influences the matrix flow (contributes 100% to groundwater recharge with a certain time lag) based on equation 12 in Table 1 in the manuscript, as it controls the maximum groundwater recharge for different soil types. Consequently, it will also impact the preferential flow, as the runoff is partially split into matrix flow and the rest to the preferential flow. Therefore, parameter $R_{s,max}$ will have light effects on the runoff generation but could have considerable impacts on the matrix flow and preferential flow. Thus, the sensitivity of the simulated preferential flow and matrix flow to the maximum groundwater recharge $R_{s,max}$ is investigated, and the results are shown in Figure S4 (pixel with clayey soil), Figure S5 (pixel with loamy soil) and Figure S6 (pixel with sandy soil). The sensitivities of WAYS to $R_{s,max}$ are checked by perturbing the parameter. We set the simulation with soil texture-specified $R_{s,max}$ as the control run, and perturbed $R_{s,max}$ by -80%, -50%, -20%, 20%, 50% and 80% for the sensitivity test. Figure S4 (bottom plot) shows the impacts of the values of $R_{s,max}$ on the simulated daily matrix flow with the soil type of clay. With the increase in $R_{s,max}$, the simulated daily matrix flow has a higher peak, while the opposite is observed with the decrease in $R_{s,max}$. It changes the scale of the simulated matrix flow but not its shape at the daily scale. Moreover, due to the change in daily simulation, the monthly simulation of the matrix and preferential flow are affected accordingly, as seen in Figure S4 (top and middle plots). The results show that parameter $R_{s,max}$ has opposite impacts on preferential flow and matrix flow, which is logical because both are part of the runoff. A similar phenomenon is found in daily simulated time series. Thus, for pixel 2 and pixel 3, only monthly simulated matrix and preferential flow are shown to visualize the uncertainties stemming from $R_{s,max}$ for loamy and sandy soil. The simulated matrix flow time series with a decreased value of $R_{s,max}$ shows are found to have larger uncertainties than time series with an increased value of $R_{s,max}$, because the maximum value of matrix flow is not only determined by $R_{s,max}$ but also the groundwater recharge factor $f_s$.

[Figure]

*Figure S3. Grouped soil texture classes for the study based on the FAO Harmonized World Soil Database.*

[Figure]

*Figure S4. Sensitivity of the simulated preferential flow and matrix flow to the maximum groundwater recharge $R_{s,max}$ for the pixel with clayey soil.*

[Figure]

*Figure S5. Sensitivity of the simulated preferential flow and matrix flow to the maximum groundwater recharge $R_{s,max}$ for the pixel with loamy soil.*

[Figure]

*Figure S6. Sensitivity of the simulated preferential flow and matrix flow to the maximum groundwater recharge $R_{s,max}$ for the pixel with sandy soil.*

**Supplementary Section: Model Evaluation against GRDC observed discharge**

To further evaluate the model performance, we have evaluated our results with observed discharges from the Global Runoff Data Centre (GRDC). The CaMa-flood model is the only available open-source global runoff routing model (http://hydro.iis.u-tokyo.ac.jp/~yamadai/cama-flood/) that is capable of simulating backwater effects, which is important for plain regions, making it a popular choice for many studies (Hirabayashi et al., 2013; Mateo et al., 2014; Pappenberger et al., 2012).

The GRDC stations along a river were selected with interstation areas larger than 7000 km$^2$ to omit catchments with hydrological processes that are not properly represented by global hydrological models operating at a 0.5° resolution (Hunger and Döll, 2008). In total, 154 stations are selected for major river basins worldwide. For discharge simulation, the CaMa-flood is run at a 0.5° resolution to maintain consistency with the WAYS simulated runoff. The WAYS_CRU simulation is used for routing due to global coverage of the data. The discharge is simulated for the 1971-2010 period.

For the evaluation, the simulated discharge is compared with the GRDC data at each selected station depending on the data availability. Since the observations provided by GRDC are on a monthly time scale, the simulated data are also aggregated to the monthly scale for the comparison. The correlation coefficient and Nash-Sutcliffe efficiency coefficient are calculated, while the correlation coefficient between the simulated discharges and GRDC station records are visualized in Figure S7.

[Figure]

*Figure S7. The evaluation of simulated discharge by comparison with the GRDC observations. The discharge is simulated by the CaMa-flood model, and the WAYS simulated runoff based on the RZSC data ($S_{R,CRU\text{-}SM}$) is used as the input data for routing. The background of the figure is the annual averaged discharge for the 1971-2010 period. The point indicates the correlation coefficient between simulated discharge and GRDC observations. The location of the points implies the location of the GRDC station. Different colors at the points represent the magnitudes of the correlation coefficient.*

The simulation shows a generally good correlation with the GRDC observations, while poor performance in the discharge simulation is also found in a few stations. The errors between the simulated discharge and observations could be caused by both the WAYS model for runoff simulation as well as the CaMa-Flood model for runoff routing, as the CaMa-Flood model itself also shows different performances in basins across the world (Yamazaki et al., 2011). The relatively low performance of WAYS is found in middle-high latitudes compared with low-middle latitude regions. This result could be explained by the relatively simple snow-melt module in the WAYS model, which thus could consequently produce low-quality runoff for river routing in cold regions. In Australia, only two GRDC stations in the Murray Darling basin are selected for the evaluation, and the correlation coefficient between simulated discharge and GRDC station is less than 0.5, indicating the large difference between them.

Figure S8 shows the histogram of the data points within different intervals of the correlation coefficient. Only in 7.2% of the stations are the correlations between simulation and

observation less than 0.5. For more than half the stations, the correlations are higher than 0.7. The results show a generally good correspondence between the simulated and observed discharge. The generally good performance in the discharge simulation confirms the strong capacity of WAYS for runoff generation.

[Figure]

*Figure S8. Histogram showing the percentage of data points within different intervals of the correlation coefficient.*

**Supplementary Section: Evaporation Evaluation against LandFluxEVAL data**

The model simulation is further compared to a gridded data set, LandFluxEVAL data, for evaporation evaluation. The LandFluxEVAL data are a merged benchmark synthesis product of evaporation on a global scale and a combination of land-surface model simulations, remote sensing products, reanalysis data and ground observation data (Mueller et al., 2013). The LandFluxEVAL data are used in many studies as reference data for evaporation evaluations (Lorenz et al., 2014; Martens et al., 2017; Wartenburger et al., 2018). Since the LandFluxEVAL data are only available at 1-degree spatial resolution, the WAYS simulated evaporation is aggregated to 1 degree to match the resolution of the reference data. The evaluation is executed for 1989-2005 based on the availability of the LandFluxEVAL data. For the spatial evaluation, the WAYS simulation based on RZSC ($S_{R,CRU-SM}$) is used due to the global coverage of the RZSC product. For latitudinal comparison, both runs of WAYS simulated evaporation are used.

A promising relationship between WAYS simulated evaporation and LandFluxEVAL evaporation is found both in spatial pattern and in latitudinal average (see Figure S9). The generally high correlation coefficient (Figure S9, a) confirms the good performance of the WAYS model. However, relatively poor performance is also found in some regions in

Europe, North America and South America (Amazon basin). It can also be seen that the spatial pattern of WAYS simulated annual averaged evaporation follows that of LandFluxEVAL data, while overestimations are found in regions, e.g., the Amazon basin and southeast Asia. The latitudinal evaluation shows that both WAYS simulations (WAYS_CRU and WAYS_CHIRPS) display a slight overestimation.

[Figure]

*Figure S9. Validation results of the evaporation of WAYS simulation against the LandFluxEVAL data (1989-2005). (a) The calculated correlation coefficient between LandFluxEVAL data and WAYS simulation, (b) the annual averaged evaporation of LandFluxEVAL data, (c) the annual averaged evaporation of WAYS simulation based on RZSC $S_{R,CRU-SM}$, and (d) the comparison of the averaged latitudinal evaporation for WAYS model runs as well as the LandFluxEVAL data.*

**Supplementary Section: Investigation of the spatial pattern of simulated RZWS**

Figure S10 shows the spatial distribution of the annual averaged RZWS simulated by WAYS in the 1971-2010 period. It shows that RZWS is high in low-middle latitudes, while RZWS in middle-high regions is relatively low. RZWS represents the water content that is stored in the root zone as well as the available water for plants. Therefore, for lower or middle latitudes, the available water for plants is relative higher than for high-latitude areas.

To further investigate the soil water condition, we have calculated the root zone soil wetness by dividing RZWS by RZSC, and the results are shown in Figure S11. The root zone soil wetness follows the spatial patterns of RZWS in general. However, differences can also be found in regions such as Europe, South America and the eastern part of North America.

[Figure]

*Figure S10. The spatial distribution of the annual averaged RZWS simulated by WAYS in the 1971-2010 period.*

[Figure]

*Figure S11. The spatial distribution of the annual averaged root zone soil wetness in the 1971-2010 period.*

The simulated RZWS is shown in Figure S12 to reveal dynamics in different latitudes and in different months. The latitudinal averaged RZWS again confirms that the RZWS are relatively plentiful compared with the high latitudes. However, a decreasing trend can also be found by moving from a low latitude to the equator. Two simulations of the WAYS model show similar fluctuation along the latitudes, while the simulation with $S_{R,CHIRPS}$ is slightly higher. Figure S12 shows that RZWS in low-middle latitudes has a larger monthly variation than in other regions, while the Northern Hemisphere and Southern Hemisphere show opposite changing trends. In the low latitudes in Northern Hemisphere, the RZWS peak occurs in May-June and the off-peak in October-November. In the Southern Hemisphere, the RZWS off-peak occurs in May-June and the peak in October-November.

[Figure]

*Figure S12. The dynamics of the simulated RZWS for different latitudes in different months.*

**Supplementary Figures:**

(a)

(b)

[Figure]

Root Zone Storage Capacity (mm)

*Figure S13. Two global root zone storage capacity products at 0.5 degrees: (a) $S_{R,CRU-SM}$; (b) $S_{R,CHIRPS-CSM}$. Figures are produced based on the data provided by Wang-Erlandsson et al. (2016). Gray color indicates no data.*

[Figure]

Figure S14. Mean latitudinal root zone storage capacity of $S_{R,CRU-SM}$ and $S_{R,CHIRPS-CSM}$

[Figure]

*Figure S15. Evaluation of the simulated evaporation by comparison to the flux tower data (FLUXNET2015). The flux tower data are converted from latent heat to evaporation before the comparison. The background of the figure is the annual averaged evaporation for the 1971-2010 period. The color at the points indicates the correlation coefficient between the simulated discharge and flux tower observation. The location of the points implies the location of the flux tower.*

[Figure]

*Figure S16. Histogram showing the percentage of data points within different intervals of the correlation coefficient.*

[Figure]

*Figure S17. Spatial distribution of uncertain RZSC (SR,LOOKUP-TABLE).*

[Figure]

Figure S18. The difference between $S_{R,CRU-SM}$ and $S_{R,LOOKUP-TABLE}$ ($S_{R,LOOKUP-TABLE}$ - $S_{R,CRU-SM}$).

[Figure]

*Figure S19. Latitudinal averaged RZSC of different products.*

[Figure]

*Figure S20. The impacts of RZSC on the model simulation. Blue color indicates the improvement of the simulated results by replacing the uncertain RZSC ($S_{R,LOOKUP-TABLE}$) with satellite data-derived RZSC ($S_{R,CRU-SM}$), while red color implies the opposite. (a) The result for runoff and the reference data for comparison is ERA-Interim/Land data (2001-2010, monthly), (b) the result for evaporation and the reference data for comparison is LandFluxEVAL data*

*(1989-2005, monthly), and (c) the result for RZWS and the reference data is NDII data (2001-2010, 8 days).*

[Figure]

*Figure S21. The spatial distribution of model parameter β.*

[Figure]

*Figure S22. The spatial distribution of model parameter $C_e$.*

**References:**

Döll, P. and Fiedler, K.: Global-scale modeling of groundwater recharge, Hydrol. Earth Syst. Sci., 12(3), 863–885, doi:10.5194/hess-12-863-2008, 2008.

De Graaf, I. E. M., Sutanudjaja, E. H., Van Beek, L. P. H. and Bierkens, M. F. P.: A high-resolution global-scale groundwater model, Hydrol. Earth Syst. Sci., 19(2), 823–837, doi:10.5194/hess-19-823-2015, 2015.

Hirabayashi, Y., Mahendran, R., Koirala, S., Konoshima, L., Yamazaki, D., Watanabe, S., Kim, H. and Kanae, S.: Global flood risk under climate change, Nat. Clim. Chang., 3(9), 816–821, 2013.

Hunger, M. and Döll, P.: Value of river discharge data for global-scale hydrological modeling, Hydrol. Earth Syst. Sci., 12(3), 841–861, doi:10.5194/hess-12-841-2008, 2008.

Lorenz, C., Kunstmann, H., Devaraju, B., Tourian, M. J., Sneeuw, N. and Riegger, J.: Large-Scale Runoff from Landmasses: A Global Assessment of the Closure of the Hydrological and Atmospheric Water Balances*, J. Hydrometeorol., 15(6), 2111–2139, doi:10.1175/JHM-D-13-0157.1, 2014.

Martens, B., Miralles, D. G., Lievens, H., Van Der Schalie, R., De Jeu, R. A. M., Fernández-Prieto, D., Beck, H. E., Dorigo, W. A. and Verhoest, N. E. C.: GLEAM v3: Satellite-based land evaporation and root-zone soil moisture, Geosci. Model Dev., 10(5), 1903–1925, doi:10.5194/gmd-10-1903-2017, 2017.

Mateo, C. M., Hanasaki, N., Komori, D., Tanaka, K., Kiguchi, M., Champathong, A., Sukhapunnaphan, T., Yamazaki, D. and Oki, T.: Assessing the impacts of reservoir operation

to floodplain inundation by combining hydrological, reservoir management, and hydrodynamic models, Water Resour. Res., 50(9), 7245–7266, doi:10.1002/2013WR014845, 2014.

Mueller, B., Hirschi, M., Jimenez, C., Ciais, P., Dirmeyer, P. A., Dolman, A. J., Fisher, J. B., Jung, M., Ludwig, F., Maignan, F., Miralles, D. G., McCabe, M. F., Reichstein, M., Sheffield, J., Wang, K., Wood, E. F., Zhang, Y. and Seneviratne, S. I.: Benchmark products for land evapotranspiration: LandFlux-EVAL multi-data set synthesis, Hydrol. Earth Syst. Sci., 17(10), 3707–3720, doi:10.5194/hess-17-3707-2013, 2013.

Müller Schmied, H., Eisner, S., Franz, D., Wattenbach, M., Portmann, F. T., Flörke, M. and Döll, P.: Sensitivity of simulated global-scale freshwater fluxes and storages to input data, hydrological model structure, human water use and calibration, Hydrol. Earth Syst. Sci., 18(9), 3511–3538, doi:10.5194/hess-18-3511-2014, 2014.

Pappenberger, F., Dutra, E., Wetterhall, F. and Cloke, H. L.: Deriving global flood hazard maps of fluvial floods through a physical model cascade, Hydrol. Earth Syst. Sci., 16(11), 4143–4156, doi:10.5194/hess-16-4143-2012, 2012.

Wang-Erlandsson, L., Bastiaanssen, W. G. M., Gao, H., Jägermeyr, J., Senay, G. B., van Dijk, A. I. J. M., Guerschman, J. P., Keys, P. W., Gordon, L. J. and Savenije, H. H. G.: Global root zone storage capacity from satellite-based evaporation, Hydrol. Earth Syst. Sci., 20(4), 1459–1481, 2016.

Wartenburger, R., Seneviratne, S. I., Hirschi, M., Chang, J., Ciais, P., Deryng, D., Elliott, J., Folberth, C., Gosling, S. N., Gudmundsson, L., Henrot, A.-J., Hickler, T., Ito, A., Khabarov, N., Kim, H., Leng, G., Liu, J., Liu, X., Masaki, Y., Morfopoulos, C., Müller, C., Schmied, H. M., Nishina, K., Orth, R., Pokhrel, Y., Pugh, T. A. M., Satoh, Y., Schaphoff, S., Schmid, E., Sheffield, J., Stacke, T., Steinkamp, J., Tang, Q., Thiery, W., Wada, Y., Wang, X., Weedon, G. P., Yang, H. and Zhou, T.: Evapotranspiration simulations in ISIMIP2a—Evaluation of spatio-temporal characteristics with a comprehensive ensemble of independent datasets, Environ. Res. Lett., 13(7), 075001, doi:10.1088/1748-9326/aac4bb, 2018.

Yamazaki, D., Kanae, S., Kim, H. and Oki, T.: A physically based description of floodplain inundation dynamics in a global river routing model, Water Resour. Res., 47(4), 1–21, doi:10.1029/2010WR009726, 2011.

---

## Author Comment (AC10) · 26 Jul 2019

Please see the calibrated parameters and addition results in Supplement.

Please also note the supplement to this comment:
https://www.geosci-model-dev-discuss.net/gmd-2019-52/gmd-2019-52-AC10-supplement.zip